# How much water vapour does the Tibetan Plateau release into the atmosphere?

Chaolei Zheng[1], Li Jia[1], Guangcheng Hu [1], Massimo Menenti[1, 2], Joris Timmermans[2]

[1] Key Laboratory of Remote Sensing and Digital Earth, Aerospace Information Research Institute, Chinese Academy of Sciences, Beijing 100101, China

[2] Faculty of Civil Engineering and Geosciences, Delft University of Technology, Delft, the Netherlands

*Correspondence to*: Chaolei Zheng (zhengcl@aircas.ac.cn)

**Abstract.** Water vapour flux, expressed as evapotranspiration (ET), is critical for understanding the earth climate system and the complex heat/water exchange mechanisms between the land surface and the atmosphere in the high-altitude Tibetan Plateau (TP) region. However, the performance of ET products over the TP has not been adequately assessed, and there is still considerable uncertainty in the magnitude and spatial variability in the water vapour released from the TP into the atmosphere. In this study, we evaluated 22 ET products in the TP against *in-situ* observations and basin-scale water balance estimations. This study also evaluated the spatiotemporal variability of the total vapour flux and of its components to clarify the vapour flux magnitude and variability in the TP. The results showed that the remote sensing high-resolution global ET data from ETMonitor and PMLV2 had a high accuracy, with overall better accuracy than other global and regional ET data with fine spatial resolution (~1km), when comparing with *in-situ* observations. When compared with water balance estimates of ET at the basin scale, ETMonitor and PMLV2 at finer spatial resolution and GLEAM and TerraClimate at the coarse spatial resolution showed good agreement. Different products showed different patterns of spatiotemporal variability, with large differences in the central to western TP. The multi-year and multi-product mean ET in the TP was 333.1 mm/yr with a standard deviation of 38.3 mm/yr. The ET components (i.e., plant transpiration, soil evaporation, canopy rainfall interception evaporation, open water evaporation, and snow/ice sublimation) available from some products were also compared, and the contribution of these components to total ET varied considerably even in cases where the total ET from different products was similar. Soil evaporation accounts for most of the total ET in the TP, followed by plant transpiration and canopy rainfall interception evaporation, while the contributions from open water evaporation and snow/ice sublimation cannot be negligible.

## 1 Introduction

The Tibetan Plateau (TP) is also known as the 'Water Tower of Asia' as it is the source of 10 major rivers. Significant changes in the natural and social environments of the TP have occurred over the past 50 years (e.g., temperatures have warmed twice as much as the global average over the same period), and there is considerable uncertainty about further environmental change (Immerzeel et al., 2020; Yang et al., 2014; Chen et al., 2015). Observations have shown significant changes in the environment, such as increased precipitation, decreased wind speed and snow days, increased surface solar radiation, thawing of permafrost,

melting of glaciers and snow, and greening of vegetation (Yao et al., 2012; Yang et al., 2014; Kuang and Jiao et al., 2016; Bibi et al., 2018; Z. Wang et al., 2018). These changes have significant impacts on human living conditions, as well as economic and social development (Wei et al., 2022; Yang et al., 2022). The TP can also affect the atmospheric circulation by altering the release of sensible and latent heat, which has a significant implication on the climate in China, Asia, and globally (Wu et al., 2016).

Water vapour flux, expressed as evapotranspiration (ET), is critical for understanding the earth climate system and the complex heat/water exchange mechanisms between the land surface and the atmosphere in the high-altitude TP region (Shen et al., 2015; Yang et al., 2023). It is important as a covariate of the water and heat fluxes in the soil-vegetation-atmosphere system in the TP and as an indicator of climate and land surface changes (Sun et al., 2023; Yang et al., 2023; Zhang et al., 2010). The TP is experiencing faster water phase transitions, with more solid water becoming liquid water through melting glacier/snow and more liquid water vaporized through ET (Z. Li et al., 2019; Yao et al., 2019). Accurate estimation of ET at a large scale in the TP has always been challenging due to the high heterogeneity and the complex topography. The TP is rich in land cover types, including grasslands, deserts, lakes, forests, glaciers, snow, and so on. The dynamics and thermodynamics of the subsurface vary greatly among different climate types, making it a great challenge to conduct large-scale studies of ET processes on TP and explore the governing mechanism and feedback effect on the climate system and hydrological processes. In addition, the harsh natural conditions and ecological environment of the plateau make ground-based observations difficult, and the high cost of instrumentation and routine maintenance have resulted in a scarcity of ET stations on the TP and a relatively short time series of observations (Ma et al., 2020).

Land surface models (LSMs) and climate reanalysis have been widely used to estimate ET, but generally at coarse spatial resolutions (e.g., 0.25°) and suffer from large cumulative errors due to many factors, e.g., the uncertainty of forcing and model parametrization, surface heterogeneity, etc. (Chen et al., 2019; Khan et al., 2020; X. Li et al., 2019). In contrast, ET estimation based on satellite remote sensing observations, which allow high-resolutions estimation, has clear advantages, especially in the spatially heterogeneous regions of the TP (Ma et al., 2006; Jia et al., 2018; Zheng et al., 2016). In recent decades, remote sensing-based ET datasets have improved significantly and several regional and global high-quality ET datasets have been produced (e.g., Chen et al., 2021; Martens et al., 2017; Elnashar et al., 2021; Jia et al., 2018). For example, the validation results based on the global flux network show that the PMLV2 and ETMonitor global ET products have good accuracy (with RMSE<1mm/d) (Zhang et al., 2019b; Zheng and Jia, 2020). These improved ET datasets also have many advantages, e.g., the ability to distinguish different components (vegetation transpiration, soil evaporation, and canopy rainfall interception loss), higher spatial resolution (e.g., 1-km), better performance in the heterogeneous land surface, and their application to the TP deserves further attention. However, previous studies also found significant differences between different products, such as different magnitudes of the annual mean ET in the TP ranging from 294 mm/yr to 543 mm/yr (Chen et al., 2024; Wang et al., 2020; Yuan et al., 2024; Zhang et al., 2018), and diverse trends of ET depending on the adopted datasets and study periods (Chen et al., 2024; Ma and Zhang, 2022; Wang et al., 2022). The contributions of different processes (e.g., plant transpiration,

soil evaporation, water evaporation from canopy intercepted rainfall and open water bodies) to the total water flux by different products also vary, most likely due to the theoretical and technological differences in different models and driving factors (Chen et al., 2021; Cui et al., 2020; Hu et al., 2009; Zhu et al., 2021, Ma and Zhang, 2022; Miralles et al., 2020), Recent studies on lake water evaporation suggest that it accounts for about 4%-8% of the total annual ET from the whole TP (Wang et al., 2020; Chen et al., 2024). The comparison of different ET products certainly contributes to the understand of the ET process in the TP, as well as the magnitude of ET and its spatiotemporal variability, including the ET components. There is also a need to enhance research on the whole TP as a region, and improve the understanding of the evolution of the water cycle and eco-hydrological processes in the TP (W. Wang et al., 2018).

The performance of an ET product is related to both the adopted algorithms and the forcing variables (Mueller et al., 2013; Wang et al., 2022). The global ET datasets based on satellite remote sensing have been criticized for the lack of adaptation to the specificity of the TP environment and the uncertainty inherited from the input global meteorological reanalysis data, which may lead to a large uncertainty in the direct application of the global ET datasets for studies in the TP (Zou, 2020; Song et al., 2017; Chang et al., 2018; Xue et al., 2013). These evaluations have generally been based on either *in-situ* measurements using eddy covariance systems or the basin-scale ET estimates using water balance method, which represents the surface net liquid water flux at different scales, while ET products are estimates of the upward water vapour flux, which contributes to the uncertainty. Recently, Chen et al. (2024) evaluated several ET products with spatial resolutions ranging from 1km to 50km against site-scale eddy covariance observations. It is important to note that the observations from tower-based eddy covariance systems have a very small footprint (approximately several hundred metres depending on weather conditions), and direct comparison of site-scale observations with the coarse-resolution ET products (e.g., 25km) is problematic due to the severe problem of spatial mismatch. In order to increase the credibility of currently available ET products, this study will undertake a more comprehensive evaluation, taking into account both *in-situ* observations and basin-scale measurements.

The following questions emerge from this brief literature review: 1) How accurate are these improved ET products, and how well do different products capture the magnitude and variability of ET in the TP? 2) How much water is vaporized in the TP and which processes, e.g., plant transpiration, soil evaporation, snow/ice sublimation, play a significant role? Answering these questions would reveal the strengths and weaknesses of different ET products and address the knowledge gaps on relevant processes in the TP, which are fundamental for various scientific and practical purposes.

The aim of this paper is to clarify the magnitude and variability of ET in the TP by assessing the accuracy and spatiotemporal variability of ET in the TP according to commonly available gridded products. Specifically, the main objectives are 1) to estimate the absolute uncertainties of individual ET products using flux tower data and water balance estimates; 2) to evaluate and compare the spatiotemporal variability of total ET and its components from different ET products.

## 2 Methodology and Data

### 2.1 Study region

The Tibetan Plateau (25-40°N, 70-105°E) is the highest elevated region in the world, covering an area of approximately 3.0 million km², with most areas above 2,500 meters (Figure 1). It has complex climatic regimes, ranging from a humid climate with an aridity ratio less than 0.3 to hyper-arid climate with an aridity ratio larger than 3 (Feng et al., 2024). The climate is influenced by both westerly and the Asian monsoon, which is also enhanced by the thermal forcing of the TP (Zhou et al., 2009; Wu et al., 2012; Yang et al., 2014). Influenced by multiple sources of water vapor through atmospheric circulation and alpine terrain, its precipitation shows spatial variability with the average annual precipitation gradually increasing from less than 50 mm/yr in the northwest to more than 1000 mm/yr in the southeast, and most of the precipitation is concentrated in summer (Jiang et al., 2023). The TP is also known for its extensive snow and glacier cover, with a total glacier area of approximately 50,000 km² (Yao et al., 2007) and 77% of the area above 6000 m is covered by snow (Chu et al., 2023). The main land cover types are forest, grassland, bare soil, glaciers and snow (Supplementary Figure S1). The water supply for the dense river network on the TP, which includes the headwaters of five major Asian rivers, is mainly precipitation and meltwater from glaciers and snowpack.

The TP region consists of 12 subregions: Hexi, Tarim, Qaidam, Upper Amu Darya, Inner TP, Upper Yellow, Upper Yangtze, Upper Salween, Upper Mekong, Upper Brahmaputra, Upper Ganges, and Upper Indus (Figure 1). The first five subregions, Hexi, Tarim, Qaidam, Amu Darya, and Inner TP, are located in the northern, western, and central parts of the TP and receive relatively low precipitation. The remaining watersheds receive high precipitation due to the monsoons originating from the Arabian Sea, the South China Sea, and the Western Pacific, and extremely high annual precipitation (>1000 mm/yr) is found in the Upper Salween, Upper Brahmaputra, and Upper Ganges river basins.

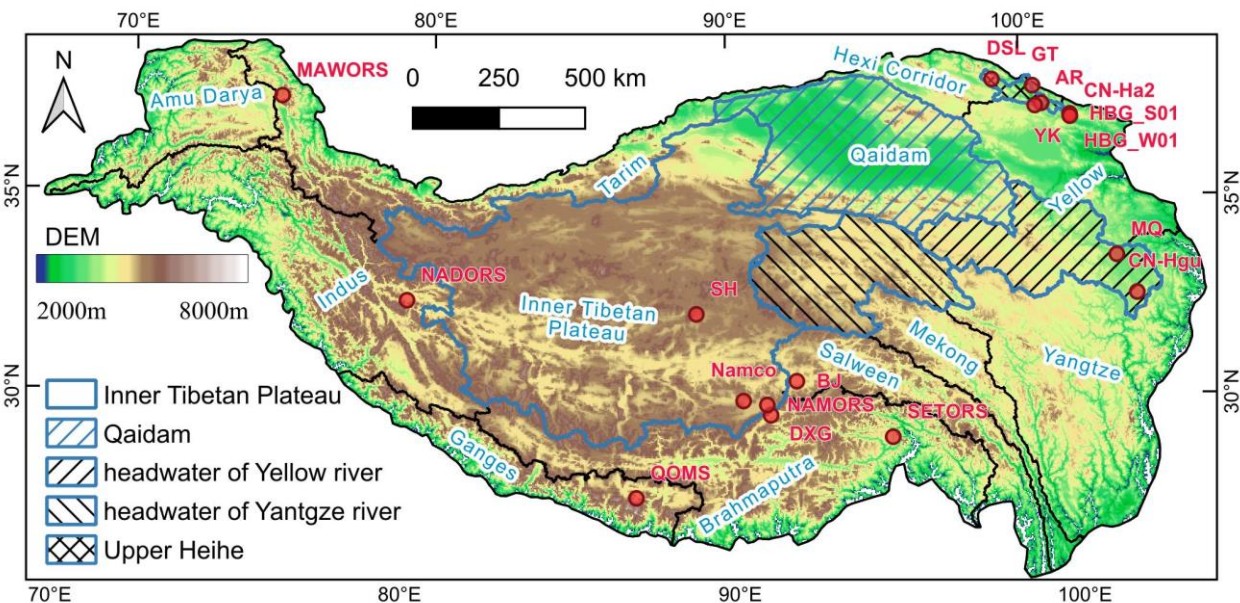

**Figure 1: Location of the selected ground flux tower observation sites and major river basins in the TP, with elevation shown as background. The selected basins where the evaluation of ET products using water balance-based data was carried out are also shown.**

## 2.2 Data sources

### 2.2.1 Flux tower data

To validate ET at high spatial resolution (≤ 1km), measurements of near-surface turbulent fluxes by the eddy-covariance method at 18 flux towers were collected. The measurements were aggregated to total monthly ET to carry out the evaluation study. Figure 1 presents the spatial distribution of these sites, and the details are provided in Table 1. The quality of flux observation data at each site was evaluated after data screening, and only reliable observations were selected following the methodology described by Zheng et al. (2022). The sites, where gap-filled daily or monthly ET data with reliable quality were already available, i.e., DXG, HBG-S01, HBG-W01, CN-HaM, CN-Hgu, SH and Maqu, were directly adopted without further modifications. For sites that provide half-hourly or hourly data, the observed latent heat flux data were gap-filled after energy closure correction, and this includes six sites (BJ, NADORS, SETORS, QOMS, NAMORS, MAWORS) from the Tibetan Observation and Research Platform (Ma et al., 2020; 2008), four sites from the Heihe Integrated Observatory Network (Liu et al., 2018; Li et al., 2009), and our own site at Namco. The Bowen ratio energy balance correction method preserves the Bowen ratio by attributing the residual term of the energy balance to the latent heat flux and sensible heat flux (Twine et al., 2000; Foken, 2008; Chen et al., 2014). The corrected half-hourly or hourly LE data was then averaged to obtain daily ET values, and only the days with more than 80% of the hourly flux were retained as valid observations. The missing daily ET values were further filled using the constant reference evapotranspiration fraction method (Jiang et al., 2022). The monthly ET was finally

calculated by accumulating the daily ET values, and those months with less than 50% valid daily ET values were treated as missing values. The missing data was not further filled, and it was not used for validation to avoid the impact of uncertainty introduced by gap-filling.

**Table 1: List of ground flux tower observation sites.**

| Site Code | Site Name | Latitude/Longitude | Elevation | Land Cover | Periods | Sources / reference |
|---|---|---|---|---|---|---|
| MAWORS | TORP MAWORS | 38.41 °N, 75.05 °E | 3668 | Desert steppe | 2012-2016 | Ma et al., 2020 |
| NADORS | TORP NADORS | 33.39 °N, 79.70 °E | 4270 | Desert steppe | 2010–2018 | Ma et al., 2020 |
| NAMORS | TORP NAMORS | 30.77 °N, 90.96 °E | 4730 | Alpine steppe | 2008–2018 | Ma et al., 2020 |
| QOMS | TORP QOMS | 28.36 °N, 86.95 °E | 4298 | Desert steppe | 2007–2018 | Ma et al., 2020 |
| SETORS | TORP SETORS | 29.77 °N, 94.74 °E | 3327 | Alpine meadow | 2007–2018 | Ma et al., 2020 |
| BJ | TORP BJ | 31.37 °N, 91.90 °E | 4509 | Sparse alpine meadow | 2010–2016 | Ma et al., 2020 |
| SH | TORP Shuanghu | 33.21 °N, 88.83 °E | 4947 | Alpine steppe | 2013–2018 | Ma et al., 2015 |
| ARS | HiWATER A'rou | 38.05 °N, 100.46 °E | 3033 | Dense alpine meadow | 2008–2018 | Liu et al., 2018 |
| DSL | HiWATER Dashalong | 38.84 °N, 98.94 °E | 3739 | Alpine meadow | 2013-2018 | Liu et al., 2018 |
| YK | HiWATER Yakou | 38.01 °N, 100.24 °E | 4148 | Alpine steppe | 2014-2018 | Liu et al., 2018 |
| GT | HiWATER Guantan | 38.53 °N, 100.25 °E | 2835 | Needleleaf forest | 2009-2011 | Li et al., 2009 |
| DXG | ChinaFLUX Dangxiong | 30.49 °N, 91.07 °E | 4333 | Alpine meadow | 2004–2010 | Yu et al., 2006 |
| HBG-S01 | ChinaFLUX Haibei grassland | 37.67 °N, 101.33 °E | 3358 | Dense alpine meadow | 2003-2010 | Yu et al., 2006 |
| HBG-W01 | ChinaFLUX Haibei wetland | 37.61 °N, 101.33 °E | 3357 | Alpine wetland | 2004-2009 | Zhang et al., 2020 |
| CN-Ha2 | FLUXNET Tibet Hai-bei Alpine | 37.37 °N, 101.18 °E | 3824 | Alpine meadow | 2002-2004 | Kato et al., 2004 |
| CN-Hgu | FLUXNET-CH4 CN-Hgu Hongyuan | 32.85 °N, 102.59 °E | 3500 | Alpine meadow | 2015-2017 | Niu and Chen, 2020 |
| MQ | Maqu site | 33.89 °N, 102.14 °E | 3423 | Dense alpine meadow | 2013-2016 | Shang et al., 2015 |
| Namco | Namco site | 30.89 °N, 90.24 °E | 4760 | Alpine steppe | 2019-2021 | this study |

**2.2.2 Water balance-based ET data**

We also collected monthly water balance-based evapotranspiration ($ET_{wb}$) from other studies at the basin-scale to evaluate the accuracy of ET data products at monthly scale. Compared to the flux tower data, $ET_{wb}$ can also be used to evaluate the products with coarse spatial resolution (≥5km). The monthly $ET_{wb}$ may also contain uncertainties due to propagated errors from

precipitation and water storage, although $ET_{wb}$ is often considered as the 'ground truth' for validating basin-wide ET estimates. In total, monthly $ET_{wb}$ from five river basins were extracted from previous studies (Ma and Zhang., 2022; Wang et al., 2021), including the headwaters of the Yellow basin (HYE), the headwaters of Yangtze basin (HYA), the Inner TP (INTP), Qaidam (QDM) basins, and the upper Heihe basin (UH), as shown in Figure 1 and Table 2.

**Table 2: Basins with water balance-based ET data for validation.**

| Basin name | Periods | Area (km$^2$) | Runoff gauging station | Description of the dataset used for water balance-based ET estimation | Sources /reference |
|---|---|---|---|---|---|
| Headwaters of Yellow basin (HYE) | 2003-2015 | 122,890 | Tangnaihai | Precipitation was from the ensemble mean of CMFD (https://doi.org/10.11888/AtmosphericPhysics.tpe.249369.file), CN05.1 (http://data.cma.cn), and MSWEP (http://www.gloh2o.org/mswep/). Terrestrial water storage changes were derived from the Gravity Recovery and Climate Experiment (GRACE) (https://grace.jpl.nasa.govs/). Monthly $ET_{wb}$ was turned into zero whenever it is negative, and ET from lakes was excluded. | Ma and Zhang., 2022 |
| Headwaters of Yangtze basin (HYA) | 2003-2015 | 140,270 | Zhimenda | | |
| Inner Tibet Plateau (INTP) | 2003-2015 | 708,252 | - (endorheic river) | | |
| Qaidam (QDM) | 2003-2015 | 253,252 | - (endorheic river) | | |
| Upper Heihe basin (UH) | 2004-2008 | 10,011 | Yingluoxia | Precipitation was from MSWEP after comparing with five datasets. Terrestrial water storage changes were derived from GRACE. | Wang et al. 2022 |

### 2.2.3 ET products

This study examined 22 ET datasets (including 20 global datasets and 2 regional datasets) (Table 3), and detailed descriptions of ET data can be found in Appendix I in Supplementary materials. Among these, 7 datasets were at high spatial resolution (≤1km), including ETMonitor (Zheng et al., 2022), MOD16 (Mu et al., 2011), MOD16-STM (Yuan et al., 2021), the Penman–
150 Monteith–Leuning Version 2 (PMLV2) (Zhang et al., 2019), the operational Simplified Surface Energy Balance (SSEBop) (Senay et al., 2020), GLASS (Yao et al., 2014), and SynthesisET (Elnashar et al., 2021). Most of these high-resolution ET datasets used different variables or indices from Moderate Resolution Imaging Spectroradiometer (MODIS) as main inputs. Two products (GLASS, SynthesisET) are ensemble ET products generated by fusing other ET models or datasets. Remote sensing ET datasets with coarse resolution were also collected, including Thermal Energy Balance (EB) ET (Chen et al., 2021),
Breathing Earth System Simulator version 2 (BESSv2) (Li et al., 2023), GLEAM (version 3.5a based on satellite and reanalysis data with long-term coverage, and version 3.5b based on mainly satellite data) (Martens et al., 2017), and FLUXCOM (RS version using MODIS remote sensing data as input, and RS_METEO version using remote sensing and meteorological data as input) (Jung et al., 2019). MOD16-STM and PMLV2-Tibet are regional ET datasets that were calibrated against ground-based eddy-covariance measurements on the TP. MOD16-STM is an enhanced version of the MOD16 algorithm by redefining
the transpiration and soil evaporation module (Yuan et al., 2021), while PMLV2-Tibet is a calibrated version of PMLV2 (Ma and Zhang, 2022). We also collected some ET products based on land surface models and climate reanalysis datasets, including

calibration-free complementary relationship (CR) ET (Ma et al., 2021), TerraClimate (Abatzoglou et al., 2018), MERRA2 (Gelaro et al., 2017), ERA5 (Hersbach et al., 2020), ERA5-Land (Muñoz-Sabater et al., 2021), GLDAS-VIC (Rodell et al., 2004), GLDAS-Noah (Rodell et al., 2004), GLDAS-CLSM (B. Li et al., 2019). In summary, among these evaluated ET prod-

165 ucts, there are 14 products that primarily use remote sensing products, including 2 products (SSEBop and EB) based on land surface temperature (LST), 8 products (ETMonitor, MOD16, MOD16-STM, PMLV2, PMLV2-Tibet, GLEAMv35a, GLEAMv35b, BESSv2) based on PM-types models (including Penman-Monteith equation, Priestley-Taylor equation, Shuttleworth-Wallace equation), 4 products (FLUXCOM-RS, FLUXCOM-RS-METEO, GLASS, SynthesisET) based on data-driven methods. Among the 8 PM-type models, there are 3 models that incorporate soil moisture to account for the influence

of available soil water on ET, including ETMonitor, GLEAMv35a, and GLEAMv35b.

All products were temporally aggregated to monthly total ET from their native temporal resolutions prior to evaluation. For the daily resolution products, simple summation operations were performed to obtain the monthly ET values. For 8-day reso-lution data, a mean daily ET value was first estimated with the available data in that month, and the monthly ET value was then obtained by multiplying the mean daily values by the number of days in the month.

**Table 3: List of ET products evaluated in this study.**

| Prod-ucts | Tem-poral resolu-tion | Spatial resolu-tion | Temporal coverage | Basic principle or algo-rithm | Main forcing data | ET com-po-nents | Valida-tion method | Refer-ence |
|---|---|---|---|---|---|---|---|---|
| **ETMon itor** | Daily | 1km | 2000-2021 | Shuttleworth–Wallace model combined with Jar-vis-type method for Ec and Es, revised Gash model for Ei, Penman-Monteith equa-tion for Ew and Ess. | ERA5 meteorological data, GLASS (LAI, FVC, al-bedo), MODIS land cover, dynamic water and snow cover, downscaled ESA-CCI soil moisture. | Ec, Es, Ei, Ew, Ess | ground observa-tion and $ET_{wb}$ | Zheng et al., 2022 |
| **MOD16** | 8-day | 500m | 2000- pre-sent | MOD-PM based algorithm for vegetation covered re-gion. | NASA GMAO meteo-rological data, MODIS (land cover, LAI). | Ec, Es, Ei | ground observa-tion and $ET_{wb}$ | Mu et al., 2011 |
| **PMLV2** | 8-day | 500m | 2002-2019 | Penman-Monteith-Leuning model V2 using remote-sensing as input. | GLDAS meteorological data, MODIS (land cover, LAI, albedo, emissivity). | Ec, Es, Ei, Ew | ground observa-tion and $ET_{wb}$ | Zhang et al., 2019 |
| **SSEBop** | 10-day | 1km | 2002-2019 | Operational Simplified Surface Energy Balance using satellite psychromet-ric principle. | Daymet Ta, and GLDAS PET data, MODIS (NDVI, LST, albedo). | - | ground observa-tion and $ET_{wb}$ | Senay et al., 2020 |
| **GLASS** | 8-day | 1km | 2000-2018 | Bayesian multi-model en-semble of different ET products. | MOD16 ET, PT-JPL ET, and other ET datasets | - | ground observa-tion and $ET_{wb}$ | Yao et al., 2014 |

| | | | | | | | | |
|---|---|---|---|---|---|---|---|---|
| **Synthe-sisET** | Monthly | 1km | 1982-2019 | Synthetization of different ET products based on ranking of validation metrics. | MOD16 ET, PML ET, SSEBop ET, GLEAM ET, GLDAS ET, etc. | - | ground observation and $ET_{wb}$ | Elnashar et al., 2021 |
| **MOD16 -STM** | Monthly | 1km | 1982-2018 | Enhanced MOD16 algorithm by redefining the transpiration and soil evaporation module. MODIS yearly constant land cover is used to extract water cover. | Regional CMFD meteorological data, ERA5-Land LST, GLASS albedo and emissivity, AVHRR NDVI, GLEAM soil moisture. | Ec, Es, Ei | ground observation and $ET_{wb}$ | Yuan et al., 2024; Yuan et al., 2021 |
| **PMLV2 -Tibet** | 8-day | 5km | 1982-2016 | Penman-Monteith-Leuning V2 model calibrated in the Tibet Plateau | Regional CMFD meteorological data, ERA5-Land LST, GLASS albedo, GLASS and GIMSS LAI. | - | $ET_{wb}$ | Ma et al., 2022 |
| **EB** | Daily | 0.05º | 2000-2017 | Improved Surface Energy Balance method based monthly LST | ERA-Interim meteorological data, GLASS (LAI, FVC, albedo), MODIS (land cover, LST). | - | $ET_{wb}$ | Chen et al., 2021 |
| **BESSv2** | Monthly | 5km | 1982-2019 | Quadratic form of the Penman-Monteith equation to estimate ET uses various satellite remote-sensing as input | ERA5 meteorological data, GLASS (LAI, albedo), MODIS (land cover, cloud, aerosol, LAI, etc.). | Ec, Es, Ei | $ET_{wb}$ | Li et al., 2023 |
| **FLUXC OM-RS** | 8-day | 0.0833º | 2001-2015 | FLUXNET and ensemble multiple machine learning | Multiple meteorological data, MODIS (land cover, LST, fPAR, NDVI, EVI, NDWI). | - | $ET_{wb}$ | Jung et al., 2019 |
| **FLUXC OM-RS-ME-TEO** | 8-day | 0.5º | 2001-2013 | FLUXNET and ensemble multiple machine learning | Multiple meteorological data | - | $ET_{wb}$ | Jung et al., 2019 |
| **GLEA Mv3.5a** | Daily | 0.25º | 1980-2018 | Priestley-Taylor equation and data assimilation of soil moisture | ERA5 meteorological data, ESA-CCI soil moisture, NSWEP precipitation, GLOBSNOW SWE,etc. | Ec, Es, Ei, Ew, Ess | $ET_{wb}$ | Martens et al., 2017 |
| **GLEA Mv3.5b** | Daily | 0.25º | 2003-2018 | Priestley-Taylor equation and data assimilation of soil moisture | CERES radiation, AIRS temperature, NSWEP precipitation, GLOBSNOW SWE. | Ec, Es, Ei, Ew, Ess | $ET_{wb}$ | Martens et al., 2017 |
| **CR** | Monthly | 0.25º | 2000-2022 | calibration-free complementary relationship model | ERA5 meteorological data, ERA5-Land LST, GLASS albedo, CERES net radiation. | - | $ET_{wb}$ | Ma et al., 2021 |
| **GLDAS -CLSM** | Daily | 0.25º | 2003- present | Global Land Data Assimilation System,Catchment Land Surface Model | GLDAS-v2.2 forcing data from ECWMF and Princeton, GRACE TWS data. | Ec, Es, Ei, Ess | $ET_{wb}$ | B. Li et al., 2019 |

| | | | | | | | | |
|---|---|---|---|---|---|---|---|---|
| | | | | (GLDAS_CLSM025_DA1_D.2.2) | | | | |
| **GLDAS-Noah** | Monthly | 0.25º | 2000-present | Global Land Data Assimilation System Version 2,Noah Land Surface Model (GLDAS_NOAH025_3H.2.1) | GLDAS-v2.1 forcing data, combination of GDAS, dis-aggregated daily GPCP precipitation, and AFWA radiation datasets. | Ec, Es, Ei | $ET_{wb}$ | Rodell et al., 2004 |
| **GLDAS-VIC** | Monthly | 1º | 2000-present | Global Land Data Assimilation System Version 2.1,Noah Land Surface Model (GLDAS_VIC10_3M.2.1) | GLDAS-v2.1 forcing data, combination of GDAS, dis-aggregated daily GPCP precipitation, and AFWA radiation datasets. | Ec, Es, Ei | $ET_{wb}$ | Rodell et al., 2004 |
| **Terra-Climate** | Monthly | 0.25º | 1958-2020 | modified Thornthwaite-Mather climatic water-balance model | Meteorological data from WorldClim, CRU, JRA, etc. | - | $ET_{wb}$ | Abatzoglou et al., 2018 |
| **MERRA2** | Monthly | 0.25º | 1979-present | The Modern-Era Retrospective analysis for Research and Applications, Version 2, by NASA Global Modeling and Assimilation Office (GMAO) using the Goddard Earth Observing System Model (GEOS) | MERRA-2 global atmospheric reanalysis data | Ec, Es, Ei, Ew, Ess | $ET_{wb}$ | Gelaro et al., 2017 |
| **ERA5** | Monthly | 0.25º | 1979-present | The fifth generation of European ReAnalysis of ECMWF based on Hydrology Tiled ECMWF Scheme for Surface Exchanges over Land (HTESSEL). | ECMWF ERA5 global climate reanalysis data. | - | $ET_{wb}$ | Hersbach et al., 2020 |
| **ERA5-Land** | Monthly | 0.25º | 1979-present | New land component of the fifth generation of European ReAnalysis of ECMWF: Carbon Hydrology-Tiled ECMWF Scheme for Surface Exchanges over Land (CHTESSEL). | Downscaled meteorological forcing from the ERA5 climate reanalysis | - | $ET_{wb}$ | Muñoz-Sabater et al., 2021 |

### 2.2.4 Other data

The precipitation data used in this study are from the TPHiPr dataset, which is long-term high-resolution (1/3°, daily) precipitation datasets for the TP obtained by merging the atmospheric model output with gauge observations from more than 9000 rain gauges around the TP (Jiang et al., 2023). Compared to other widely used precipitation datasets, this dataset has remarkably better accuracy in the TP, with a generally unbiased and root mean square error of 5.0 mm/d.

## 2.3 Methodology

### 2.3.1 Evaluation of ET products

We evaluated different ET data products (Table 2) at monthly scale against ground observations and basin-scale estimates of the water balance. Various error metrics were calculated to assess the accuracy of these ET datasets. These ET datasets were mainstream gridded ET products obtained by a variety of algorithms applied to the TP or globally. Considering that the footprint of the *in-situ* flux tower observations was generally in the range of several hundred meters to kilometers, they were used to evaluate ET datasets at relatively high resolution (≤1km), including six global ET products and one regional ET product, i.e., ETMonitor, MOD16, PMLV2, SSEBop, GLASS, SynthesisET, and MOD16-STM. All products were evaluated against estimates of the basin-scale water balance, regardless of their spatial resolution. When validating with ground observations, the ET values of the ET products in the pixels where the flux sites are located were extracted directly for comparison. For comparison with the basin-scale water balance data, the basin-scale monthly averaged ET values of different products were calculated using the area-weighted averaging method according to the basin boundary.

The following commonly used accuracy metrics were applied, including the correlation coefficient (R), the bias (BIAS), the root mean square error (RMSE), and the Kling-Gupta efficiency (KGE) (Gupta et al., 2009). The KGE is a multi-objective statistical indicator that incorporates the correlation, relative variability ratio and mean value ratio, to comprehensively evaluate the accuracy. The metrics were calculated as:

$$R = \frac{\sum_{i=1}^{n}(ET_e - \overline{ET_e})(ET_o - \overline{ET_o})}{\sqrt{\sum_{i=1}^{n}(ET_e - \overline{ET_e})^2}\sqrt{\sum_{i=1}^{n}(ET_o - \overline{ET_o})^2}} \tag{1}$$

$$BIAS = \sum_{i=1}^{n}(ET_e - ET_o)/n \tag{2}$$

$$RMSE = \sqrt{\sum_{i=1}^{n}(ET_e - ET_o)^2} \tag{3}$$

$$KGE = 1 - \sqrt{(R-1)^2 + (\frac{\mu(ET_e)}{\mu(ET_o)} - 1)^2 + (\frac{\sigma(ET_e)}{\sigma(ET_o)} - 1)^2} \tag{4}$$

where $ET_e$ (mm/month) indicates the ET values of different products, $ET_o$ (mm/month) indicates the ground-truth ET values, either from *in-situ* observations or basin-scale water balance estimates, $\mu$ is the mean value, $\sigma$ is the standard deviation, and R is the Pearson correlation coefficient between the ET values and the ground-truth ET values. KGE is smaller than 1, and higher KGE means better agreement between observations and estimates.

### 2.3.2 Inter-comparison of different products

In order to inter-compare the spatial variation of ET by different products, multiple-year average annual ET was calculated and analysed for each product during their overlap period from 2003 to 2013, unless specific period was redefined. The averaged and median values of ET, as well as the standard deviation, of different products were calculated at both pixel-wise and

basin-wise level, to explore the discrepancy of ET magnitude by different products. The ratio of standard deviation to multi-products average ET values was used as an indicator of uncertainty. For products that also provide the ET components (i.e., plant transpiration, soil evaporation, canopy rainfall interception evaporation, open water evaporation, and snow/ice sublimation), the individual contributions of these components to the total ET were also calculated and compared.

Monthly ET values were produced for each product to analyse the seasonal variation in ET. It is generally agreed that long-term temporal coverage (i.e., at least 30 years) is required to estimate the trend of climate variables. However, most ET products cover a relatively short period. Although the relatively short period of time can be debated, these time series are helpful to clarify the trend in recent years and to understand the difference in trend among products. The calculation of the trends can be affected by exceptional years (outliers) with extremely high or low ET. To reduce the influence of these outliers, we used the robust regression method instead of the simple linear regression method. The significance level of the trend was estimated using a *t*-test.

## 3. Results

### 3.1 Evaluation of ET products

#### 3.1.1 Validation of ET products against flux tower measurements

Figure 2 and Supplementary Figure S2 show the validation results. It should be noted that all the products have different temporal coverage and the eddy covariance observations at the flux tower sites also cover different years. In addition, some ET products do not have valid values over certain land cover types, e.g., the MOD16 ET algorithm does not work over non-vegetated areas, so MOD16 ET has no data at the QOMS and NADORS sites (both have land cover as desert steppe). Therefore, the accuracy metrics for each ET product in Figure 2 have only be calculated for those periods when both ground measurements and the ET product data are available at each site. To provide a fair and overall comparison, Figure 2 also shows the metrics for the condition only when all products and ground data are overlapped ('*Overlap*') and the overall metrics that include all conditions ('*All*'). More information on the validation period and relevant information can be found in Supplementary Table S1.

Among all the global ET datasets, ETMonitor and PMLV2 ET achieved the highest accuracy with the highest KGE (>0.77) and lowest RMSE (<20 mm/month). As expected, the regional ET product MOD16-STM showed good performance with low RMSE and high KGE (15.84 mm/month and 0.77 when using the '*overlap*' validation samples). This can be attributed to the fact that MOD16-STM ET was calibrated using the flux observations from the TP sites and was estimated based on the regional bias-corrected climate data with better accuracy than the global forcing data. These three PM -type model-based products (ETMonitor, PMLV2, MOD16-STM) showed overall better accuracy than other products. The energy balance-based SSEBop

ET product had the largest negative bias and lowest KGE for relatively wet sites and desert sites, but showed good accuracy for some alpine steppe sites with sparse vegetation cover (e.g., SH, YK, NAMORS). Figure 2 also indicates that the ensemble ET datasets (GLASS and SynthesisET) showed poorer accuracy than other ET products, e.g., both with KGE less than 0.6 and negative bias (-13.76 ~ -10.82mm/month), which is most likely related to the ensembled data sources and algorithms. Most products showed better accuracy at the relatively wet sites with dense vegetation cover (e.g., GT, HBG, ARS, CN-Ha2 sites), as judged by relatively higher values of KGE and R, than that at the relatively dry sites with sparse vegetation cover or desert (e.g., QOMS, NADORS).

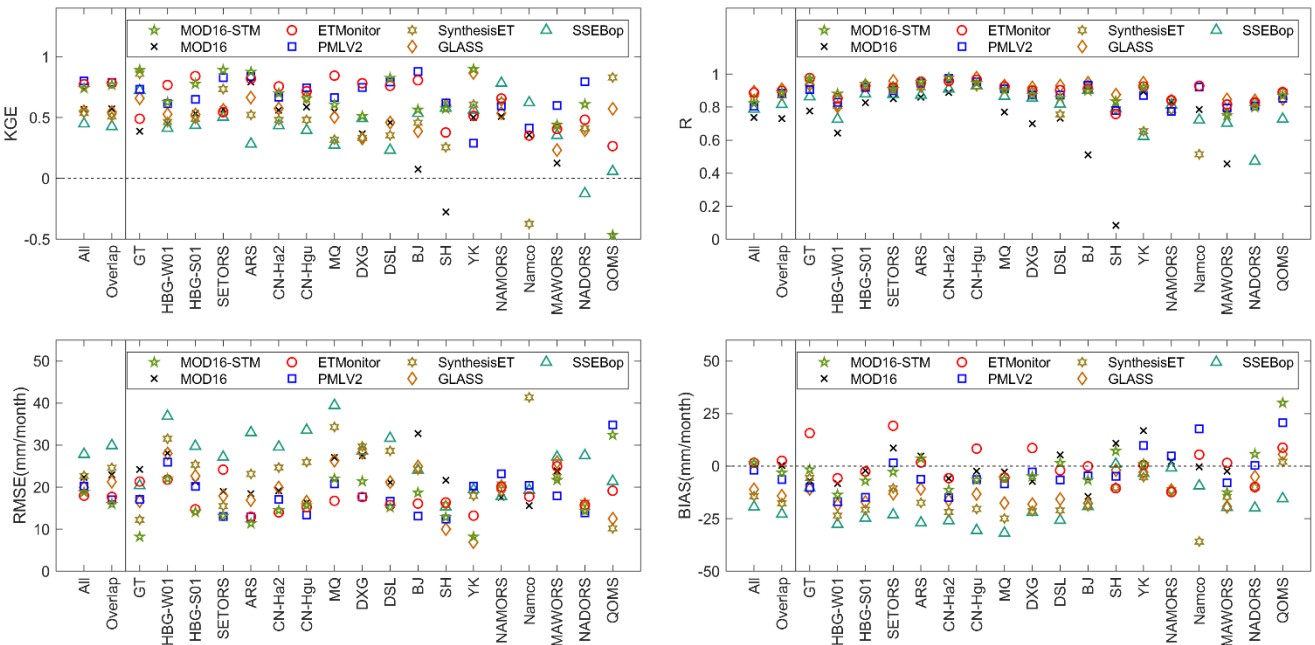

**Figure 2: Summary of the validation results of high-resolution ET products against flux tower measurements. '*All*' indicates that the validation results are obtained based on different samples depending on the availability of each product, while '*Overlap*' represents the validation results are obtained based on same sample numbers (mainly vegetation covered sites) for every product.**

**3.1.2 Evaluation of ET products against water balance model estimates**

Figure 3 and Supplementary Figure S3 showed the comparison of all ET products with the basin-scale water balance ET ($ET_{wb}$). As expected, the regional ET products MOD16-STM and PMLV2-Tibet showed good agreement with the water balance-based ET of the five river basins described in Section 2.2.2, with KGE of 0.64~0.87 and RMSE of 12.19~15.60 mm/month. Although both MOD16-STM and PMLV2-Tibet were calibrated using the ground flux observations from the TP, their accuracy is different, with the MOD16-STM ET showing a slightly lower KGE, most likely due to its underestimation at high ET levels (Figure 3 and Supplementary Figure S3). ETMonitor and PMLV2 ET also had high KGE ($\geq 0.80$) and low RMSE (<14mm/month). SynthesisET had the highest RMSE and BIAS, this is due to the fact that SynthesisET ensembles different

data sources in different time periods, resulting in inconsistent time series. Among the coarse resolution reanalysis and LSM ET products, TerraClimate, ERA5, and ERA5-Land showed overall good accuracy with KGE ≥ 0.78 and RMSE ≈ 13

260 mm/month, while GLDAS-CLSM and GLDAS-VIC showed large errors with RMSE ≥ 20 mm/month and KGE ≤ 0.41. CR also showed overall good accuracy in the TP, but had relatively lower KGE in arid basins (e.g., InnerTP), where GLEAM and SSEBop showed relatively higher KGE. Of all products, the PMLV2-Tibet and ETMonitor ET products showed the lowest RMSE (<13mm/month) and the highest KGE (0.87) and R (>0.90) when compared with $ET_{wb}$. The global ET products ETMonitor, PMLV2, GLEAM35a, GLEAM35b, TerraClimate, ERA5, and ERA5Land showed above-average accuracy due to their

lower RMSE and higher KGE. When regressed against $ET_{wb}$, most ET products showed slope values less than one, indicating these ET products underestimate ET in regions or periods with high ET values (Supplementary Figure S3). Among them, ETMonitor, CR, and TerraClimate ET showed slope values close to 1 (larger than 0.9), which highlights their good accuracy in the reference basins.

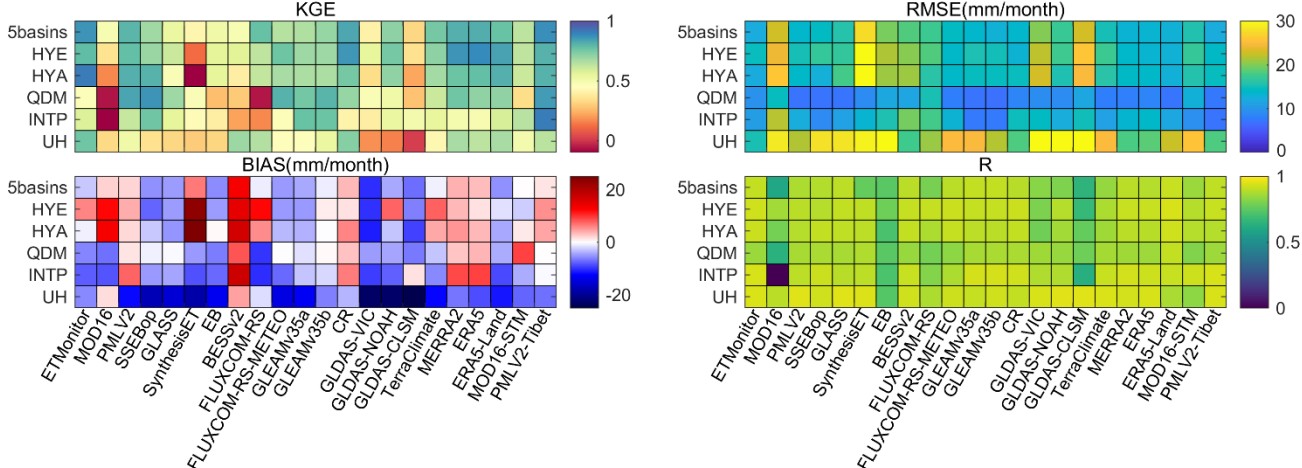

**Figure 3: Summary of the evaluation of ET products against basin-scale ET based on water balance estimates for the headwaters of Yellow basin (HYE), headwaters of Yangtze basin (HYA), upper Heihe basin (UH), Inner TP (INTP) and Qaidam (QDM). '5basins' presents the validation results when data from all five basins were used together.**

## 3.2 Variability of ET across the TP

### 3.2.1 Spatial variability in ET across the TP

Figure 4 shows the mean value of the multi-year average annual ET from the 22 ET products and the standard deviation across the TP, and Supplementary Figure S4 documents the spatial variability of multi-year average annual ET across the TP by each product. The annual ET in the river basins over the TP by different products is summarized in Supplementary Table S2. In general, most of the products showed pixel-wise ET values below 800 mm/yr and showed a similar spatial pattern, with high ET values in the eastern part and low ET in the western part of the TP. The regional ET histogram showed two peaks for some

datasets, e.g., ETMonitor, EB, MERRA2. These peaks correspond to the low ET values of non-vegetated or sparsely vegetated

regions in the central and western TP and the high ET values of regions in the eastern TP with dense vegetation and relatively humid climate. The spatial variability, expressed by the standard deviation of different products (Figure 4), suggest large differences between different products in the central to western TP, where most ET products show low ET values, e.g., ET values from ETMonitor, SSEBop, EB, are generally less than 200 mm/yr, while some ET products show much higher ET values, e.g., ET values from BESSv2, ERA5, and ERA5-Land reach 400 mm/yr, illustrating their overestimation in the arid regions/basins (Supplementary Table S2).

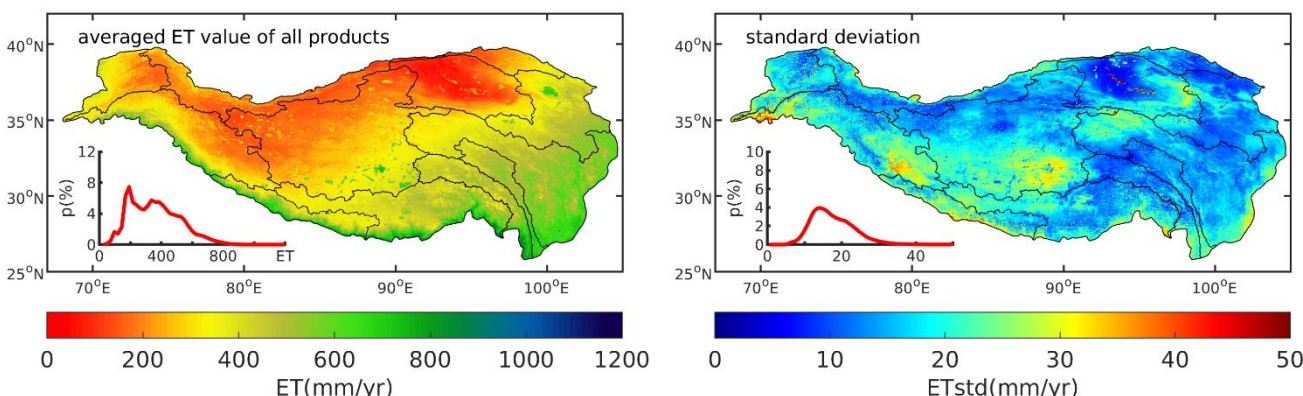

**Figure 4: Mean multi-year average annual ET from the 22 products and the standard deviation across the TP during their overlap period (2003~2013). The inset in each panel shows the histogram.**

Figure 5 summarizes the multiple-year average ET over different products in the TP. Among all the ET products, BESSv2 ET presents highest multiple-year average ET value in the TP, while GLDAS-VIC shows the lowest ET values (Figure 5). The basins with low ET and sparse vegetation cover (e.g., Qaidam, Inner TP, Hexi Corridor, Tarim, and Amu Darya) have the largest uncertainty between products, expressed as the ratio of standard deviation to the mean values (Supplementary Table S2). Uncertainty is also high in the Indus and Brahmaputra basins, most likely due to their complex topography, extreme altitude, and large areas of permanent glaciers and snow, which make it difficult to obtain reliable estimates. According to the above-mentioned evaluation results, five ET products (ETMonitor, PMLV2, GLEAMv3.5a, GLEAMv3.5b, TerraClimate) were found to have continuous spatial coverage and provide reliable estimates; the median and average annual ET from these five products in the TP are 339.9 mm/yr and 333.1 mm/yr, respectively, with a standard deviation of 38.3 mm/yr. Based on the TPHiPr precipitation data (Jiang et al., 2023), the total annual precipitation in the TP is 631 mm/yr, so ET accounts for about 52($\pm$7)% of the total annual precipitation. The difference among these products is also noticeable at the basin scale.

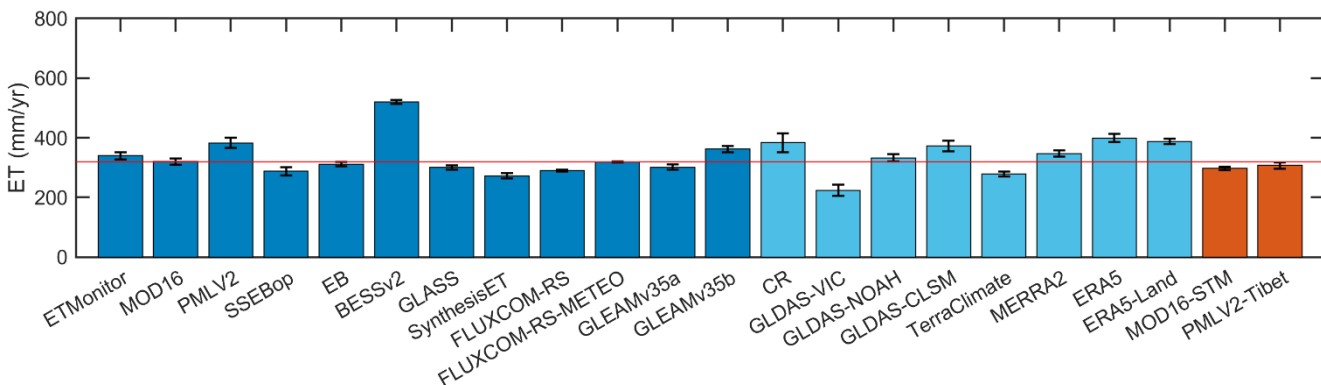

**Figure 5: Bar plot of the multi-year (2003~2013) averaged ET of different products in the TP. The red horizontal line represents the average ET of all products. The global ET datasets based on satellite remote sensing are in dark blue, the global ET datasets based on land surface models and analysis global ET dataset are in light blue, the regional ET datasets are in red. It should be noted that some products only provide ET values for the vegetation-covered regions, e.g., MOD16, FLUXCOM-RS and FLUXCOM-RS-Meteo, and there are two products (MOD16-STM, PMLV2-Tibet) cannot cover the regions outside of China.**

### 3.2.2 Temporal variability in ET across the TP

Figure 6 shows the monthly variation of ET across the TP, while Supplementary Figure S5 illustrates the differences between different products. Despite the diverse temporal profiles observed, most products indicate that the highest ET occurs in July and August. ET during the monsoon (June to September) and pre-monsoon seasons (March to May) accounts for 62% (±7%) and 23% (±4%) of the annual total ET, respectively. The remaining 15% of ET occurs during October to next February. In summary, 66% and 22% of the annual precipitation occurs during the monsoon and pre-monsoon seasons respectively, with the remaining 12% occurring during the rest of the year. The monthly patterns of ET variability are similar in all basins, with differences in magnitude. The proportion of ET during the monsoon season is higher in the dry basins, e.g., 69% in Hexi Corridor and 68 % in Qaidam, compared to the wet basins, e.g., 53% in the Ganges and 58% in the Brahmaputra.

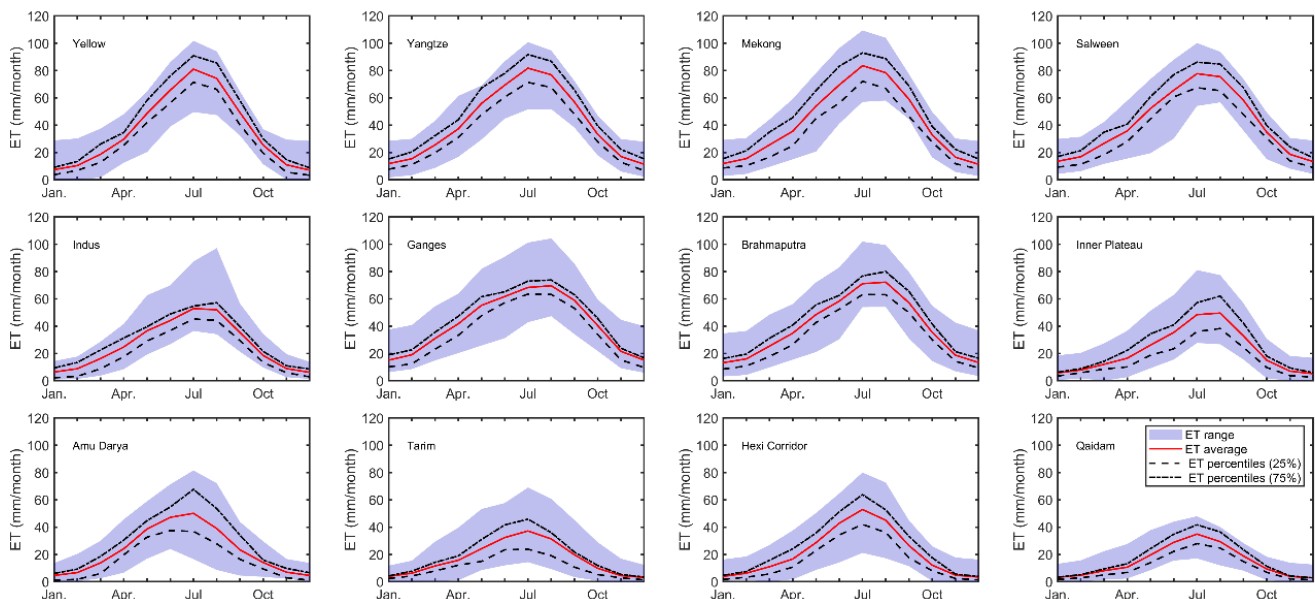

**Figure 6: Seasonal evolution of ET in different basins in the TP. Data shown are monthly averaged ET values during 2003~2013.**

Figure 7 shows the time series of annual ET spatially averaged over the TP for different products. Large deviations were
observed among the products, with BESSv2 having the highest value of spatial-average annual ET and the GLDAS-VIC
having the lowest. The trend of annual ET varies with different products and their temporal coverage (Figure 7 and Supple-
mentary Figure S6). The results suggest a general, significant, increasing trend of ET since the 1980s (most products with
$p<0.05$). Since 2000, the annual ET has shown both positive and negative trends depending on the product. Most products
showed a significant increasing trend ($p<0.05$), and the median ET of all products increased at a rate of 1.70 mm/yr from 2000
to 2020 in TP ($p<0.05$). The SynthesisET showed the largest significant negative trend. At the basin scale, the difference in
annual trends between different products is also clearly illustrated (Supplementary Figure S6). Most basins showed significant
increasing trend of ET, especially the Yellow, Yangtze, Mekong, Tarim, Hexi Corridor, Tarim, and Qaidam basins, where
most products had a positive ET trend. The median ET trend is either negative or close to zero in the Ganges, Brahmaputra,
Amu Darya, and Inner TP basins, probably indicating a decreasing or non-monotonic trend for these basins.

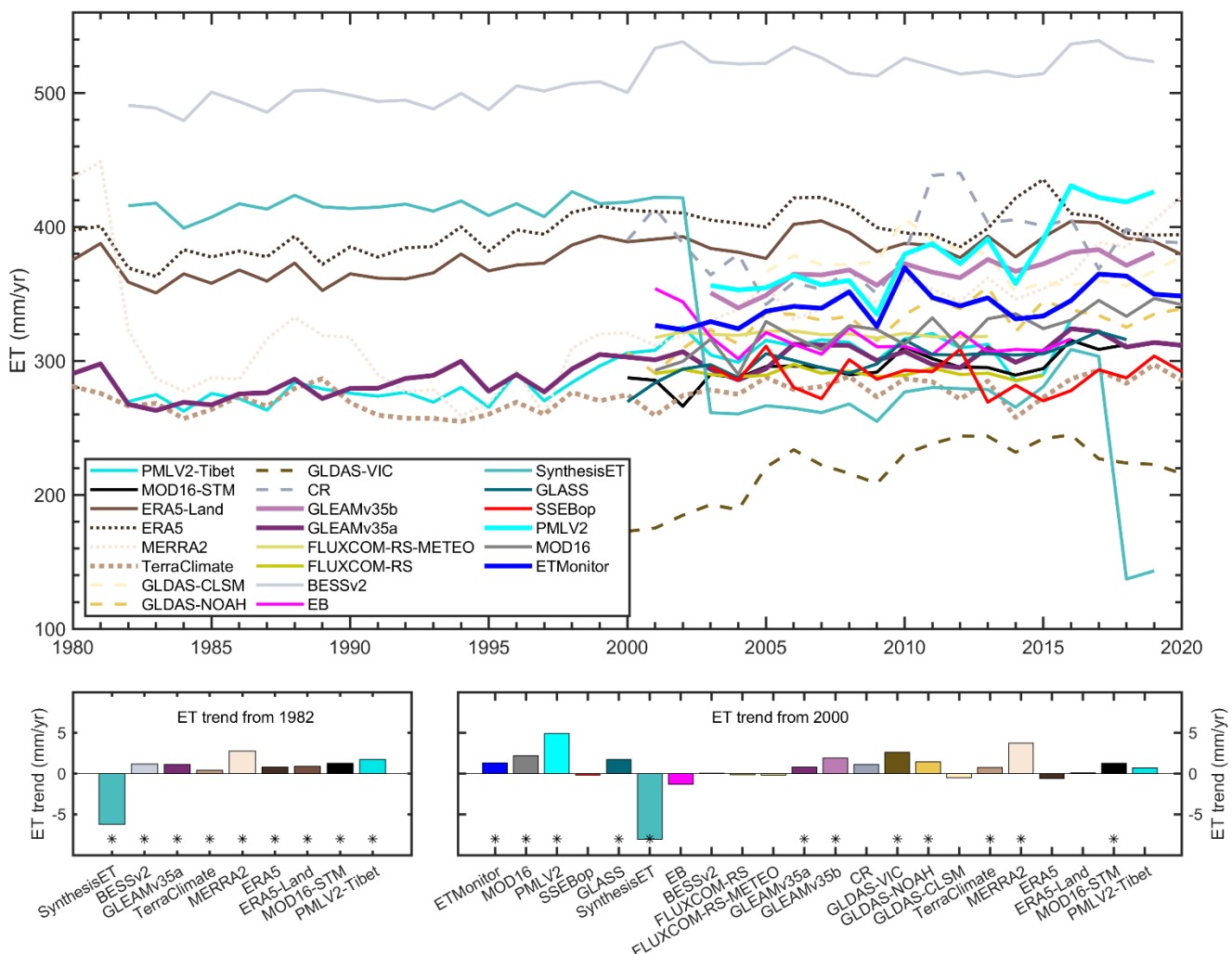

**Figure 7: Time series of annual ET by different products in the TP. The inset panel shows the annual ET trend by different products. \*: trend with significance level (*p<0.05*). In the upper panel, the reanalysis data are shown by a dotted line, and the land surface model-based data are shown by a dashed line.**

## 3.3 ET components

We also compared the main ET components, i.e., Ec, Es, and Ei from nine products, including ETMonitor, PMLV2, MOD16-STM, GLEAMv35a, GLEAMv35b, GLDAS-VIC, GLDAS-NOAH, GLDAS-CLSM, and MERRA2. It is important to note that there is no independent reference data available to validate the ET components, and each model has a different way of estimating these components. Even when the total ET is consistent across different products, the individual components can differ significantly (Figure 8 and Supplementary Figure S7). All products show higher Ec and Ei values in the eastern TP and lower values in the central and western TP (Supplementary Figure S7). This pattern follows the spatial distribution of environmental factors (e.g., LAI and precipitation), i.e., regions with high ET values are mostly covered by forest and alpine

meadow with higher precipitation, whereas regions with low ET values are dominated by sparse vegetation (alpine steppe and desert steppe) with lower precipitation. Large deviations in Es values were observed, with several products showing high Es

values in the eastern TP, e.g., MERRA2, GLDAS-CLSM, ETMonitor, and MOD16-STM, while some products showed extremely low Es values, e.g., GLEAMv35a, GLEAMv35b, and GLDAS-VIC.

Figure 8 shows the false color composite maps of the relative magnitude of transpiration (Ec), soil evaporation (Es), and interception (Ei) from different products, with Red (Es is largest), Green (Ei is largest), Blue (Ec is largest). In the false color composite maps, the red (green, blue) color means that Es (Ei, Ec) contributes most to total ET. There are clear differences

between different products. Most products generally indicate that Es is the major contributor to total ET (Figure 8, pie diagram). In contrast, three products (GLEAMv35a, GLEAMv35b, and GLDAS-VIC) show that plant transpiration is the main contributor to total ET (Figure 8, pie diagram), most likely due to the extremely low Es values in the eastern TP (Supplementary Figure S7). The averaged Es/ET values range from 18% in GLDAS-VIC to 84% in MOD16-STM, with a median value of 50%. Averaged Ec/ET values range from 11% in GLDAS-CLSM to 58% in GLEAMv35a, with a median value of 30%. Most

products generally showed low Ei/ET values with a median value of 5%, while GLDAS-VIC and GLDAS-NOAH show the highest Ei/ET values (20% ~ 36%). Overall, the ET partitioning ratio in ETMonitor is the closest one to the median value of all products.

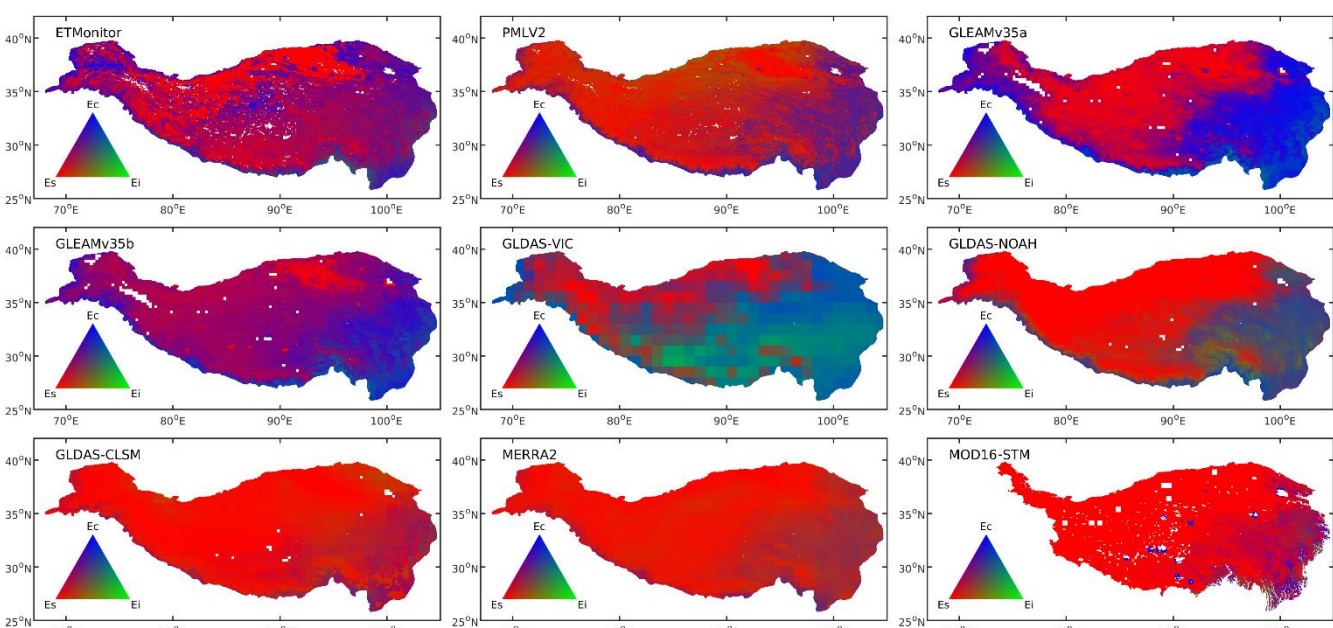

**Figure 8: False color composite maps to visualize the relative magnitudes of the transpiration (Ec), soil evaporation (Es), and inter-**
360 **ception (Ei) contribution to total ET according to different products.**

In addition, two other components of water vapour flux are considered separately: evaporation from open water bodies (Ew) and sublimation from snow/ice-covered surfaces (Ess). There are three products providing open water evaporation, including ETMonitor, GLEAMv35a and GLEAMv35b (Figure 9, upper panel), and five products providing snow/ice sublimation,

including ETMonitor, GLEAMv35a, GLEAMv35b, GLDAS-CLSM, and MERRA2 (Figure 9, middle and lower panels). For open water evaporation, three products provide comparable Ew results with average Ew/ET from 3.45% to 4.10%. According to Wang et al. (2020), the total water evaporation is about $29.4 \pm 1.2$ km$^3$/yr ($\approx$1111.5 mm/yr) from the 75 lakes in the TP with total area of 26,450 km$^2$ (accounting for approximately 56.9% of the total lake area in the whole TP), and the total lake evaporation ($51.7 \pm 2.1$ km$^3$/yr) for all plateau lakes. The total open water evaporation amount from ETMonitor gives a value of 945.3mm/yr for the permanent water surface over the TP. The total water area is $1.29 \times 10^6$ km$^2$ in the TP when seasonal water bodies are taken into account, which is much larger than the permanent water surface. ETMonitor takes into account the seasonality of water surface areas when estimate ET, and the multi-year mean total annual water evaporation in the TP estimated by the ETMonitor is about at 44.4 km$^3$/yr, which is lower than that given by Wang et al. (2020). For snow/ice cover surface, GLDAS-CLSM provided the overall highest ratio of sublimation (Ess) to total ET (i.e. Ess/ET) with a regional mean of 7.79%, and GLEAMv35a provided the overall lowest Ess/ET value with a regional mean of 1.20%. This difference is mainly caused by large differences in Ess between different products in the southern TP, e.g., in the Indus, Ganges and Brahmaputra watersheds, where Ess is not well captured by GLEAM. The sublimation (Ess) estimated by the ETMonitor falls in the middle of these ET products, with a regional average of 4.3%.

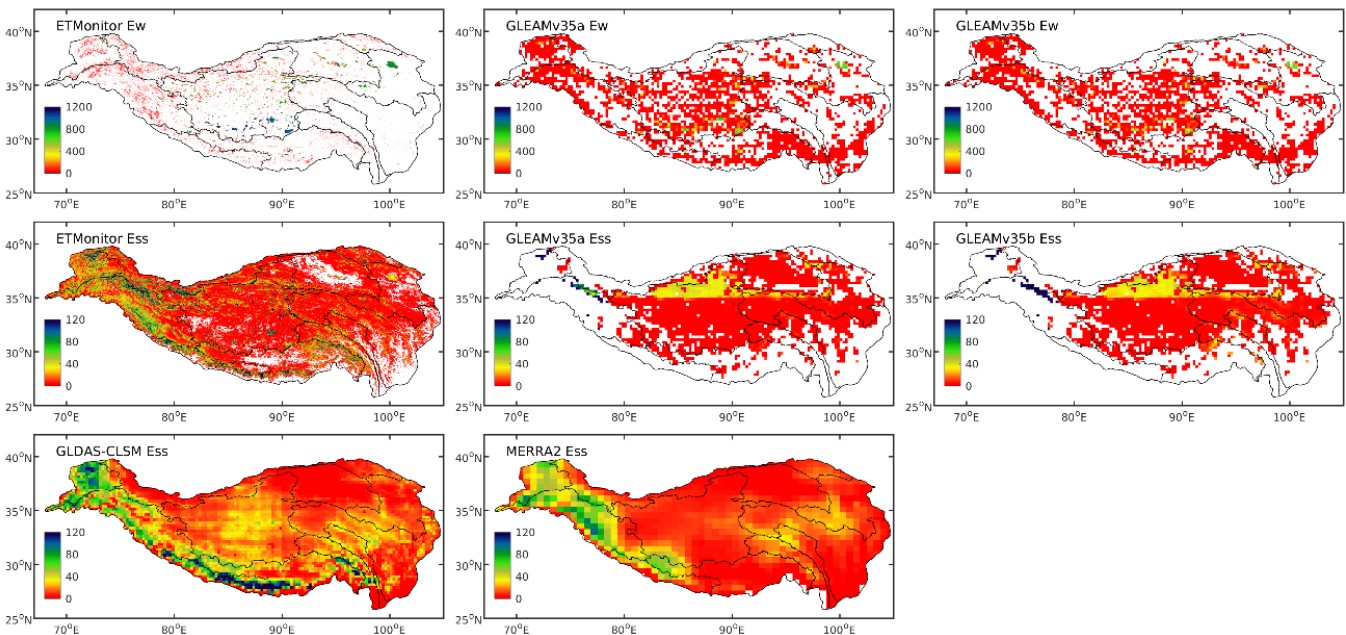

**Figure 9: Spatial variability of open water evaporation (Ew) and snow/ice sublimation (Ess) in the TP by different products.**

## 4. Discussion

To understand regional and global climate change, as well as regional ecohydrological processes in the TP, knowledge of the changes in ET over time and space is required. In this study, 22 ET products were evaluated using various methods, i.e.,

comparing ET products with ground eddy covariance observations and basin-scale water balance estimates, assessing the spatiotemporal variability of ET and its components, to assess the performance of ET products and clarify the ET amount, spatiotemporal variability and trends of ET in the TP. After this comprehensive evaluation and analysis, we have gained a clear understanding of the water vapour released from the TP. In addition, we find that the evaluation results are highly relevant to better understand the performance of ET models and the underlying vaporization processes, thus providing suggestions for further improvements in the ET estimation for the TP.

## 4.1 Relevance of evaluation results towards a better understanding of the vaporization processes

The *in-situ* observations with an eddy covariance system are recognized as the standard method for monitoring energy and mass fluxes to validate high-resolution ET (Baldocchi, 2020). In addition, the ET products were compared with the basin-scale water balance estimates $ET_{wb}$. $ET_{wb}$ is obtained at the basin scale (several hundred km$^2$), which is much larger than the footprint of flux tower observations (approximately km$^2$, depending on meteorological conditions). Given the relatively sparse distribution and small footprint of the flux-tower-based eddy covariance system observations, the water balance method can serve as a useful complementary reference for ET estimates. This is especially true for the coarse-resolution ET, which has a much larger spatial footprints than eddy covariance observations.

In this study, these two methods gave generally consistent results when evaluating the high-resolution ET. When judged by the KGE of site-scale estimates, the accuracy of the high-resolution ET products can be ranked as follows: PMLV2 > ETMonitor > MOD16-STM > GLASS > MOD16 > SynthesisET > SSEBop. When judged by the KGE of basin-scale validation, the accuracy of the high-resolution ET products can be ranked as: ETMonitor > PMLV2 > MOD16STM > SSEBop > GLASS > MOD16 > SynthesisET. Although both indicate that ETMonitor, PMLV2, and MOD16STM are the most accurate and the remaining four are less accurate among the high-resolution ET products, some differences in the ranking of the ET products can be observed. This is probably related to the processes captured by the 'ground-truth' data at different scale used in the two evaluation methods. An eddy covariance observation represents the net water vapour flux integrated across different processes at given point (e.g., plant transpiration in the dense vegetation regions, snow sublimation in dry snow cover regions, evaporation of canopy-intercepted water when the canopy is wet due to intercepted rainfall). In addition, the observed vaporization process depends on the land surface conditions at the observation sites during particular times, which may vary seasonally and annually due to factors such as snow/ice, intercepted water, and vegetation. The estimated basin-scale ET by water balance ($ET_{wb}$) was essentially the residual of the observed water balance terms, which is assumed to be the net liquid water flux loss to the atmosphere at the basin scale. Compared to the site-scale observation, the basin-scale $ET_{wb}$ can capture the effect of land cover dynamic on the ET within the basin. For example, the mean water level of lakes in TP increased by 0.20 m/yr from 2000 to 2009, and the lake water mass increased significantly (Zhang et al., 2013), which caused higher ET in the TP because water evaporation is generally higher than other land cover types. However, most ET products (e.g., MOD16, PMLV2, etc) assume constant land surface conditions throughout the year or multiple years, which means that they cannot capture the temporal

transitions of the vaporization process associated with changes in land cover. In contrast, ETMonitor adjusts the daily land cover based on dynamic land cover conditions, including water bodies cover and snow/ice cover, which allows it to reflect the impact of seasonal and annual open water extent and snow/ice cover on total ET (Zheng et al., 2022). This probably explains in part why ETMonitor performs slightly better than PMLV2 when validated by basin-scale water balance methods, while they are comparable when validated by *in-situ* observations.

Eddy covariance observations capture condensation when negative latent heat flux (i.e. ET) occurs. Remote sensing-based ET products mainly focus on positive ET (positive upward latent heat flux) and omit processes such as condensation. For example, in the MOD16 ET product algorithm, net radiation (Rn) is constrained to positive values (Rn is set to 0 if Rn<0) indicating that negative ET is not allowed. Negative ET (e.g., condensation) can also occur when VPD is negative. Depending on whether negative Rn or negative VPD is allowed, the considered water phase changes are different and this will certainly affect the accuracy of the ET products.

The evaluation using the basin water balance method gave slightly higher metrics compared to the flux tower results. This may be attributed to the disparity in spatial resolution between the flux tower measurements and the basin-scale $ET_{wb}$ estimates. Basin-scale $ET_{wb}$ may offset the positive and negative biases within the basin, resulting in better evaluation metrics (Liu et al., 2023). However, the uncertainties in the water balance method as ground-truth data should also be acknowledged. This method is based on the validity of several assumptions (e.g., negligible subsurface leakage to adjacent basins) and the reliability of data on precipitation, runoff, and water storage (Mao et al., 2016). In cold regions such as the TP, where glaciers and snow have a substantial impact on the water balance, meltwater should also be considered (Wang et al., 2022).

### 4.2 Implications for the estimation of ET in the TP

### 4.2.1 ET estimation using PM-type models

This study found that ET products generated using PM-type models were more accurate than other models. In particular, ETMonitor and PMLV2 were the most accurate when evaluated by both *in-situ* flux observations and estimates of ET based on the basin-scale water balance. This is consistent with the conventional wisdom that energy balance-based ET models are suitable for water-limited conditions in bare and partially vegetated areas, while PM-type ET models are more effective for both energy-limited and water-limited conditions in vegetated areas (Chen and Liu, 2020). An exception was found for MOD16, which had below-average accuracy overall, but its regionally improved version (MOD16-STM) gave significantly more accurate estimates of ET after regional parameter calibration and improvement of the soil evaporation module (Yuan et al., 2021). The reason for this is that MOD16 is only applicable to limited areas and seasons of the TP due to its unfavourable parameterization, which does not account for conditions in the central to western TP due to the lack of estimation of bare soil and open water evaporation.

This study also highlights the potential for improving model parameters to estimate ET using PM-based models, e.g., by incorporating soil moisture to compute a water stress indicator, by integrating the water balance simulation and data assimilation, or by coupling the water and carbon cycles to estimate ET. We found that PM-type models incorporating soil moisture to parameterize water stress gave very good results. For instance, to improve the accuracy of ET estimate, ETMonitor utilized high-resolution soil moisture data to refine the parameterizations of soil and canopy surface resistances to estimate soil evap-

oration and plant transpiration (Hu and Jia, 2015; Zheng et al., 2022). GLEAM assimilates surface soil moisture to estimate water availability in the root zone, and applies it to determine the water stress (Miralles et al., 2011), which also gave accurate estimates of ET in the TP. Coupling the water and carbon cycles can also be helpful for better estimates of ET, e.g., PMLV2 adopted water-carbon cycle coupling to estimate ET (Zhang et al., 2019), since canopy conductance controls both transpiration and photosynthesis. The regional adaptation of parameterizations and better forcing are also beneficial, as shown in this study,

where MOD16-STM and PMLV2-Tibet products showed better agreement with reference values than MOD16 and PMLV2. Furthermore, PM-type model-based ET products (especially those based on duel-source or multi-source models) can provide different ET components, benefiting from the more realistic representation of biotic and abiotic processes.

### 4.2.2 ET estimation using LST-based model

Although the absolute accuracy of the energy balance-based EB and SSEBop products may be lower than that of the ET

products from the optimized PM-types models, they have some advantages, such as the close coupling of energy balance with sensible heat flux and the good ability to present the spatial variability of ET, especially for the high-resolution dataset. Previous studies pointed out that the LST-based models fail to produce temporally and spatially continuous ET fields under variable cloud conditions. The continuity of LST was significantly improved recently through temporal upscaling technologies , which may further benefit the ET estimation. The relatively good accuracy of SSEBop at some sites (e.g., SH, YK, NAMORS) in

this study also demonstrates the potential of LST-based models to achieve estimate ET accurately in arid and sparsely vegetated regions.

### 4.2.3 Uncertainty propagation in data-driven ensemble ET products

The accuracy of ET products based on data-driven models has been quite variable in the TP. GLASS and SynthesisET are both ensemble ET products, with GLASS employing Bayesian averaging and SynthesisET using a ranking-based method (Yao et

al., 2014; Elnashar et al., 2021). However, these two products showed significant differences, with SynthesisET showing much larger errors (Figure 3 and Figure 4). This finding on SynthesisET differs significantly from a previous study that validated ET product at the global scale (Liu et al., 2023), which claimed that SynthesisET was the best product when applied in its time span based on accuracy indicators (e.g., RMSE) by comparing to *in-situ* observations and water-balance estimates. After screening the time series of SynthesisET, we found significant temporal inconsistencies (much higher ET values before 2000

than after, which is also shown in Figure 7), mainly caused by its synthesis method. SynthesisET ensembled two or three high-

ranking ET datasets at each time step according to the evaluation metrics. The use of different products for different time periods, without correcting for the differences in different products, eliminates the possibility of improving the quality of a data product through an advanced ensemble method or critical selection of inputs (Wang et al., 2021). Therefore, to ensure a more reliable and comprehensive assessment, we propose to analyse the spatial and temporal variability of ET as in Section 480 3.2 of this study.

Data-driven methods, especially machine or deep learning methods, are increasingly applied in the geosciences to extract land surface information (Karpatne et al., 2017). The FLUXCOM product integrates the ET results upscaled from *in-situ* observations using various machine learning models (Jung et al., 2019). The FLUXCOM-RS-METEO product, which is obtained using both meteorological datasets and remote sensing datasets as drivers, is also found to have a good accuracy in TP. However, the FLUXCOM-RS product, which differs from FLUXCOM-RS-METEO, performs poorly in the TP according to the findings of this study, indicating the importance of meteorological variables in estimating ET.

### 4.2.4 LSM and reanalysis ET products

We also compared several ET products from LSM and climate reanalysis, including GLDAS-Noah, GLDAS-VIC, GLDAS-CLSM, CR, TerraClimate, ERA5, ERA5Land, and MERRA2. Although these products generally have low spatial resolution 490 (0.1˚~1.0˚), they have a long temporal coverage, making them more suitable for climate studies. Among them, TerraClimate, CR, ERA5, and ERA5-Land showed overall good accuracy when compared to $ET_{wb}$, while GLDAS products had a relatively low accuracy. The poorer accuracy of GLDAS ET datasets is mainly caused by the forcing data and parameter settings, which need significant improvement when applied to the TP (X. Li et al., 2019). In the central and western regions of the TP, where the surface vegetation cover is sparse and the climate is arid or semi-arid with low precipitation (roughly 300 mm/yr or less), 495 CR, ERA5, and ERA5-Land produce higher ET values than other products. The high ET values of ERA5 and ERA5-Land are most likely due to the overestimation of precipitation in the TP by ERA5 (Jiao et al., 2021; Xie et al., 2022), which leads to the overestimation of both ET and runoff (Sun et al., 2021). Previous studies have reported relatively high ET values from CR methods in the central and western TP (Yang et al., 2020) and Arctic basins (Ma et al., 2021), which can be partly explained by the uncertainty of the forcing (Ma et al., 2021) and by the applicability of CR in cold regions during non-thawing periods 500 (Yang et al., 2021). A basic assumption of CR is that the energy difference between potential ET ($ET_P$) and the ET under wet conditions has a linear or nonlinear relationship with the energy difference between $ET_P$ and actual ET when water is limited. This relationship fails during periods of soil freezing and thawing, when the available energy is mainly used for the phase change of ice-water (with higher latent heat) (Yang et al., 2021). Furthermore, CR also assumes that the changes in land surface properties can be accurately and promptly estimated from changes in atmospheric conditions, neglecting regional or large-505 scale advection, which makes it inapplicable in heterogeneous areas (Morton, 1983; Han and Tian, 2020; Crago et al., 2021).

### 4.2.4 Suggestions for further ET estimations in TP

Several aspects could be addressed to improve the ET estimation in the TP. The current ET models could be improved by integrating different models and processes, such as combination of LST-based models and conductance-based PM-types models (Chen and Liu., 2020) or data-driven algorithms (Zhao et al., 2019; Shang et al., 2023), combining ET processes with carbon cycle and hydrological processes (Zhang et al., 2019; Abatzoglou et al., 2018). The appropriate combination of PM-type models and machine learning algorithms could benefit from both and result in a more powerful model for ET estimation (Koppa et al., 2022). Recent studies have highlighted the improved accuracy of the hybrid model by estimating the canopy conductance using machine learning methods and applying PM-type models (Zhao et al., 2019; Shang et al., 2023), which is direction towards better estimates of ET in the TP. A major challenge in improving or evaluating ET algorithms is the scarcity of ground measurements, which highlights the need for the long-term comprehensive observation network in the TP (Ma et al., 2020; Zhang et al., 2021). Furthermore, to improve the accuracy of the estimated ET, it is recommended to use regionally optimized forcing data, e.g., climate reanalysis data, which account for the specific climate of the TP with higher accuracy and resolution (He et al., 2020).

### 4.3 Differences in ET components

Previous studies have mostly focused on the total net vapour flux, e.g., magnitude, spatial variability, temporal trend, etc., while the ET components have not been fully investigated. The partitioning of ET into its components, such as soil evaporation (Es) and plant transpiration (Ec), can vary significantly between different datasets. These components reflect the different water phase transitions and vapour flow processes that are regulated by different factors, i.e. vapour flow within plant leaves is mainly controlled by the stomatal behaviour in response to environmental conditions, soil evaporation is controlled by soil structure and water content, the rainfall interception is determined by canopy morphology and rainfall intensity, and vapour transport after sublimation is determined by near-surface boundary layer conditions and the higher latent heat of sublimation. A recent study shows that the contributions of Es, Ec, and Ei to total ET are 68.2 %, 23.6 %, and 8.2 %, respectively, at the Three Rivers Source of the TP (Zhuang et al., 2024). Our study suggests that soil evaporation is the largest contributor to total ET in the whole TP, and further study should be given more attention in further studies. We also found that the evaluation of different ET components is still limited due to the scarcity of available data, and comprehensive evaluations based on more observations would help to further evaluate the ET components and improve the algorithm performance.

This discrepancy in the ET partitioning across different datasets cannot be explained by a single factor, and it is difficult to say which one plays a dominant role as they all contribute in some way to the uncertainty in modelling ET, and may even compensate for each other. In general, these differences stem from factors such as differences in the forcing data, model structure and parameterization, spatial and temporal resolution of the products, and the assumptions embedded in each dataset.

Differences in the forcing data. The forcing data could lead to differences in both the total ET and its components. This explains why GLEAMv35a and GLEAMv35b showed different ET partitioning results, although they are based on exactly the same algorithm. ETMonitor uses GLASS-MODIS data (LAI, FVC, and albedo), PMLV2 use the official MODIS dataset (LAI, albedo, and emissivity). A study by Li et al. (2018) has shown that GLASS LAI is more accurate than MODIS LAI, and

540 MODIS LAI is much lower than GLASS LAI in the eastern TP, which partly explains the relatively lower Ec values by PMLV2 than ETMonitor. Moreover, they also use different meteorological datasets. GLDAS-CLSM uses ERA5 data, while GLDAS-Noah and GLDAS-VIC use GLDAS-2.1 meteorological forcing data as input. A recent study shows that GLDAS-2.1 highly overestimates relative humidity during spring and winter time (Xu et al., 2024), which may lead to lower Es.

Model structure and parameterization. As a most intuitive example, GLDAS-VIC and GLDAS-Noah share the same forcing

data, but the estimated ET partitioning differs significantly. GLDAS-VIC gives a much higher Ec/ET and lower Es/ET, consistent with previous studies. This is most likely due to the weaker soil moisture-ET coupling in the applied physical scheme (Feng et al., 2023). The extremely high Ec/ET ratio is mainly due to the "big leaf" vegetation scheme, which assumes that there are no canopy gaps or exposed soil between plants, so that soil evaporation only occurs in unvegetated areas (Bohn and Vivoni, 2016; Sun et al., 2021). It has also been reported that VIC model, with FVC set to 1 as default value, significantly

overestimate Ec and suppresses Es in sparse vegetation types with a true FVC between 0.1 and 0.5 (Schaperow et al., 2021). In contrast, GLDAS-CLSM tends to underestimate the Ec/ET ratio and overestimate Es/ET, possibly due to parameterization issues related to the soil or vegetation resistance, or the non-traditional approach of accounting for subgrid heterogeneity in soil moisture (Feng et al., 2023; Sun et al., 2023). CLSM estimates of ET are adjusted by varying the sub-ranges of soil water availability, i.e. the saturation, transpiration and wilting sub-ranges (where transpiration is shut off), which differs from the

continuous soil water stress function used in other models. Some other factors, such as the absence of irrigation and the data assimilation procedure, could also affect the ET partitioning in GLDAS models (Li et al., 2022).

Calibration of model parameter. Some ET algorithms may have been calibrated and evaluated against different observations, which can lead to variations in the model performance and, consequently, the partitioning of ET. The global ET datasets use default parameters assigned according to land surface characteristics, which are inappropriate for TP and certainly contribute

to differences in ET partitioning. Many studies have also highlighted the importance of parameter optimization to reflect the local vegetation and soil properties for modelling ET processes (Xu et al., 2019; Zheng et al., 2022).

Effects of spatial heterogeneity and resolution. Higher spatial resolution data may more accurately capture details of the local variability in land surface characteristics and associated vapour fluxes in heterogeneous areas (Chen et al., 2019), leading to differences in ET estimates compared to coarser resolution datasets.

### 4.4 Water vapor released by the TP

#### 4.4.1 ET magnitude and variability in the TP

This study confirms the large discrepancy in the magnitude of ET among different products, as previously reported (e.g., Wang et al., 2020; W. Wang et al., 2018; X. Li et al., 2019). It also shows significant differences in the spatiotemporal distribution of ET and ET components according to different products. Our study suggests that the ET over the TP ranges from 224 mm/yr to 519 mm/yr depending on the products used, with a mean (median) value of 333.1 (339.8) mm/yr and a standard deviation of 42.5 mm/yr. ET accounts for about 52% of the total annual precipitation. This study focused on the vapor released into the atmosphere, while the downward vapor flux (mainly condensation) was not considered. A recent study based on ERA5 reanalysis data found that the annual mean condensation in the TP is about 8.45 mm/yr, which accounts for roughly 2% of the upward vapor flux (Li et al., 2022). We also noticed that the boundary of the TP used in this study differs from that used in some previous studies (e.g., Wang et al., 2020; Ma et al., 2021). The boundary we adopted is more reliable because it is based on geomorphology and formation processes that take into account factors such as elevation, hydrological watershed, etc., which we believe is more appropriate for the analysis of land surface processes (Zhang et al., 2013; Zhang et al., 2021).

Due to the heterogeneity of the climate and land surface, the dominant processes vary between the different sub-regions of the TP. For example, plant transpiration is expected to be the dominant process in the humid plant-soil systems that are more common in the eastern and southeastern TP, whereas soil evaporation is expected to be the dominant process in the central to western TP where arid sparse-vegetated or bare soil cover is prevalent. The difference between these processes certainly affects the magnitude of ET. In the eastern and southeastern TP where ET is higher due to the humid climate and high vegetation cover, there are strong correlations between Rn and ET indicating that the water and carbon cycles play an important role and that the stomatal openness and closure of plant leaves are closely related to the radiation forcing. In contrast, in the central and western TP, there are high correlations between precipitation and ET due to the cold arid climate and sparse vegetation cover, i.e. abiotic processes are dominant. Plants tend to grow more (high LAI) in regions where water is abundant, while higher LAI leads to higher Rn due to the generally low albedo of vegetation compared to soil. This may be more important in the energy-limited regions of the southeastern TP.

#### 4.4.2 Impact of cryosphere on surface water flux

The dynamics of cryosphere elements, such as glaciers and snow, have a significant impact on hydrological processes. Snow/ice sublimation is one of the most important aspects of water resources and hydrology at high altitude (MacDonald et al., 2010). Sublimation is a major contributor to the decrease in snow cover during winter. This study found that snow/ice sublimation in the TP is about 14 mm/yr (median value of different products). It may lead to an error of 4% if sublimation is not taken into account when estimating the total vapour flux released from the TP to the atmosphere. Sublimation from snow and ice surfaces occurs mainly at high elevations when snow/ice covers large parts of the catchments and atmospheric

conditions are cold and dry, as dictated by the Clausius-Clapeyron equation. The maximum sublimation value is higher than 100 mm/yr in TP (Figure 12). A recent observational study of the Langtang Valley in the central Himalaya of Nepal showed that snow sublimation was 32~74 mm/yr during 2017~2019 (Stigter et al., 2021), which is consistent with the ETMonitor estimation (48 mm/yr) (Zheng et al., 2022). Meltwater from glaciers is a significant proportion of the water available down-stream, which also increases ET. A study has reported contrasting trends in ET in the central TP between a wetland replenished by glacial meltwater and a nearby alpine steppe with water supply by precipitation only (Ma et al., 2021).

## 5. Conclusions

To clarify the magnitude and variability of water vapour released to the atmosphere in the TP, this study evaluated f 22 ET products in the TP in terms of accuracy, spatial and temporal variability and ET components. The accuracy of the ET products was evaluated against either eddy covariance observations or basin-scale estimates of the water balance. The spatiotemporal variability of ET and its components was evaluated. The main conclusions were:

- The high-resolution remote sensing-based ET data from ETMonitor and PMLV2 generally showed high accuracy a comparable to the regional MOD16-STM ET product, with overall better accuracy than other fine spatial resolution (~1km) global ET data. The accuracy of these ET estimates was confirmed by the comparison with the water balance-based ET at basin scale, which further indicated overall accuracy of GLEAM and TerraClimate for the coarse-resolution ET products.

- The median and mean values of annual ET in the TP, according to the different products evaluated in this study, are 339.8 mm/yr and 333.1 mm/yr respectively, with a standard deviation of 38.3 mm/yr. Different products showed different spatial and temporal patterns, and large deviations occurred in the central and western TP. Most products showed an increasing trend of annual ET in the TP from 2000 to 2020, with the annual rate varying between data products.

- The separate contributions of the different components, i.e. plant transpiration, soil evaporation, interception loss, open water evaporation, and snow/ice sublimation, vary considerably between data products, even in cases where total ET is in good agreement between the different products, and soil evaporation accounts for the majority of ET. The contributions of open water evaporation and snow/ice sublimation are also not negligible.

**Acknowledgements:**

This work is funded by the Second Tibetan Plateau Scientific Expedition and Research Program (Grant No. 2019QZKK0206, 2019QZKK0103) and National Natural Science Foundation of China (Grant No. 42171039). This work is also supported by the International Fellowship Initiative of CEOP – AEGIS and CSC Fellowship. MM acknowledges the MOST High Level Foreign Expert program (Grant No. G2022055010L) and the Chinese Academy of Sciences President's International

**Author Contributions:**

Dr. C. Zheng: Conceptualization, Methodology, Software, Validation, Formal analysis, Investigation, Resources, Data curation, Writing - original draft, Visualization. Prof. L. Jia: Conceptualization, Methodology, Resources, Data curation, Writing
- review & editing, Supervision, Project administration, Funding acquisition. Dr. G Hu: Methodology, Validation, Writing - review & editing. Prof. M. Menenti: Conceptualization, writing - review & editing. Dr. J. Timmermans: Writing - review & editing.

**Data availability:**

The data in study are all from open accessed datasets. FLUXNET data is available from data portal of fluxnet
(https://fluxnet.org/ ). The ChinaFLUX data is available from the data portal of National Ecosystem Research Network of China (http://www.cnern.org.cn/). TORP data and TPHiPr precipitation data is available from National Tibetan Plateau Data Center (http://data.tpdc.ac.cn/). GLASS datasets are available from the University of Maryland (http://glass.umd.edu/). MERRA2 and GLDAS data are available from the Goddard Earth Sciences Data and Information Services Center (https://disc.gsfc.nasa.gov/). The MODIS datasets are available from NASA Earthdata Search
(https://search.earthdata.nasa.gov/). The ETMonitor ET is available from CASEARTH Data Sharing and Service Portal (https://data.casearth.cn/ ). The GLEAM product is available from its official site (www.gleam.eu). The MOD16-STM, EB ET, CR, PMLV2 and PMLV2-Tibet ET datasets are available from TPDC (https://data.tpdc.ac.cn/). The FLUXCOM ET dataset is available from its official website (www.fluxcom.org). The ERA5 and ERA5-Land datasets are available from the Copernicus Climate Data Store (https://cds.climate.copernicus.eu/). SSEBop is available from USGS (https://earlywarn-
ing.usgs.gov/ssebop/). TerraClimate is available from Climatology Lab (https://www.climatologylab.org/). SynthesisET is available from the Harvard Data public repository (https://doi.org/10.7910/DVN/ZGOUED). BESSv2 is available from the Seoul National University (https://www.environment.snu.ac.kr/bessv2).

**Conflicts of Interest:**

The authors declare that they have no conflict of interest.

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
