# Peer review of "How much water vapour does the Tibetan Plateau release into the atmosphere?"

_Hydrology and Earth System Sciences, 2024_

## Author Comment (AC1)

We are very grateful to Reviewer #1 (Dr. Marloes Mul) for the in-depth reading and the thorough review we received. We present below our detailed reply to the discussed points and further revision plan. The reviewer's comments appear in black and our responses appear in blue.

REVIEWER #1

I read the manuscript "How much water vapour does the Tibetan Plateau release into the atmosphere?" with great interest. The validation of many different ET products over these water towers of Asia has a lot of value. While the manuscript is generally well written and clear, I do have some specific comments and requests for clarification of the presented analyses.

Reply: We thank you for the review and the constructive feedback that will help us to further improve our work.

Regarding the validation:

- Provide clear explanation on the temporal scale the analyses were conducted (monthly?), this is not always clear

Reply: Both the validations based on eddy covariance observations and the basin-scale water balance method were conducted at monthly scale. We will make it clearer in the revised version.

- Provide clear explanation on the period used for the analyses (in some cases the overlap of the *in situ* data (either EC towers or water balance estimates) and products is rather short

Reply: It is true that the overlap period of the *in situ* data and products is short in some instances, and in some case such as the Namco site there is no overlap, since the in-situ measurements started in 2019 while some products did not extend beyond 2019. As regards the site-scale validation, we will add a table in the supplementary materials to include the information necessary for this purpose, i.e. the overlap period for validation, the number of observations, and values of the error metrics. As regards the basin-scale validation, the validation period will also be added in the main text.

Our approach was to utilize long time series data (as long as possible) for the inter-comparison and trend analyses. More precisely, for the inter-comparison analysis we used the overlap period of all products (2003~2013). The trend analysis was carried out for the available period for each dataset, being aware that the overlap period of all products was relatively short. We note that many satellite remote sensing ET datasets with high spatial resolution are estimated based on MODIS data, which started from 2000, while long-term ET datasets at comparable spatial resolution are still scarce.

- Basins used in the water balance estimation is not always clear, eg figure 1 doesn't show the Heihe basin (is this the Hexi corridor and is the entire basin included in the map/analyses?). In figure 1 what does the stripped area refer to? A table with information would be useful with some additional information on the data used from the studies by Ma and Zhang and Wang et al. Also the basins are referred to as the Yangtze/ Yellow river basin, but as far as I understand these only cover the upper part of the basin. Please provide some additional information on the extent of each of the basins analyses (eg provide name of the gauging station where the basin was

delineated). Also in figure 3, there is a reference to TP, which basin/ area does this refer to (the entire TP area shown in figure 1 or the area of all the basins combined, which are two different areas)?

**Reply**: Sorry for the ambiguities in some of our illustrations and related information. We will also revise the figures and add more information accordingly. Overall, we used monthly $ET_{wb}$ from five basins from previous studies (Ma and Zhang., 2022; Wang et al. 2021), including the headwaters of Yellow basin (UYE), headwaters of Yangtze basin (UYA), upper Heihe basin (UH), Inner Tibet Plateau (INTP) and Qaidam (QDM) basins. It is true that these only cover the upper part of the basin, and we will define explicitly the extent of these regions and present this information clearly. A new table will be added to provide additional information, i.e., the extent of the basins and the names of the gauging stations.

As regards Figure 3, we intended to use TP to represent the area of all the five basins combined. To avoid the potential misunderstanding, we will revise it to *5 basins* (the area of all the basins combined) in the revised version.

- Color scheme of figure 3 is not fully intuitive, for example the r2 is deep red for high (=good) values)

**Reply**: We will redraw the figure to make it more intuitive.

Figure 5: what do the different colors of the bars mean?

**Reply**: We intend to show the global satellite remote sensing-based ET dataset in dark blue and model-based ET dataset in light blue, and the regional ET dataset in red. We will add this description in Figure 5.

Trend analyses (figure 7):

- The calculation of the trends could be affected by an exceptional year with high or low ET at the beginning or end of the time series (since there is quite some yearly variation and the trends are often relatively minor). Could you say something about the significance of these trends as well? Also for the SynthesisET both the first two years and the last two years seem to be outliers and related to the "temporal inconsistencies" of the product. Was this data properly vetted before including in the analyses?

**Reply**: We fully agree with you that the trend could be affected by the exceptional years at the beginning or end of the time series. This is also why we choose a robust regression method to estimate the trend of ET, rather using simple linear regression, since the robust regression can reduce the impact of outliers. We will add the significance level of the trends in the figure and main text.

As regards the temporal inconsistencies of SynthesisET, we carefully checked it for several times and we are pretty sure about the existence of the temporal inconstancies. In fact, this issue was also noticed by the authors of the SynthesisET dataset, and they tried a different synthesis strategy d in a later regional study on the Northern China (Wang et al., 2021). This could also be seen from the

temporal variation SynthesisET in Figure RC1-R1. Figure RC1-R1 will used to replace the Figure 7 in the manuscript.

[Figure]

Figure RC1-R1: Yearly variation of ET in the TP by different products. The inset panel shows the annual ET trend by different products. *: trend with significance level (p<0.05).    In the top panel, the reanalysis data is shown in dotted line, and the land surface model-based data is shown in dash line.

We also check the spatial variation of ET by SynthesisET (as shown in following Figure RC1-R2). Before 2000, SynthesisET showed quit high ET values (e.g., in the eastern TP). While after 2019, SynthesisET showed extremely low ET values in the eastern TP.

[Figure]

SynthesisET, July, 2000          SynthesisET, July, 2008

SynthesisET, July, 2013          SynthesisET, July, 2019

Figure RC1-R2: Example of spatial variation of ET by SynthesisET in July of different years.

Reference:

Wang, L.;Wu, B.; Elnashar, A.; Zeng, H.; Zhu,W.; Yan, N. Synthesizing a Regional Territorial Evapotranspiration Dataset for Northern China. Remote Sens. 2021, 13, 1076. https://doi.org/10.3390/rs13061076.

- Why are many of the products with longer time series (eg ERA5Land, SynthesisET, BESS, MERRA2) not presented with their full timeseries?

**Reply**: The ERA5Land ET shows very similar trend with ERA5. As regards SynthesisET, we already noticed its temporal inconsistence, thus we did not include it in the annual trend analysis. To reduce the concerns of reviewers, we will include all the long-term ET products in the revised version.

Analyses of "ET components"

- As mentioned by the authors these different sub-components of ET are not validated and with the wide range of values derived from the different products, what conclusions can really be drawn? This is especially a question for the open water ET (maps in figure 9 shows large areas evaporating from water surfaces) and sublimation (which is validated how?)

**Reply**: It is true that the evaluation of different ET components was still limited due to the scarcity of available data and a comprehensive evaluation based on more observations would help to further evaluate the ET components and improve the algorithm performance. This analysis on the ET components has not been fully investigated by previous publications. We intended to use it to explain the difference among ET products and to answer the question: which processes play a significant role in determining the total ET. We also noticed that previous studies mostly focus on total ET, e.g., magnitude, spatial variation, temporal trend, etc., while the ET components were not fully investigated. Meanwhile, many studies estimated based on big-leaf model, and a few studies estimate total ET based on the separate estimation of ET components. These components reflect the different water phase change processes that are regulated by different factors, e.g., transpiration is mainly controlled by the plant physiology through the regulation of stomata behavior, soil evaporation is determined by heat and mass transfer in the top soil with liquid water present at some depth below the surface t, the rainfall interception loss is mainly related to the canopy morphology and rainfall intensity and the sublimation is associated with higher enthalpy change than vaporization process and near surface air humidity and temperature. So, we believe this analysis on the ET components is helpful, because at least starts with treating correctly each water phase change.

It is important to note that reliable independent reference measurements on each component of total vapour flux are very scarce. The anonymous Referee #3 (RC3) suggest us to use the ensemble mean of the ET components by different products, which may be close to the truth. We will try to check if it works. We also notice that averaging properly would not provide good estimates, since the it applies only to random errors, not to the use of the wrong algorithm. According to the results in Section 3.2.3, the median values of the ratio of Es, Ec, and Ei to total ET is 50%, 30%, and 5%. A recent study shows the contributions of Es, Ec, and Ei to total ET are 68.21 %, 23.57 %, and 8.21 %, respectively in the Three Rivers Source of the Tibetan Plateau (Zhuang et al., 2024), which is

actually quite close to our estimates. After the analysis in our study, we may generally conclude that soil evaporation (Es) contributes most to total ET in the whole TP, and further study should pay more attention to it.

Reference:

Zhuang, J., Li, Y, Bai, P, Chen, L, Guo, X., Xing, Y., Feng, A, Yu, W., Huang, M.: Changed evapotranspiration and its components induced by greening vegetation in the Three Rivers Source of the Tibetan Plateau. J. Hydrol., 633, 130970, https://doi.org/10.1016/j.jhydrol.2024.130970, 2024.

Analyses related to the "response to different environmental factors"

- The purpose of these analyses are not entirely clear to me. First, the analyses are done for the median value of the correlation, whereas it was already very clear that there is a large variance between the different products. Also several products utilize these input data (Rn, LAI, P) for estimating ET, how is this kind of dependency considered in the analyses? Do different types of models have stronger or weaker correlation with these environmental factors? And what does that mean for the interpretation of the analyses?

**Reply**: Thank you very much for the comments. Analyzing the impact of environmental factors on ET is helpful to reveal the governing factors and the mechanisms determining the variability of ET. Itis also helpful to analyze why and how the ET algorithms/product capture the ET variation caused by the environmental change. It is also true that different models have stronger or weaker correlation with these environmental factors, which indicate the observed response to forcing factors is algorithm dependent. Meanwhile, several products utilize these input data (Rn, LAI, P) for estimating ET, and these products may show higher correlation with these factors. Hence, we think both the algorithm itself and the input data can impact the response of estimated ET to environmental factors.

We also noticed that the current analysis is very limited and a more comprehensive analysis could be done to illustrate this issue better. Hence, we will remove it from current manuscript and prepare another paper on it for a more robust analysis.

- Did any of these factors also influence the partitioning of ET into ETc and ETs?

**Reply**: We did not mention this issue in the manuscript. But, we think the answer is yes. This is especially true for leaf area index. Higher leaf area index is generally associated with higher plant transpiration and interception loss. For example, a recent study shows that the vegetation greening (judged by increasing LAI by 0.009 $m^2/(m^2\ a)$ with $p < 0.05$) caused unequal different changes in ET and its components, i.e., 1.95 mm/a, $-2.41$ mm/a, 1.33 mm/a, and 3.03 mm/a for ET, Es, Ec, and Ei, respectively, in the Three Rivers Source of the Tibetan Plateau (Zhuang et al., 2024), which clearly indicates its influence on the ET partitioning.

Reference:
Zhuang, J., Li, Y., Bai, P., Chen, C., Guo, X., Xing Y., Feng A., Yu W., Huang, M.: Changed evapotranspiration and its components induced by greening vegetation in the Three Rivers Source

of the Tibetan Plateau. Journal of Hydrology, 633, https://doi.org/10.1016/j.jhydrol.2024.130970, 2024.

Discussion:

- General reflection of the validation methods employed, doesn't really add much information. The incorporation of seasonal land cover conditions or lack thereof is only explained for 3 products, but then no reflection on how that has affected the results. Or how relevant negative latent heat fluxes are (does this happen often or only occasionally?). The reflection on the water balance estimations are also very general and could have been included in the introduction (there is no reflection based on this specific study). For example, the assumption of not incorporating meltwater could have been explained in the method but is not an outcome of this research.

**Reply**: We understand the reviewer' concern, and will revise the manuscript to focus more on the findings of the current study. We will further revise our discussion by focusing more on the topic and results of this study.

The in-situ observation by eddy covariance system is recognized as the standard method for monitoring energy and mass fluxes to validate high-resolution ET (Baldocchi, 2020). For example, Chen et al. (2024) validated several ET products with spatial resolution ranging from 1km to 50km by comparing ET estimates with eddy covariance observations at site scale. However, it is important to note that the tower-based eddy covariance observations have a very small footprint (approximately several hundred square meters depending on the weather conditions). Consequently, the direct comparison of site-scale observations with the coarse-resolution ET products (e.g., 25km) is problematic due to the severe problem of spatial mismatch of footprints. A more comprehensive validation approach, that considers both the in-situ measurements and basin-scale estimations, has been suggested to improve the reliability of estimated accuracy (Liu et al., 2023). Therefore, to increase the quality of our validation results, we also included validation based on basin-scale estimates of ET, which have a much larger footprint (roughly several hundred to kilo meters depending on meteorological conditions). Considering the relatively sparse distribution and small footprint of the flux-tower based eddy covariance observations, the water balance method is an useful validation method, especially of the coarse-resolution ET data products.

In this study, these two validation methods showed generally consistent results when validating the high-resolution ET. If judged by the KGE of site-scale validation, the accuracy of the high-resolution ET products can be ranked as: PMLV2 > ETMonitor > MOD16STM > GLASS > MOD16 > SynthesisET > SSEBop. If judged by the KGE of basin-scale validation, the accuracy of the high-resolution ET products can be list as: ETMonitor > PMLV2 > MOD16STM > SSEBop > GLASS > MOD16 > SynthesisET. Although both indicate that ETMonitor, PMLV2, and MOD16STM are most accurate and the rest four are less accurate among the high-resolution ET products, there was a difference in the ranking of ET products. This is probably related to the processes captured by these two validation methods. The eddy covariance observations captures the net water vapour flux integrated across different processes at certain point and during a certain period of time, i.e. plant transpiration in dense vegetation regions, snow sublimation in dry snow regions, evaporation of canopy-intercepted water when the canopy is wet due to intercepted rainfall, and the observed vaporization depends on the land site condition during the observation period, which may vary

seasonally and yearly due to factors such as snow/ice occurrence, intercepted water and vegetation growth. The basin-scale water balance estimated *ETwb* is essentially the residual of observed liquid water fluxes, which is assumed to be the net water loss to the atmosphere at basin scale. Compared with the site-scale observations, the basin-scale *ETwb* can capture the impact of land cover change within a large catchment on the ET. For example, the mean water level of lakes in the TP increased by 0.20 m/yr from 2000 to 2009 and lake water mass increased significantly (Zhang et al. 2013), which surely caused higher ET in TP since open water evaporation is generally higher than other land cover types. However, most ET products (e.g., MOD16, PMLV2, etc.) assume constant land surface conditions throughout the year or multiple years, which indicates that they cannot capture the temporal changes of these vaporization process associated with changes in land cover. In contrast, ETMonitor adjusts the daily land cover based on seasonal land cover condition (water cover and snow/ice cover), which enables it to partly reflect the impact of seasonal and yearly extent of liquid and solid water on total ET (Zheng et al., 2022). This probably explains partly why ETMonitor performs slightly better than PMLV2 when validated by basin-scale water balance methods, while it is the opposite when validated with in-situ observations.

Reference:

Liu, H, Xin, X, Su, Z., Zeng, Y., Lian, T., Li, L., Shanshan S.: Hailong Zhang Intercomparison and evaluation of ten global ET products at site and basin scales. J. Hydrol., 617, 128887, https://doi.org/10.1016/j.jhydrol.2022.128887, 2023.

Zhang, G., Yao, T., Xie, H., Kang, S., and Lei, Y.: Increased mass over the Tibetan Plateau: From lakes or glaciers?, Geophys. Res. Lett., 40, https://doi.org/10.1002/grl.50462, 2013.

Zheng, C., Jia, L., and Hu, G.: Global land surface evapotranspiration monitoring by ETMonitor model driven by multi-source satellite earth observations, J. Hydrol., 613, 128444, https://doi.org/10.1016/j.jhydrol.2022.128444, 2022.

Baldocchi, D. D.: How eddy covariance flux measurements have contributed to our understanding of Global Change Biology, https://doi.org/10.1111/gcb.14807, 2020.

Chen, X. Yuan, L., Ma, Y., Chen, D., Su, Z., Cao., D.: A doubled increasing trend of evapotranspiration on the Tibetan Plateau. Sci. Bull., https://doi.org/10.1016/j.scib.2024.03.046, 2024.
* * *
- The discussion related to the different types of models comes a bit out of the blue, for example in table 2 the model type is not provided, which makes is difficult to validate a statement such as (first sentence) " PM-type model demonstrated superior accuracy compared to other models". Also ".. models that incorporate soil moisture to detect water stress…" can not be checked, which models do or do not incorporate soil moisture? Also to go in depth into the methodology of each product seems to go beyond the objective of this research, especially since it unclear why some models are singled out and others not (nor a statistical comparison between for example PM vs non-PM models is not done.

**Reply**: We will double check and revise the manuscript to make sure all the necessary information is included and the statements can be easily checked. We already stated in the manuscript that "Among the evaluated ET products, there are 14 products that primarily use remote sensing products, including 2 products (SSEBop and EB) based on land surface temperature (LST), 8 products

(ETMonitor, MOD16, MOD16-STM, PMLV2, PMLV2-Tibet, GLEAMv35a, GLEAMv35b, BESSv2) based on PM-types models (including Penman-Monteith equation, Priestley-Taylor equation, Shuttleworth-Wallace equation), 4 products (FLUXCOM-RS, FLUXCOM-RS-METEO, GLASS, SynthesisET) based on data-driven methods (machine learning method or ET products ensemble method)." To make the information more intuitive, we will move it to the Section 2.2.2. More information on whether soil moisture is considered in a given data product will be added in Table 2 by listing the main forcing data.

Our primary objective is to find out how accurate are the ET products in the TP, which is closely related to the algorithm applied in each product. Since we evaluate 22 products, there are 22 models to be discussed, which is actually too much and will make the manuscript unfocused. Therefore, we discussed the methodology of some representative ET products. The difference between the PM and non-PM model could be checked in Section 3.1.1, which showed that the best three products are all PM -type model-based products (ETMonitor, PMLV2, MOD16STM), while the LST-base (SSEBop) and data-driven products (GLASS and SynthesisET) had overall a low accuracy. We will present this statement more clearly in the revised version.

- The uncertainty of the SynthesisET product was already mentioned in the results section, is this really an important outcome of this research (important enough to single it out in the discussion?)

Reply: Thank you for the comments. In the results section, we evaluated its accuracy and compared with other products to identify a temporal inconsistence. In the discussion section, we try to explain the reason of its relatively poor performance, since we expected the fusion of different datasets should have improved the overall accuracy. We addressed the importance of the ensemble method in the discussion, which might be helpful to guide further studies.

---

## Author Comment (AC2)

We are very grateful to Reviewer #2 (Dr. Prajwal Khanal) for the review and the constructive suggestions. We present below our detailed reply to the discussed points and further revision plan. The reviewer comments appear in black and our responses appear in blue.

REVIEWER 2

The article "How much water vapor does the Tibetan Plateau (TP) release into the atmosphere?" by Zheng et al. provides a comparative analysis of evapotranspiration (ET) on the Tibetan Plateau, an essential yet uncertain component of the water cycle. This comprehensive review examines various streams of ET data and compares them with in-situ flux measurements, aiming to address a significant research gap: Can ET estimates derived from satellite and land surface models accurately reflect in-situ ET observations?

While I appreciate the insights offered by this article, particularly its thorough incorporation of diverse data sources, there are concerns regarding the clarity and completeness of the methodology. Consequently, the obtained results lack sufficient substantiation. Therefore, before publication, these concerns need to be addressed thoroughly.

**Reply**: We thank you for the positive and constructive feedback that will help us to further improve our work.

**Major comments:**

**Regarding the Methodology:**

1. The temporal coverage of the analysis is not clearly defined throughout the article. In line 197, it is written 2003 to 2015, in line 221, 2001 to 2018 while in line 312, it is written 2000 to 2020. These discrepancies need clarification to ensure consistency and accuracy in the reporting of the study period. I suggest keeping the results with consistent temporal coverage in the main section while any other information on supplementary information (SI).

**Reply**: Thank you very much for the suggestion, and we will try to use consistent temporal coverage in the main section. The overlap period difference was caused by differences in the temporal coverage of ET products and *in situ* observations. In section 3.1.1 on the validation of ET products against flux tower measurements, the overlap period for *in situ* eddy covariance observations and high-resolution ET products was in most cases from 2001 to 2018, but there are differences for some sites and products. We will add a table in the supplementary materials to show the temporal coverage of the study for each site and each product for site-scale validation. In section 3.1.2, when the validation of ET products against basin-scale water balance $ET_{wb}$ is presented the temporal coverage was from 2001 to 2015 with some gaps for some catchments and data products. We will also add a table to show the temporal coverage period and integrating the information related to basin-scale validation where necessary. In section 3.2.2, '2000 to 2020' is not the precise temporal coverage of different products. Instead, we determined the median value of ET of all available products for each year between 2000 and 2020 (noted that different year, the different products may be selected). And the median value of ET was further used to obtain the overall trend of ET from 2000 to 2020. We will revise this part to make it clearer.

Furthermore, to avoid unnecessary confusion caused by differences in the temporal coverage, we will try to keep a consistent temporal coverage and add information in each section where necessary. Considering the temporal coverage of all products is from 2003 to 2015, the comparative analysis in Section 2.3.2 and Section 3.2 will be conducted by applying the period 2003~2015, unless gaps in data needs to be taken into account, leading to a different temporal coverage.

2. Although it appears to be conducted at a monthly scale based on the information provided, it is unclear whether all datasets, such as ETMonitor with daily resolution and MOD16 with 8-daily resolution, were aggregated to a monthly scale for comparison or were based on the native resolution of the dataset. Clarity is needed regarding the aggregation process of these datasets to ensure transparency and understanding of the methodology employed.

**Reply**: It is true that the validation was carried out using monthly data. All the products were temporally aggregated to the monthly scale from their native temporal resolutions prior to validation and comparison. The data products with a daily resolution were just added up to obtain the monthly ET values. For the data with 8-days resolution, an average ET value was first estimated for the available data in that month, and the monthly ET value was subsequently obtained by multiplying the averaged values by the number of days in the month. We will add this description to clarify how the monthly data were obtained.

3. In line 135, it is mentioned that months with less than 50% valid daily ET values were excluded from the analysis. However, it remains unclear whether these excluded months were filled to maintain a continuous ET time series or if the comparison was limited to months with more than 50% valid ET values. Clarification on how the missing data was handled and its impact on the analysis is necessary for a comprehensive understanding of the methodology.

**Reply**: The missing data was not further filled and gaps were excluded to avoid the impact of uncertainty introduced by gap-filling. We will state this in the methodology.

4. Providing information on the number of valid observations available for each dataset, either in the supplementary information or elsewhere, would be beneficial for assessing the comparability of sample sizes across datasets, especially if they are not analyzed for same temporal coverage.

**Reply**: Thank you very much. We will add a table in the supplementary materials to include the temporal coverage for site-scale validation and number of valid observations.

**Regarding the results:**

1. It appears that the regional-based formulations of ET, such as MOD16STM and PLMV2 ET Tibet, demonstrated the highest accuracy when compared to in-situ flux towers. However, it is crucial to ensure that the flux stations utilized in this study for comparison were not already included in the calibration of these datasets. If the same flux stations were used for calibration, the greater accuracy of these products may not be fully substantiated. Therefore, it is imperative to verify whether there is any overlap between the flux stations utilized in this study and those used for calibration to accurately assess the reliability of the results.

**Reply**: The issue you mention is very important, and we agree with you that the validation results are influenced by the calibration. As a summary, there are three high-resolution products (ETMonitor, PMLV2, MOD16STM) clearly list their calibration sites, and some of them were involved in this study, while others do not used flux sites in TP for calibration or their information is not presented in their studies. Although some relative coarse-resolution products (e.g., PMLV2-Tibet) were also reported to use flux sites as calibration, they are not validated based on flux site observation in this study considering the mismatch of spatial representative. In this study, we did not exclude the calibration sites when validation for the following reasons:

- The difficulty in maintaining ground-based observations have resulted in a scarcity of flux towers on the TP. If sites were excluded, the validation sites will be scare, which would raise further concern on the sites' representativeness and relevant uncertainty.
- Different products use different sites for calibration, and some studies did not provide such information. Some products are designed with a clear separation between calibration and validation sites, while others do not (e.g., some separate calibration and validation samples using data of different years from same sites, some do not provide clear information at all). It seems to be not practical to separate them clearly.
- To achieve the high accuracy, model calibration is valid approach that utilized by many models before generating datasets. The purpose of the study is to identify how accurate the current ET products are, which are helpful to achieve an ET product with better accuracy, and those effort on model calibration should be encouraged.

To address this, we will include the information on whether the sites were utilized for each ET product calibration in the supplementary materials Tables S1. Meanwhile, we utilize basin-scale validation as a complementary. As our knowledge, there is no products use basin-scale water balance estimates for calibration, which enable it as an independent validation method.
* * *
2. In Figure 3, it is unclear how the metrics were calculated for the entire Tibetan Plateau (TP). Does the metrics for TP represent averages or medians across the basins or was TP treated as a single basin?

**Reply**: Sorry for the caused misunderstanding. We intended to use *TP* to represent the area of all the five basins combined, including headwaters of Yellow basin (UYE), headwaters of Yangtze basin (UYA), upper Heihe basin (UH), Inner Tibet Plateau (INTP) and Qaidam (QDM) basins. To avoid the misinterpretation, we will revise it to *5basins* (the area of all the basins combined) in the revised version. A new table will be added to provide additional information in the Supplementary., We simply used all samples (each sample represents a valid group of reference data and to-be-validated ET data from one basin) from all 5 basins to estimate the metrics for the *5basins*.
* * *
3. In Figure 4, the color bar for ET standard deviation (ETsd) differs from the color bars used for other variables. This inconsistency can lead to confusion, particularly since the figures are presented together. Also, if possible, please keep the results in the order of datasets that appears in the Table 1.

**Reply**: Thank you for the suggestion. We will revise the figure accordingly to avoid the confusion. We will also move the information on the spatial variability of ET in each product to the supplementary materials to make the manuscript more concise.

4. Regarding Figure 7, it would be beneficial to highlight the trends observed specifically from data with long records to discern the presence of significant trends in ET, because the trend calculated with only some years of data would not add any conclusion to the overall trends in the ET. Additionally, it's essential to clarify how the trends were calculated—whether through linear regression or another method—and whether the significance of these trends was assessed.

**Reply**: Thank you for the suggestion. We also noticed that the trend could be affected by the temporal coverage of the ET time series, and we also agree that longer records provide more reliable information on trends. We will identify the trends estimated with long records in the revised version. We believe that relatively shorter data records (especially in recent years) remain relevant to document differences across data products, so we kept the results on trends after 2000s.

We choose robust regression method to estimate the trends, rather than using simple linear regression, since the robust regression reduces the impact of outliers. We will add the significance level of the trends in the figure and main text.

**On results specific to "Response of the ET to main governing factor."**

The author's intended message or purpose behind the analysis is not clearly conveyed. It seems to explore the relationship between annual ET and various water, energy, and vegetation variables. I will try to highlight my concerns in points here:

1. In my belief, the analysis of how annual ET responds to different water, energy, and vegetation variables could potentially be a separate study requiring a more comprehensive approach.

**Reply**: Thank you very much for the suggestion. We agree with you that the response of ET to water, energy, and vegetation variables could be a topic to be addressed by a more comprehensive analysis. We will remove it from current manuscript and prepare another paper on it for a more robust analysis.

2. For instance, If Leaf Area Index (LAI) correlates well with both/or net radiation (Rn) and precipitation (P), which I believe will be the case, raises doubts about the conclusions drawn regarding the relative influence of these variables on evapotranspiration (ET). This is true especially when conclusion on influence of these variables on ET is drawn simply from correlation of ET with these variables without controlling for the other confounding factors. To check whether this is the case or not, we can simply correlate LAI with Rn and P, as well as by correlating Rn with P.

**Reply**: It is true that LAI correlates with both Rn and P, and we will check the correlation between LAI with Rn and P. LAI is a critical variable that correlates with several climatic and environmental factors as it represents the amount of leaf area per unit ground area and characterizes the canopy structure. LAI influences the interception of radiation and the distribution of light within the canopy, which in turn affects the energy balance of the surface, e.g., net radiation (Rn) and latent heat flux (LE). Meanwhile, plant generally growth better in the regions with sufficient water supply (high precipitation) and adequate APAR (highly related to Rn).

3. Even if one were to accept the current analysis, which I personally disagree with for the reasons outlined in points 2, there remains a crucial need for clarification regarding the rationale behind correlating median ET from all datasets (if I understood it properly) with environmental variables (Figure 10). This need arises primarily from the significant variability observed among different ET datasets in terms of magnitude and hence I believe that the relative importance assessed from the simple correlation of ET with these variables will also vary. Consequently, any conclusions drawn from these correlations may lack robustness.

**Reply**: We agree that there could be issues with determining the independent effects of these variables on ET if only simple correlations were used. Correlation does not imply causation, and more sophisticated statistical methods need to be used, e.g., multiple regression analysis, to control for confounding factors and to determine the relative influence of Rn and P on ET while considering LAI. This would allow to estimate the unique variance explained by each predictor while holding the others constant.

Furthermore, itis important to consider that the relationships between these variables can be complex and non-linear, and they might be influenced by other factors such as soil moisture, air temperature, humidity, wind speed, and atmospheric pressure. To accurately assess the relationships and the potential for misinterpretation, we will try to employ a multivariate analysis approach to establish the unique contributions of Rn, P, and other factors on ET, while controlling for the potential influence of other relevant factors to form another paper based on a more robust analysis.

4. Again, in regions where Ec and Ei are the dominant modes of evapotranspiration (Figure 8), it would be valuable to investigate their correlation of ET with LAI compared to Rn and P, after removing the confounding effects.

**Reply**: It is true that the above-mentioned issue for ET is also applicable for Ec and Ei, and the multivariate analysis can be applied to investigate the response of Ec and Ei to environmental factors.

5. Nevertheless, I still believe this could be separate research with robust approach.

**Reply**: Thank you very much for the suggestion again. We will remove this part accordingly and make another draft for a more robust analysis.

**Additional technical comments:**

1. Before highlighting the monthly RMSE, it would be helpful to provide information on the magnitude of monthly ET observed at different flux stations based on in-situ observations. This would allow for a comparison of the magnitude of observed ET with the error represented by the RMSE.

**Reply**: Thank you very much for the suggestion again, we will add a table in Supplementary with the mean value of observed ET.

2. It's advisable to maintain analysis with consistent spatial and temporal coverage in the main section, while keeping analyses involving datasets with inconsistent coverage to the supplementary section. This will enhance clarity of the manuscript.

**Reply**: Thank you very much for the suggestion again, and we will focus on these products with spatial and temporal continuous for the analysis in the main text to retain the results and analysis within the same spatial and temporal coverages. There are two products (MOD16-STM, PMLV2-Tibet) that can not cover the regions outside China, while there are some products that are spatiotemporally continuous. These will be noted when presented or mentioned.

3. In Figure 8, it is noted that while the total evapotranspiration (ET) may appear similar across different datasets, the partitioning of ET between datasets is not consistent. This observation is indeed a significant finding. However, the substantial explanation provided does not sufficiently clarify why the datasets differ so much, particularly for GLDAS and MERRA2.

**Reply**: Thank you for appreciating our findings. The partitioning of ET into its components, such as evaporation from the soil (Es) and transpiration from plants (Ec), can vary significantly between different datasets. This discrepancy in the ET partitioning among different datasets cannot be explained by a single factor, and it is difficult to tell which one plays a dominant role as they all contribute in some way to the uncertainty in modeling ET and may even compensate for each other. Generally, it mainly stems from factors including differences in the forcing data, model structure and parameterizations, spatial and temporal resolution of the products, and assumptions used in each dataset.

- Difference in the forcing data. The forcing data could lead to difference in both the total ET and its components. This forcing data difference cause the different ET partition results by GLEAMv35a and GLEAMv35b, since they are based on exactly the same algorithm. ETMonitor uses GLASS-MODIS data (LAI, FVC, and albedo), PMLV2 use the MODIS datasets (LAI, albedo, and emissivity) provided by NASA. A study has shown that GLASS LAI is more accurate than MODIS LAI, and a much lower LAI was found in the eastern TP in the MODIS LAI than in the GLASS LAI dataset (Li et al., 2018), which partly explain relative lower Ec values by PMLV2 than ETMonitor (higher LAI). Moreover, they use different meteorological datasets. GLDAS-CLSM uses ERA5 data, and GLDAS-Noah and GLDAS-VIC use GLDAS-2.1 meteorological forcing data as input. A recent study shows GLDAS-2.1 highly overestimates relative humidity during spring and winter time (Xu et al., 2024), which possibly lead to lower Es.
- Model structure and parameterization. As a most intuitive example, GLDAS-VIC and GLDAS-Noah share same driving factors, but their partition results differ significantly. GLDAS-VIC presents much higher Ec/ET and lower Es/ET, consistently with a previous study, which is most likely due to issues with the weaker soil moisture-ET coupling caused by the physical schemes (Feng et al., 2023). The extreme high Ec/ET ratio is mainly attributed to the "big leaf" vegetation scheme assuming that there are no canopy gaps or exposed soil between plants, so soil evaporation only occurs in unvegetated areas (Bohn and Vivoni 2016; Sun et al., 2021). Itis also reported that the VIC model sets FVC to 1 as default value and will substantially overestimate Ec and suppress Es in sparse vegetation types with a real FVC between 0.1 and 0.5 (Schaperow et al., 2021). In contrast, GLDAS-CLSM tends to underestimate Ec/ET ratio and overestimate Es/ET, possibly due to the problems related to the parameterization of soil evaporation resistance or vegetation related resistance or the non-traditional approach to consider the subgrid heterogeneity of soil moisture (Feng et al., 2023; Sun et al., 2021). CLSM

estimated ET is adjusted by varying the relative fractions of subregions (saturated region, transpiration region, and wilting region where transpiration is shut off), which is different from the continuous soil water stress function used in other models. Some other factors, e.g., the lack of irrigation, the data assimilation, etc., could also impact the ET portioning results in GLDAS models (Li et al, 2022).

- Calibration of model parameters. These ET products algorithms may have been calibrated and validated against different observations, which can lead to variations in the model performance and, consequently, the partitioning of ET. The global ET datasets use default parameters assigned according to generic land surface characteristics, which may be inappropriate for TP and certainly contribute to the difference of ET partitioning. Many studies have also highlighted the importance of parameter optimization to reflect the local vegetation and soil properties on ET processes modeling (Xu et al., 2019, Zheng et al., 2022).

- Impact of spatial heterogeneity and resolution. Higher spatial resolution datasets may capture local variations in land surface characteristics and water flux more accurately in heterogeneous areas (Chen et al., 2019), leading to differences in ET estimates compared to coarser resolution datasets.

Reference:

Bohn, T.J., Vivoni, E.R.: Process-based characterization of evapotranspiration sources over the north american monsoon region. Water Resour. Res., 52 (1), 358–384, https://doi.org/10.1002/2015WR017934. 2016.

Chen, Q., Jia, L., Menenti, M., Hutjes, R., Hu, G., Zheng, C., and Wang, K.: A numerical analysis of aggregation error in evapotranspiration estimates due to heterogeneity of soil moisture and leaf area index, Agric. For. Meteorol., 269–270, 335–350, https://doi.org/10.1016/j.agrformet.2019.02.017, 2019.

Feng, H., Wu, Z., Dong, J., Zhou, J., Brocca, L., He, H.: Transpiration – Soil evaporation partitioning determines inter-model differences in soil moisture and evapotranspiration coupling. Remote Sensing of Environment, 298, https://doi.org/10.1016/j.rse.2023.113841, 2023.

Li X, Lu H, Yu L, Yang K.: Comparison of the Spatial Characteristics of Four Remotely Sensed Leaf Area Index Products over China: Direct Validation and Relative Uncertainties. Remote Sensing. 10(1),148, https://doi.org/10.3390/rs10010148, 2018.

Li, C., Liu, Z., Tu, Z., Shen, J., He, Y., Yang., H.: Assessment of global gridded transpiration products using the extended instrumental variable technique (EIVD). J. Hydrol., 623, https://doi.org/10.1016/j.jhydrol.2023.129880, 2023.

Li, C., Yang, H., Yang, W., Liu, Z., Jia, Y., Li, S., Yang, D.: Error characterization of global land evapotranspiration products: collocation-based approach. J. Hydrol. 612, 128102 https://doi.org/10.1016/j.jhydrol.2022.128102. 2022.

Schaperow, J.R., Li, D., Margulis, S.A., Lettenmaier, D.P.: A near-global, high resolution land surface parameter dataset for the variable infiltration capacity model. Sci. Data, 8 (1), 216. https://doi.org/10.1038/s41597-021-00999-4. 2021.

Sun, R., Duan Q., Wang, J.: Understanding the spatial patterns of evapotranspiration estimates from land surface models over China. J. Hydrol., 595, 126021, https://doi.org/10.1016/j.jhydrol.2021.126021, 2021.

Xu, C., Wang, W., Hu, Y., Liu. Y.: Evaluation of ERA5, ERA5-Land, GLDAS-2.1, and GLEAM potential evapotranspiration data over mainland China. Journal of Hydrology: Regional Studies, 51, https://doi.org/10.1016/j.ejrh.2023.101651, 2024.

Xu, T., Guo, Z., Xia, Y., Ferreira, V.G., Liu, S., Wang, K., Yao, Y., Zhang, X., Zhao, C.: Evaluation of twelve evapotranspiration products from machine learning, remote sensing and land surface models over conterminous United States. J. Hydrol., 578, 124105, https://doi.org/10.1016/j.jhydrol.2019.124105. 2019.

Zheng, C., Jia, L., and Hu, G.: Global land surface evapotranspiration monitoring by ETMonitor model driven by multi-source satellite earth observations, J. Hydrol., 613, 128444, https://doi.org/10.1016/j.jhydrol.2022.128444, 2022.

**Other comments:**

Overall, there are numerous instances in the text which exhibits repetition and with typos, with numerous lines conveying similar information and occasionally out of context. Therefore, significant restructuring of the article's text is necessary.

**Reply**: We apologize for the repetition and typos, and will go through the manuscript again to improve it.

**For instances:**

1. The passage from lines 60-65 highlights the significant uncertainty surrounding evapotranspiration (ET) estimation on the Tibetan Plateau (TP). However, the paragraph falls short in effectively conveying how the present research differs from existing literature. It is evident that this study introduces novelty to the field, particularly through its comprehensive comparison of various ET products with in-situ observations in TP. This contribution warrants greater emphasis in the introduction section.

**Reply**: Thank you. We will emphasize the novelty of this introduction section in the revised version. Previous validations were generally based on either in-situ measurement by the eddy covariance system or the basin-scale ET estimated by water balance method, which represent the surface net water flux at different scales, while these ET products mainly focus on the upward water vapour flux. Recently, Chen et al. (2024) validated several ET products by comparing with eddy covariance systems observations at site scale. It is important to note that the tower-based eddy covariance systems observations have a very small footprint (approximately several hundred meters depending on the weather conditions). Consequently, any direct comparison of site-scale observations with the coarse-resolution ET products (e.g., 25km) is problematic due to the severe problem of spatial mismatch. A more comprehensive validation combining both the in-situ observations and basin-scale measurements was suggested by (Liu et al., 2023), as done in this study.

Reference:

Chen, X. Yuan, L., Ma, Y., Chen, D., Su, Z., Cao., D.: A doubled increasing trend of evapotranspiration on the Tibetan Plateau. Sci. Bull., https://doi.org/10.1016/j.scib.2024.03.046, 2024.

Liu, H, Xin, X, Su, Z., Zeng, Y., Lian, T., Li, L., Shanshan S.: Hailong Zhang Intercomparison and evaluation of ten global ET products at site and basin scales. J. Hydrol., 617, 128887, https://doi.org/10.1016/j.jhydrol.2022.128887, 2023.

2. The final three paragraphs in the introduction section needs to be rephrased for coherence, eliminating redundancy to convey the message clearly. For example, lines 81-86 present the research questions effectively. However, the same information is reiterated in the following paragraph (lines 88-91) within the main objectives, which essentially duplicates the content. This and other redundancy should be streamlined for clarity.

**Reply**: The questions presented in lines 81-86 serve to identify the scientific problems after the literature review. The last paragraph is intended to clarify the research objectives. To avoid the duplicates, we will revise the introduction to ensure that these issues are addressed in a clear and concise manner.

3. The classifications in the discussion sections (ET based on PM model, LST-based model, data-driven, and LSM type) seem abrupt as they haven't been introduced earlier in the text. Section 4.1 should be emphasized when analyzing the results, as much of the content there appears redundant in the manuscript, despite its scientific validity. This caveat should be highlighted without unnecessary repetition

**Reply**: We will move the classifications in the discussion sections to the Section 2.2.2 ET Products when they are first introduced. We will also revise the results section to emphasize the discussions relevant to Section 4.1.

**On the introduction section, it appears:**

These validations were generally based on either in-situ measurement by the eddy covariance system or the basin-scale ET estimated by water balance method, which represent the surface net water flux that integrates different processes (e.g., plant transpiration for the dense vegetation regions, snow sublimation for the dry snow cover periods for the eddy covariance system observations, even condensation when negative latent heat flux occurs),while these ET products mainly focus on the ET (positive upward latent heat flux), which attributes to the validation uncertainty.

*While in the section 4.1.1, it appears:*
The eddy covariance system observation represents the net water flux integrated across different processes (e.g., plant transpiration                in the dense vegetation regions, snow sublimation during the dry snow cover periods, evaporation of canopy-intercepted water when                the canopy is wet due to intercepted rainfall). The vaporization process observed by the eddy covariance system depends on the land                     surface condition, which may vary seasonally and yearly due to factors such as snow/ice, intercepted water, and vegetation. Meanwhile, eddy covariance system observation includes condensation when negative latent heat flux occurs. Remote sensing-based ET products mainly focus on positive ET (positive upward latent heat flux) and omit processes such as condensation.

These two instances basically convey same information. I do agree this is important point to make reader aware about the validation. However, I think the author could be concise about it and avoid unnecessary repetitions.

**Reply**: In the introduction, we intended to introduce generically the uncertainty caused by the validation method, while in the discussion we focused on the processes captured by tower-based observations, as documented by our findings. To avoid the repetitions, we will revise the introduction and discussions accordingly.

4.  In line 100, it might be more appropriate to adhere to existing climatic regime classifications, such as those based on AI or other established frameworks. Because the term rather "monsoon" is kept here in between arid and humid climate types. So, how "different" is "monsoon" from the humid in these classifications? Or what does that monsoon mean when compared with "arid" and "humid"?

**Reply**: We agree with you that it is more appropriate to use the existing climatic regime classification and monsoon is not a standard climate type. According to the Köppen classification, there are dry, subtropical, temperate, subpolar and polar climate types in the TP. These climate types are influenced by both westerlies and the Asian monsoon, which is also enhanced by the thermal forcing of the TP (Zhou et al., 2009; Wu et al., 2012; Yang et al., 2014). The aridity index (P/PET) or Budyko dryness ratio (PET/P) are also widely utilized to characterize the aridity level. A recent study has shown that the dryness ratio has a large spatial variability in the TP, from humid climate with dryness ratio less than 0.3 to hyper-arid climate with dryness ratio larger than 3 (Feng et al., 2024). For simplification, we will revise it according to the dryness ratio to avoid any ambiguity.

Reference:

Feng, Y., Du, S., Fraedrich, K., Zhang, X., Du, M., Cheng, W.: Local climate regionalization of the Tibetan Plateau: A data-driven scale-dependent analysis. Theor. Appl. Climatol., https://doi.org/10.1007/s00704-024-04916-8, 2024.
Wu, G., Liu, Y., He, B., Bao, Q., Duan, A., Jin, F.F.: Thermal controls on the Asian summer monsoon. Sci. Rep., 2. http://dx.doi.org/10.1038/srep00404, 2012.
Yang, K., Wu, H., Qin, J., Lin, C., Tang, W., and Chen, Y.: Recent climate changes over the Tibetan Plateau and their impacts on energy and water cycle: A review, Glob. Planet. Change, 112, https://doi.org/10.1016/j.gloplacha.2013.12.001, 2014.
Zhou, X., Zhao, P., Chen, J., Chen, L., Li, W.: Impacts of thermodynamic processes over the Tibetan Plateau on the Northern Hemispheric climate. Sci. China Ser. D Earth Sci. 52, 1679–169,. https://doi.org/10.1007/s11430-009-0194-9, 2009.

5. Figure 1: It's not clear what does hashing represent. And for some "red" labels, they are not clear like names around "XG".

**Reply**: We will redraw Figure1 to improve clarity.

6. Equation (1), please write equations of all metrics or skip even KGE. Please make it coherent.

**Reply**: We will write equations of all metrics.

---

## Author Comment (AC3)

We thank you for the review and the constructive suggestions. We present below our detailed reply to the discussed points and further revision plan. The reviewer comments appear in black and our responses appear in blue.

When I read this paper, I found that the author may not comprehensively review the following papers: Chen, X. et al., 2024. A doubled increasing trend of evapotranspiration on the Tibetan Plateau. Science Bulletin.

Yuan, L. et al., 2024. Long-term monthly 0.05° terrestrial evapotranspiration dataset (1982–2018) for the Tibetan Plateau. Earth Syst. Sci. Data, 16(2): 775-801.

Wang, B.*, Y. Ma*, Z. Su, Y. Wang and W. Ma. Quantifying the evaporation amounts of 75 high-elevation large dimictic lakes on the Tibetan Plateau. Science Advances, 2020, 6, eaay8558.

I agree to the author that they have collected more ET products in this study, but the generally conclusions are not really new compared with previous ET studies on the TP. Hereby, I suggest to focus more on ET components verification and their trends. This part has not been fully investigated by previous publications. The ET trends and annual ET estimation does not deserve more energy on it. This means that the title should be also changed. There are also some water balance ET studies. Hereby, this analysis is also not new. Introduction should really have a in depth review of previous work.

**Reply**: Thank you for providing the latest publications and constructive suggestions. Itis true that ET components are important and not well studied, however we think clarify the total ET and ET trends is also helpful, especially considering that differences in ET components can surely lead to different total ET. Although the previous studies by Chen et al. (2024) and Yuan et al. (2024) have demonstrated the difference of area-averaged ET in the TP, they did not investigate the spatial variability of this difference which actually is very large. Furthermore, previous studies on ET mostly applied the old TP boundary, which only includes the region inside China. Recent studies emphasized the geographic integrity of the TP and a new boundary of TP was applied (Zhang et al., 2013; Zhang et al., 2021) and adopted in this study. This boundary is more reliable as it is based on geomorphology and formation processes that considers factors such as elevation and watershed boundaries. Hence, the comparison of ET amount and trend is still necessary. We will strengthen the materials on ET components in the revised version following the suggestion.

Reference

Chen, X. Yuan, L., Ma, Y., Chen, D., Su, Z., Cao., D.: A doubled increasing trend of evapotranspiration on the Tibetan Plateau. Sci. Bull., https://doi.org/10.1016/j.scib.2024.03.046, 2024.

Wang, B., Y. Ma, Z. Su, Y. Wang and W. Ma. Quantifying the evaporation amounts of 75 high-elevation large dimictic lakes on the Tibetan Plateau. Sci. Adv., 6, https://doi.org/10.1126/sciadv.aay8558, 2020.

Yuan, L., Chen, X., Ma, Y., Han, C., Wang, B., and Ma, W.: Long-term monthly 0.05° terrestrial evapotranspiration dataset (1982–2018) for the Tibetan Plateau, Earth Syst. Sci. Data, 16, 775–801. https://doi.org/10.5194/essd-16-775-2024, 2024.

Zhang, Y., Li, B., Liu, L., Zheng, D: Redetermine the region and boundaries of Tibetan Plateau, Geogr. Res., 40, https://doi.org/10.11821/dlyj020210138, 2021.

Zhang, G., Yao, T., Xie, H., Kang, S., and Lei, Y.: Increased mass over the Tibetan Plateau: From lakes or glaciers?, Geophys. Res. Lett., 40, https://doi.org/10.1002/grl.50462, 2013.

The large uncertainty of ET products over the TP has been reported by Chen et al. 2024 and Yuan et al. 2024. The abstract should more focus on the new scientific questions. Please revise the sentence: there is still significant uncertainty regarding the amount of water vapour released by the TP into the atmosphere, otherwise remove it. The abstract should emphasize the innovative results, not repeated information.

**Reply**: We agree with you that Chen et al. (2024) and Yuan et al. (2024) have reported the large uncertainty of ET data products. However, their studies did not mention the spatial variability of the uncertainty and the ET components. We will revise that sentence to 'there is still significant uncertainty regarding the variation of water vapour released by the TP into the atmosphere'.

The response of annual ET to total precipitation, net radiation and leaf area index was explored to present their governing effect on ET, and the results indicated that precipitation effect mostly in the middle and northern TP and net radiation play significant role in the eastern TP. There are many other factors which also influence ET. But they are not included in this paper. In addition, this conclusion is normal as other study. I suggest to remove this weak point from this paper.

**Reply**: Thank you very much for the suggestion. We agree with you that the response of ET to water, energy, and vegetation variables could be done with a more comprehensive analysis. And we will remove it from current manuscript and prepare another paper on it for a more robust analysis.

TP has been indicated before line 60, hereby please replace "Tibetan Plateau" with "TP".
**Reply**: We will revise it accordingly.

Line 61, Chen et al. 2024 and Yuan et al. 2024 have listed the big differences of annual ET estimation for the TP. It is better to cite their results directly, since they have compared most ET product for the TP region.
**Reply**: We will revise it accordingly.

Line 80, these specificities, are you talking about negative latent heat? If yes, please use negative latent heat directly.
**Reply**: We will revise it accordingly.

Line 81, How accurate are these improved ET products, I understand that this question is already answered at least partly in Chen and Yuan`s publication. The snow/ice sublimation is new in this study. I suggest to revise the second question to: which processes play a significant role to the ET components trend. The third question, I did not find the author provide answers to which factor dominant different ET products. Hereby, the introduction should be rewritten and new scientific questions to be raised. Current formation is quite weak and not comprehensive.

**Reply**: It is true that previous studies did some evaluation already of new data products. However, it should be noticed that this validation was conducted only at site scale by comparison of eddy covariance observations and ET products. The tower-based eddy covariance observations have a very small footprint (roughly several hundred square meters depending on the weather conditions), and direct comparison of site-scale observations with the coarse-resolution ET product (e.g., 25km),

suffers severe problem of spatial mismatch. Hence, we only used site-scale observations to validate the high-resolution ET products ($\sim 1 \text{ km}^2$). We used basin-scale $ET_{wb}$ to validate both high-resolution and low-resolution ET product. In this sense, our comparison is more robust and comprehensive. To address this aspect, we will include this point in our revised version.

For the second question, we intend to compare the behavior of total ET amount and its components according to different products. The different components here correspond to different bio-geophysical processes. As regards the third question, we will remove the section on the response analysis.
* * *
The first aim of this paper is already investigated by Chen et al. Please change this point or further deep this aim. Actually, there are many attribution studies of TP ET trend. Please review their studies, then make a revision for the num 3 aim.
**Reply**: The first point is addressed in our reply to the previous comment above.
* * *
Line 94, I don`t really agree that pearson correlation analysis can provide us the response of ET to precipitation, Rn and LAI. Indeed, I don`t suggest to include this correlation analysis in this paper. These analysis weaken this paper, it does not benefit to this work.
**Reply**: We agree with you and will remove it in the revised version.
* * *
Lines 122, These the sites, please correct this error.
**Reply**: We will revise it accordingly.
* * *
Table 2, EB is a daily ET product, not monthly.
**Reply**: We will revise it accordingly.
* * *
Figure 4, SEBS should be EB?
**Reply**: We will revise it accordingly.
* * *
Please revise 'in Tibetan Plateau' to be 'in the Tibetan Plateau' or 'in the TP'.
**Reply**: We will revise it accordingly.
* * *
Figure 5, the figure caption should explain what is meaning for different colored bars.
**Reply**: The global satellite remote sensing-based ET datasets are in dark blue, and the land surface model-based and analysis global ET dataset are in light blue, while the regional ET datasets are in red. We will add this explanation in the revised version.
* * *
Figure 7, it is quite difficulty to recognize which bar represent which product. Add the product name corresponding each would be more useful. All the trends are ended in 2020? Their curves in figure 7 do not exhibit the same end year.
**Reply**: We will revise Figure 7 to include the products' name. It is true that different products end in different years, and the end year for the trend analysis depend on the end year of the products. This information will be added in the revised version.

Line 322, Among these products, there are nine that provide the main components of ET (Ec, Es, and Ei), it is better to directly say that 'Nine products provide …'.

**Reply**: We will revise it accordingly.

It is important to note that there is no independent reference available for the ET components. I suggest to use the ensemble mean of ET components to check their differences with the ensemble mean. Nine products have provided the ET components. It's a lot. Their ensemble may be close to the truth.

**Reply**: Thank you for the suggestion. According to the results in Section 3.2.3, the median values of the ratio of Ec, Es, and Ei to total ET was 50%, 30%, and 5%. The ET partitioning ratio (52%, 43%, and 5%) of ETMonitor is the closest one to the median value. We will also estimate the ensemble mean of ET components by different products and check their differences. Furthermore, we will discuss the likely reasons causing such differences and the reliability of the partitioning results among different products in the discussion. For example, there already reports that the overestimation of the Ec/ET ratio by GLDAS-VIC and GLEAM is due to the "big leaf" vegetation scheme assumption that there are no canopy gaps or exposed soil between plants, so soil evaporation only occurs in unvegetated areas (Bohn and Vivoni 2016; Sun et al., 2021; Miralles, et al., 2016). In contrast, GLDAS-CLSM tends to underestimate the Ec/ET ratio and to overestimate Es/ET, possibly due to the parameter problems related to the soil evaporation resistance or vegetation related resistance or the non-traditional approach to consider the subgrid heterogeneity of soil moisture (Feng et al., 2023; Sun et al., 2021). Therefore, to avoid the bias due to these already-known uncertainties, we will remove these products in the calculation of the ensemble mean values.

Reference:

Bohn, T.J., Vivoni, E.R.: Process-based characterization of evapotranspiration sources over the north American monsoon region. Water Resour. Res., 52 (1), 358–384, https://doi.org/10.1002/2015WR017934. 2016.
Feng, H., Wu, Z., Dong, J., Zhou, J., Brocca, L., He, H.: Transpiration – Soil evaporation partitioning determines inter-model differences in soil moisture and evapotranspiration coupling. Remote Sensing of Environment, 298, https://doi.org/10.1016/j.rse.2023.113841, 2023.
Sun, R., Duan Q., Wang, J.: Understanding the spatial patterns of evapotranspiration estimates from land surface models over China. J. Hydrol., 595, 126021, https://doi.org/10.1016/j.jhydrol.2021.126021, 2021.
Miralles, D. G., C Jiménez, Jung, M., Michel, D., & D Fernández-Prieto.: The WACMOS-ET project – Part 2: evaluation of global terrestrial evaporation data sets. Hydrology and Earth System Sciences, 20(2), 823-842, https://doi.org/10.5194/hess-20-823-2016. 2016.

Figure 8, the blue color around TP lakes may not reflect the truth. Please check if this is caused by a wrong lake mask.

**Reply**: We do not use a lake mask here. Please notice that here *Ew* represents the open water evaporation, which actually comes from either lakes or other water bodies, e.g., rivers, snow/ice melt water, flooded pixels, etc.

Figure 9, there are some reports about the annual ET amount for the TP lakes. Please cite these

papers to verify Ew shown in the figure. I understand that Wang et al. Science Advance should also provide the Ew estimation for the TP. This study could benefit to verify the result in the figure.

**Reply**: According to the estimation of Wang et al. (2020), the total water evaporation is about 29.4 $\pm$ 1.2 km$^3$/yr for the 75 lakes with total area of 26,450 km$^2$ (accounts for approximately 56.9% of the lake surfaces over the whole TP), which is much smaller than the actual water cover area. They provide the total lake evaporation (51.7 $\pm$ 2.1 km$^3$/yr) for all plateau lakes based on their selected 75 lakes. We will add some comparison with these results in the revised version.

Section 3.3, this part is not really persuasive. A simple correlation is not meaningful, in addition, other factors were not fully considered, such as air temperature, soil moisture, wind speed etc. In addition, the correlation of abnormal should be analyzed, not the original signal. I suggest to remove this section.

**Reply**: Thank you again, and we will remove it accordingly.

"the daily land cover inputted" please revise this.

**Reply**: We will revise it accordingly.

---

## Author Response (AR1)

**Point-by-point reply to the comments**

This document presents comments by reviewers and our point-by-point reply. The reviewer's comments appear in black and our responses appear in blue.

**1. Reply to the comments by REVIEWER #1**

I read the manuscript "How much water vapour does the Tibetan Plateau release into the atmosphere?" with great interest. The validation of many different ET products over these water towers of Asia has a lot of value. While the manuscript is generally well written and clear, I do have some specific comments and requests for clarification of the presented analyses.

**Reply**: We thank you for the review and the constructive feedback that helps us to improve our work.

Regarding the validation:

- Provide clear explanation on the temporal scale the analyses were conducted (monthly?), this is not always clear

**Reply**: We revised accordingly to make it clear. Both the validations based on eddy covariance observations and the basin-scale water balance method were conducted against monthly observations / estimates.

- Provide clear explanation on the period used for the analyses (in some cases the overlap of the *in situ* data (either EC towers or water balance estimates) and products is rather short

**Reply**: It is true that the overlap period of the *in situ* data and products is short in some instances, and in some case such as the Namco site there is no overlap, since the in-situ measurements started in 2019 while some products did not extend beyond 2019. As regards the site-scale validation, we added a table in the supplementary materials to specify the overlap period for validation, the number of observations, and values of the error metrics. As regards the basin-scale validation, the validation period was also added in the main text.

Our approach was to utilize long time series data (as long as possible) for the inter-comparison and trend analyses. More precisely, for the inter-comparison analysis we used the overlap period of all products (2003~2013). The trend analysis was carried out for the available period for each dataset, being aware that the overlap period of all products was relatively short. We note that many satellite remote sensing ET datasets with high spatial resolution are estimated based on MODIS data, which started from 2000, while there is still a lack of long-term ET datasets with high spatial and temporal resolutions.

- Basins used in the water balance estimation is not always clear, eg figure 1 doesn't show the Heihe basin (is this the Hexi corridor and is the entire basin included in the map/analyses?). In figure 1 what does the stripped area refer to? A table with information would be useful with some additional information on the data used from the studies by Ma and Zhang and Wang et al. Also

the basins are referred to as the Yangtze/ Yellow river basin, but as far as I understand these only cover the upper part of the basin. Please provide some additional information on the extent of each of the basins analyses (eg provide name of the gauging station where the basin was delineated). Also in figure 3, there is a reference to TP, which basin/ area does this refer to (the entire TP area shown in figure 1 or the area of all the basins combined, which are two different areas)?

**Reply**: Sorry for the ambiguities in some of our illustrations and related information. We revised the figures and added more information accordingly. Overall, we used monthly $ET_{wb}$ from five basins from previous studies (Ma and Zhang., 2022; Wang et al. 2021), including the headwaters of Yellow basin (HYE), headwaters of Yangtze basin (HYA), upper Heihe basin (UH), Inner Tibet Plateau (INTP) and Qaidam (QDM) basins. It is true that these only cover the upper part of the basin, and we defined explicitly the extent of these regions and presented this information. A new table (Table 2 in the revised version) was added to provide additional information, i.e., the extent of the basins and the names of the gauging stations.

As regards Figure 3, we intended to use TP to represent the area of all the five basins combined. To avoid the potential misunderstanding, we revised it to *5 basins* (the area of all the basins combined) in the new version.

- Color scheme of figure 3 is not fully intuitive, for example the r2 is deep red for high (=good) values)

**Reply**: We revised the figure to make it more intuitive.

Figure 5: what do the different colors of the bars mean?

**Reply**: We added the description in Figure 5. The global satellite remote sensing-based ET dataset are shown in dark blue and model-based ET dataset in light blue, and the regional ET datasets are in red.

Trend analyses (figure 7):

- The calculation of the trends could be affected by an exceptional year with high or low ET at the beginning or end of the time series (since there is quite some yearly variation and the trends are often relatively minor). Could you say something about the significance of these trends as well? Also for the SynthesisET both the first two years and the last two years seem to be outliers and related to the "temporal inconsistencies" of the product. Was this data properly vetted before including in the analyses?

**Reply**: We fully agree with you that the trend could be affected by the exceptional years at the beginning or end of the time series. This is also why we choose a robust regression method to estimate the trend of ET, rather using simple linear regression, since the robust regression can reduce the impact of outliers. We added the significance level of the trends in the figure and main text.

As regards the temporal inconsistencies of SynthesisET, we carefully checked it for several times and we are pretty sure about the existence of the temporal inconstancies. In fact, this issue was also

noticed by the authors of the SynthesisET dataset, and they tried a different synthesis strategy d in a later regional study on the Northern China (Wang et al., 2021). This could also be seen from the temporal variation SynthesisET in Figure RC1-R1. Figure RC1-R1 was used to replace the Figure 7 in the manuscript.

[Figure]

Figure RC1-R1: Yearly variation of ET in the TP by different products. The inset panel shows the annual ET trend by different products. *: trend with significance level (p<0.05).   In the top panel, the reanalysis data is shown as a dotted line, and the land surface model-based data is shown as a dashed line.

We also checked the spatial variation of ET by SynthesisET (as shown in the following Figures RC1-R2). Before 2000, SynthesisET showed quite high ET values (e.g., in the eastern TP), while after 2019 SynthesisET showed extremely low ET values in the eastern TP.

[Figure]

SynthesisET, July, 2000              SynthesisET, July, 2008

[Figure]

SynthesisET, July, 2013                    SynthesisET, July, 2019

Figure RC1-R2: Example of spatial variation of ET by SynthesisET in July of different years.

Reference:

Wang, L.;Wu, B.; Elnashar, A.; Zeng, H.; Zhu,W.; Yan, N. Synthesizing a Regional Territorial Evapotranspiration Dataset for Northern China. Remote Sens. 2021, 13, 1076. https://doi.org/10.3390/rs13061076.

- Why are many of the products with longer time series (eg ERA5Land, SynthesisET, BESS, MERRA2) not presented with their full timeseries?

**Reply**: The ERA5-Land ET shows a very similar trend as ERA5. As regards SynthesisET, we already noticed its temporal inconsistence, thus we did not include it in the annual trend analysis. To reduce the concerns of reviewers, we included all the long-term ET products in the revised version.

Analyses of "ET components"

- As mentioned by the authors these different sub-components of ET are not validated and with the wide range of values derived from the different products, what conclusions can really be drawn? This is especially a question for the open water ET (maps in figure 9 shows large areas evaporating from water surfaces) and sublimation (which is validated how?)

**Reply**: It is true that the evaluation of different ET components was still limited due to the scarcity of available data and a comprehensive evaluation based on more observations would help to further evaluate the ET components and improve the algorithm performance. This analysis on the ET components has not been fully investigated in previous studies. We intended to use it to explain the difference among ET products and to answer the question: which processes play a significant role in determining the total ET. We also noticed that previous studies mostly focus on total ET, e.g., magnitude, spatial variation, temporal trend, etc., while the ET components were not fully investigated. Meanwhile, many studies were based on a big-leaf model, and a few studies estimate total ET based on the separate estimation of ET components. These components reflect the different water phase change processes that are regulated by different factors, e.g., transpiration is mainly controlled by the plant physiology through the regulation of stomata behavior, soil evaporation is determined by heat and mass transfer in the top soil with liquid water present at some depth below the surface, the rainfall interception loss is mainly related to the canopy morphology and rainfall intensity and the sublimation is associated with higher enthalpy change than vaporization process

and near surface air humidity and temperature. So, we believe this analysis on the ET components is helpful, because at least starts with treating correctly each water phase change.

It is important to note that reliable independent reference measurements on each component of total vapour flux are very scarce. The anonymous Referee #3 (RC3) suggest us to use the ensemble mean of the ET components by different products, which may be close to the truth. We also notice that averaging properly would not provide good estimates, since the it applies only to random errors, not to the use of the wrong algorithm. According to the results in Section 3.2.3, the median values of the ratio of Es, Ec, and Ei to total ET is 50%, 30%, and 5%. A recent study shows the contributions of Es, Ec, and Ei to total ET are 68.21 %, 23.57 %, and 8.21 %, respectively in the Three Rivers Source of the Tibetan Plateau (Zhuang et al., 2024), which is actually quite close to our estimates. After the analysis in our study, we may generally conclude that soil evaporation (Es) contributes most to total ET in the whole TP, and further study should pay more attention to it. We also noticed that the phase change of snowfall is poorly known. These events are short but widespread in the TP, with snow-cover being extensive but short-lived. Both snow melt followed by evaporation and infiltration and sublimation are relevant and will be investigated.

Reference:

Zhuang, J., Li, Y, Bai, P, Chen, L, Guo, X., Xing, Y., Feng, A, Yu, W., Huang, M.: Changed evapotranspiration and its components induced by greening vegetation in the Three Rivers Source of the Tibetan Plateau. J. Hydrol., 633, 130970, https://doi.org/10.1016/j.jhydrol.2024.130970, 2024.

Analyses related to the "response to different environmental factors"

- The purpose of these analyses are not entirely clear to me. First, the analyses are done for the median value of the correlation, whereas it was already very clear that there is a large variance between the different products. Also several products utilize these input data (Rn, LAI, P) for estimating ET, how is this kind of dependency considered in the analyses? Do different types of models have stronger or weaker correlation with these environmental factors? And what does that mean for the interpretation of the analyses?

**Reply**: Thank you very much for the comments. Analyzing the impact of environmental factors on ET is helpful to reveal the governing factors and the mechanisms determining the variability of ET. It is also helpful to analyze whether and how the ET algorithms/product capture the ET variation caused by the environmental factors. It is true that different models have stronger or weaker correlation with these environmental factors, which indicate the observed response to forcing factors is algorithm dependent. Several products utilize these input data (Rn, LAI, P) for estimating ET, and these products may show higher correlation with these factors. Hence, we think both the algorithm itself and the input data can impact the response of estimated ET to environmental factors.

We also noticed that the current analysis is very limited and a more comprehensive analysis could be done to illustrate this issue better. A proper treatment would require a significant amount of additional materials and we decided to leave it out for the time being. Hence, we removed it from

current manuscript and prepare another paper on it for a more robust analysis following the suggestion of Referee #2 (RC2).

- Did any of these factors also influence the partitioning of ET into ETc and ETs?

**Reply**: We did not mention this issue in the manuscript. But, we think the answer is yes. This is especially true for leaf area index. Higher leaf area index is generally associated with higher plant transpiration and interception loss. For example, a recent study shows that the vegetation greening (judged by increasing LAI by 0.009 $m^2/(m^2\ a)$ with $p < 0.05$) caused different changes in ET and its components, i.e., 1.95 mm/a, $-2.41$ mm/a, 1.33 mm/a, and 3.03 mm/a for ET, Es, Ec, and Ei, respectively, in the Three Rivers Source of the Tibetan Plateau (Zhuang et al., 2024), which clearly indicates its influence on the ET partitioning.

Reference:
Zhuang, J., Li, Y., Bai, P., Chen, C., Guo, X., Xing Y., Feng A., Yu W., Huang, M.: Changed evapotranspiration and its components induced by greening vegetation in the Three Rivers Source of the Tibetan Plateau. Journal of Hydrology, 633, https://doi.org/10.1016/j.jhydrol.2024.130970, 2024.

Discussion:

- General reflection of the validation methods employed, doesn't really add much information. The incorporation of seasonal land cover conditions or lack thereof is only explained for 3 products, but then no reflection on how that has affected the results. Or how relevant negative latent heat fluxes are (does this happen often or only occasionally?). The reflection on the water balance estimations are also very general and could have been included in the introduction (there is no reflection based on this specific study). For example, the assumption of not incorporating meltwater could have been explained in the method but is not an outcome of this research.

**Reply**: We understand the reviewer' concern. We moved some general comments to the introduction and revised the discussion section to focus more on the findings of the current study as follows:

The in-situ observations with an eddy covariance system are recognized as the standard method for monitoring energy and mass fluxes to validate high-resolution ET (Baldocchi, 2020). In addition, the ET products were compared with the basin-scale water balance estimates $ET_{wb}$. $ET_{wb}$ is obtained at the basin scale (several hundred $km^2$), which is much larger than the footprint of flux tower observations (approximately $km^2$, depending on meteorological conditions). Given the relatively sparse distribution and small footprint of the flux-tower-based eddy covariance system observations, the water balance method can serve as a useful complementary reference for ET estimates. This is especially true for the coarse-resolution ET, which has a much larger spatial footprints than eddy covariance observations.

In this study, these two methods gave generally consistent results when evaluating the high-resolution ET. When judged by the KGE of site-scale estimates, the accuracy of the high-resolution ET products can be ranked as follows: PMLV2 > ETMonitor > MOD16-STM > GLASS > MOD16 > SynthesisET > SSEBop. When judged by the KGE of basin-scale validation, the accuracy of the high-resolution ET products can be ranked as: ETMonitor > PMLV2 > MOD16STM > SSEBop >

GLASS > MOD16 > SynthesisET. Although both indicate that ETMonitor, PMLV2, and MOD16STM are the most accurate and the remaining four are less accurate among the high-resolution ET products, some differences in the ranking of the ET products can be observed. This is probably related to the processes captured by the 'ground-truth' data at different scale used in the two evaluation methods. An eddy covariance observation represents the net water vapour flux integrated across different processes at given point (e.g., plant transpiration in the dense vegetation regions, snow sublimation in dry snow cover regions, evaporation of canopy-intercepted water when the canopy is wet due to intercepted rainfall). In addition, the observed vaporization process depends on the land surface conditions at the observation sites during particular times, which may vary seasonally and annually due to factors such as snow/ice, intercepted water, and vegetation. The estimated basin-scale ET by water balance ($ET_{wb}$) was essentially the residual of the observed water balance terms, which is assumed to be the net liquid water flux loss to the atmosphere at the basin scale. Compared to the site-scale observation, the basin-scale $ET_{wb}$ can capture the effect of land cover dynamic on the ET within the basin. For example, the mean water level of lakes in TP increased by 0.20 m/yr from 2000 to 2009, and the lake water mass increased significantly (Zhang et al., 2013), which caused higher ET in the TP because water evaporation is generally higher than other land cover types. However, most ET products (e.g., MOD16, PMLV2, etc.) assume constant land surface conditions throughout the year or multiple years, which means that they cannot capture the temporal transitions of the vaporization process associated with changes in land cover. In contrast, ETMonitor adjusts the daily land cover based on dynamic land cover conditions, including water bodies cover and snow/ice cover, which allows it to reflect the impact of seasonal and annual open water extent and snow/ice cover on total ET (Zheng et al., 2022). This probably explains in part why ETMonitor performs slightly better than PMLV2 when validated by basin-scale water balance methods, while they are comparable when validated by in-situ observations.

Reference:

Liu, H, Xin, X, Su, Z., Zeng, Y., Lian, T., Li, L., Shanshan S.: Hailong Zhang Intercomparison and evaluation of ten global ET products at site and basin scales. J. Hydrol., 617, 128887, https://doi.org/10.1016/j.jhydrol.2022.128887, 2023.

Zhang, G., Yao, T., Xie, H., Kang, S., and Lei, Y.: Increased mass over the Tibetan Plateau: From lakes or glaciers?, Geophys. Res. Lett., 40, https://doi.org/10.1002/grl.50462, 2013.

Zheng, C., Jia, L., and Hu, G.: Global land surface evapotranspiration monitoring by ETMonitor model driven by multi-source satellite earth observations, J. Hydrol., 613, 128444, https://doi.org/10.1016/j.jhydrol.2022.128444, 2022.

Baldocchi, D. D.: How eddy covariance flux measurements have contributed to our understanding of Global Change Biology, https://doi.org/10.1111/gcb.14807, 2020.

Chen, X. Yuan, L., Ma, Y., Chen, D., Su, Z., Cao., D.: A doubled increasing trend of evapotranspiration on the Tibetan Plateau. Sci. Bull., https://doi.org/10.1016/j.scib.2024.03.046, 2024.

- The discussion related to the different types of models comes a bit out of the blue, for example in table 2 the model type is not provided, which makes is difficult to validate a statement such as (first sentence) " PM-type model demonstrated superior accuracy compared to other models". Also ".. models that incorporate soil moisture to detect water stress…" can not be checked, which

models do or do not incorporate soil moisture? Also to go in depth into the methodology of each product seems to go beyond the objective of this research, especially since it unclear why some models are singled out and others not (nor a statistical comparison between for example PM vs non-PM models is not done.

**Reply**: We have double checked and revised the manuscript to make sure all the necessary information is included and the statements can be easily checked. We already stated in the manuscript that "Among the evaluated ET products, there are 14 products that primarily use remote sensing products, including 2 products (SSEBop and EB) based on land surface temperature (LST), 8 products (ETMonitor, MOD16, MOD16-STM, PMLV2, PMLV2-Tibet, GLEAMv35a, GLEAMv35b, BESSv2) based on PM-types models (including Penman-Monteith equation, Priestley-Taylor equation, Shuttleworth-Wallace equation), 4 products (FLUXCOM-RS, FLUXCOM-RS-METEO, GLASS, SynthesisET) based on data-driven methods (machine learning method or ET products ensemble method)." To make the information more intuitive, we moved it to the Section 2.2.2. More information on whether soil moisture is considered in a given data product was added in Table 2 by listing the main forcing data.

Our primary objective is to find out how accurate are the ET products in the TP, which is closely related to the algorithm applied in each product. Since we evaluate 22 products, there are 22 models to be discussed, which is actually too much and will make the manuscript unfocused. Therefore, we discussed the methodology of some representative ET products. Some evidence on the difference between the PM and non-PM model can be found in Section 3.1.1, which shows that the best three products are all PM -type model-based products (ETMonitor, PMLV2, MOD16STM), while the LST-base (SSEBop) and data-driven products (GLASS and SynthesisET) had overall a low accuracy. We revised the manuscript to present this statement more clearly in the revised version.

- The uncertainty of the SynthesisET product was already mentioned in the results section, is this really an important outcome of this research (important enough to single it out in the discussion?)

**Reply**: Thank you for the comments. In the results section, we evaluated its accuracy and compared with other products to identify a temporal inconsistence. In the discussion section, we try to explain the reason of its relatively poor performance, since we expected the fusion of different datasets should have improved the overall accuracy. We addressed the importance of the ensemble method in the discussion, which might be helpful to guide further studies.

**2. Reply to the comments by REVIEWER#2**

The article "How much water vapor does the Tibetan Plateau (TP) release into the atmosphere?" by Zheng et al. provides a comparative analysis of evapotranspiration (ET) on the Tibetan Plateau, an essential yet uncertain component of the water cycle. This comprehensive review examines various streams of ET data and compares them with in-situ flux measurements, aiming to address a significant research gap: Can ET estimates derived from satellite and land surface models accurately reflect in-situ ET observations?

While I appreciate the insights offered by this article, particularly its thorough incorporation of diverse data sources, there are concerns regarding the clarity and completeness of the methodology. Consequently, the obtained results lack sufficient substantiation. Therefore, before publication, these concerns need to be addressed thoroughly.

**Reply**: We thank you for the positive and constructive feedback that helps us to improve our work.

**Major comments:**

**Regarding the Methodology:**

1.  The temporal coverage of the analysis is not clearly defined throughout the article. In line 197, it is written 2003 to 2015, in line 221, 2001 to 2018 while in line 312, it is written 2000 to 2020. These discrepancies need clarification to ensure consistency and accuracy in the reporting of the study period. I suggest keeping the results with consistent temporal coverage in the main section while any other information on supplementary information (SI).

**Reply**: Thank you very much for the suggestion, and we try to use consistent temporal coverage in the main section. The differences in the overlap period were caused by differences in the temporal coverage of ET products and *in situ* observations. In section 3.1.1 on the validation of ET products against flux tower measurements, the overlap period for *in situ* eddy covariance observations and high-resolution ET products was in most cases from 2001 to 2018, but there are differences for some sites and products. We added Table S1 in the supplementary materials to show the temporal coverage for each site and each product for site-scale validation. In section 3.1.2, when the validation of ET products against basin-scale water balance $ET_{wb}$ is presented the temporal coverage was from 2001 to 2015 with some gaps for some catchments and data products. We also added a table (Table 2) to show the temporal coverage period and integrating the information related to basin-scale validation where necessary. In section 3.2.2, '2000 to 2020' is not the precise temporal coverage of different products. Instead, we determined the median value of ET of all available products for each year between 2000 and 2020. The median value of ET was further used to obtain the overall trend of ET from 2000 to 2020. We revised this part to make it clearer.

Furthermore, to avoid unnecessary confusion caused by differences in the temporal coverage, we try to keep a consistent temporal coverage and add information in each section where necessary. Considering the temporal coverage of all products is from 2003 to 2013, the comparative analysis in Section 2.3.2 and Section 3.2 were conducted by applying the period 2003~2013, unless gaps in data had to be taken into account, leading to a different temporal coverage.

2.  Although it appears to be conducted at a monthly scale based on the information provided, it is unclear whether all datasets, such as ETMonitor with daily resolution and MOD16 with 8-daily resolution, were aggregated to a monthly scale for comparison or were based on the native resolution of the dataset. Clarity is needed regarding the aggregation process of these datasets to ensure transparency and understanding of the methodology employed.

**Reply**: It is true that the validation was carried out using monthly data. All the products were temporally aggregated to monthly values from their native temporal resolutions prior to validation

and comparison. The data products with a daily resolution were just added up to obtain the monthly ET values. For the data with 8-days resolution, an average ET value was first estimated for the available data in that month, and the monthly ET value was subsequently obtained by multiplying the averaged values by the number of days in the month. We added this description to clarify how the monthly data were obtained.

3. In line 135, it is mentioned that months with less than 50% valid daily ET values were excluded from the analysis. However, it remains unclear whether these excluded months were filled to maintain a continuous ET time series or if the comparison was limited to months with more than 50% valid ET values. Clarification on how the missing data was handled and its impact on the analysis is necessary for a comprehensive understanding of the methodology.

**Reply**: The missing data was not further filled and gaps were excluded to avoid the impact of uncertainty introduced by gap-filling. We stated this in the methodology.

4. Providing information on the number of valid observations available for each dataset, either in the supplementary information or elsewhere, would be beneficial for assessing the comparability of sample sizes across datasets, especially if they are not analyzed for same temporal coverage.

**Reply**: Thank you very much. We added a table in the supplementary materials to include the temporal coverage for site-scale validation and number of valid observations.

**Regarding the results:**

1. It appears that the regional-based formulations of ET, such as MOD16STM and PLMV2 ET Tibet, demonstrated the highest accuracy when compared to in-situ flux towers. However, it is crucial to ensure that the flux stations utilized in this study for comparison were not already included in the calibration of these datasets. If the same flux stations were used for calibration, the greater accuracy of these products may not be fully substantiated. Therefore, it is imperative to verify whether there is any overlap between the flux stations utilized in this study and those used for calibration to accurately assess the reliability of the results.

**Reply**: The issue you mention is very important, and we agree with you that the validation results are influenced by the calibration. As a summary, the calibration sites were clearly listed for three high-resolution products (ETMonitor, PMLV2, MOD16STM), and some of these sites were used for validation in this study. Other products did not use flux sites for calibration or this information is not presented in the corresponding studies. Although some coarse-resolution products (e.g., PMLV2-Tibet) were also reported to use flux sites as calibration, they were not validated based on flux site observations in this study, considering the mismatch of spatial representativeness between in-situ observations and coarse-resolution products. In this study, we did not exclude the calibration sites in our validation study for the following reasons:

● The difficulty in maintaining ground-based observations have resulted in a scarcity of flux towers on the TP. If calibration sites were excluded, the validation sites would be scarce, which would raise further concern on the sites' representativeness and relevant uncertainty.

- Different products use different sites for calibration, and some studies did not provide such information. Some products were designed with a clear separation between calibration and validation sites, while others did not. For example, some studies clearly separated calibration and validation samples using data of different years from same sites, while other studies did not provide clear information at all. It seems to be not feasible in practice to apply a well-defined screening of calibration and validation data.
- To achieve high accuracy, model calibration is a valid approach applied for many models before generating datasets. The purpose of this study is to identify how accurate the current ET products are, which might help to achieve an ET product with better accuracy, and efforts on model calibration should be encouraged.

To address this, we included the information on whether the sites were utilized for each ET product calibration in the supplementary materials Tables S1. In addition, we performed basin-scale validation to strengthen our findings. To our best knowledge, there is no product using basin-scale water balance estimates for calibration, i.e. this approach as an independent validation method.
* * *
2. In Figure 3, it is unclear how the metrics were calculated for the entire Tibetan Plateau (TP). Does the metrics for TP represent averages or medians across the basins or was TP treated as a single basin?

**Reply**: Sorry for the misunderstanding. We intended to use *TP* to represent the area of all the five basins combined, including headwaters of Yellow basin (HYE), headwaters of Yangtze basin (HYA), upper Heihe basin (UH), Inner Tibet Plateau (INTP) and Qaidam (QDM) basins. To avoid any misinterpretation, we revised it to *5basins* (the area of all the basins combined) in the revised manuscript. A new table was added to provide additional information in the Supplementary., We simply used all samples (each sample represents a valid group of reference data and to-be-validated ET data from one basin) from all 5 basins to estimate the metrics for the *5basins*.
* * *
3. In Figure 4, the color bar for ET standard deviation (ETsd) differs from the color bars used for other variables. This inconsistency can lead to confusion, particularly since the figures are presented together. Also, if possible, please keep the results in the order of datasets that appears in the Table 1.

**Reply**: Thank you for the suggestion. We revised the figure accordingly to avoid the confusion. We also moved the information on the spatial variability of ET in each product to the supplementary materials to make the manuscript more concise.
* * *
4. Regarding Figure 7, it would be beneficial to highlight the trends observed specifically from data with long records to discern the presence of significant trends in ET, because the trend calculated with only some years of data would not add any conclusion to the overall trends in the ET. Additionally, it's essential to clarify how the trends were calculated—whether through linear regression or another method—and whether the significance of these trends was assessed.

**Reply**: Thank you for the suggestion. We also noticed that the trend could be affected by the temporal coverage of the ET time series, and we also agree that longer records provide more reliable information on trends. We identified the trends estimated with long records in the revised version.

We believe that relatively shorter data records (especially in recent years) remain relevant to document differences across data products, so we kept the results on trends after 2000s.

We applied a robust regression method to estimate the trends, rather than using simple linear regression, since the robust regression reduces the impact of outliers. We added the significance level of the trends in the figure and main text.

**On results specific to "Response of the ET to main governing factor."**

The author's intended message or purpose behind the analysis is not clearly conveyed. It seems to explore the relationship between annual ET and various water, energy, and vegetation variables. I will try to highlight my concerns in points here:

1. In my belief, the analysis of how annual ET responds to different water, energy, and vegetation variables could potentially be a separate study requiring a more comprehensive approach.

**Reply**: Thank you very much for the suggestion. We agree with you that the response of ET to water, energy, and vegetation variables could be a topic to be addressed by a more comprehensive analysis. We removed it from current manuscript and prepare another paper on it for a more robust analysis.

2. For instance, If Leaf Area Index (LAI) correlates well with both/or net radiation (Rn) and precipitation (P), which I believe will be the case, raises doubts about the conclusions drawn regarding the relative influence of these variables on evapotranspiration (ET). This is true especially when conclusion on influence of these variables on ET is drawn simply from correlation of ET with these variables without controlling for the other confounding factors. To check whether this is the case or not, we can simply correlate LAI with Rn and P, as well as by correlating Rn with P.

**Reply**: It is true that LAI correlates with both Rn and P. LAI is a critical variable that correlates with several climatic and environmental factors as it represents the amount of leaf area per unit ground area and characterizes the canopy structure. LAI influences the interception of radiation and the distribution of light within the canopy, which in turn affects the energy balance of the surface, e.g., net radiation (Rn) and latent heat flux (LE). Also, plant generally grow better in regions with sufficient water supply (high precipitation) and adequate APAR (highly related to Rn).

3. Even if one were to accept the current analysis, which I personally disagree with for the reasons outlined in points 2, there remains a crucial need for clarification regarding the rationale behind correlating median ET from all datasets (if I understood it properly) with environmental variables (Figure 10). This need arises primarily from the significant variability observed among different ET datasets in terms of magnitude and hence I believe that the relative importance assessed from the simple correlation of ET with these variables will also vary. Consequently, any conclusions drawn from these correlations may lack robustness.

**Reply**: We agree that there could be issues with determining the independent effects of these variables on ET if only simple correlations were used. Correlation does not imply causality and more sophisticated statistical methods need to be used, e.g., multiple regression analysis, to control

for confounding factors and to determine the relative influence of Rn and P on ET while considering LAI. This would allow to estimate the unique variance explained by each predictor while holding the others constant.

Furthermore, it is important to consider that the relationships between these variables can be complex and non-linear, and they might be influenced by other factors such as soil moisture, air temperature, humidity, wind speed, and atmospheric pressure. To accurately assess the relationships and the potential for misinterpretation, in our next study we will try to employ a multivariate analysis approach to establish the unique contributions of Rn, P, and other factors on ET, while controlling for the potential influence of other relevant factors in another study based on a more robust analysis.

4. Again, in regions where Ec and Ei are the dominant modes of evapotranspiration (Figure 8), it would be valuable to investigate their correlation of ET with LAI compared to Rn and P, after removing the confounding effects.

**Reply**: It is true that the above-mentioned issue for ET is also applicable for Ec and Ei, and the multivariate analysis can be applied to investigate the response of Ec and Ei to environmental factors.

5. Nevertheless, I still believe this could be separate research with robust approach.

**Reply**: Thank you very much for the suggestion again. We removed this part accordingly and will produce another manuscript based on a more robust analysis.

**Additional technical comments:**

1. Before highlighting the monthly RMSE, it would be helpful to provide information on the magnitude of monthly ET observed at different flux stations based on in-situ observations. This would allow for a comparison of the magnitude of observed ET with the error represented by the RMSE.

**Reply**: Thank you very much for the suggestion again, we added a table in Supplementary (Table S1) with the mean value of observed ET.

2. It's advisable to maintain analysis with consistent spatial and temporal coverage in the main section, while keeping analyses involving datasets with inconsistent coverage to the supplementary section. This will enhance clarity of the manuscript.

**Reply**: Thank you very much for the suggestion again, and we focus on the products with spatial and temporal continuity for the analysis in the main text to retain the results and analysis with the same spatial and temporal coverages.

3. In Figure 8, it is noted that while the total evapotranspiration (ET) may appear similar across different datasets, the partitioning of ET between datasets is not consistent. This observation is indeed a significant finding. However, the substantial explanation provided does not sufficiently clarify why the datasets differ so much, particularly for GLDAS and MERRA2.

**Reply**: Thank you for appreciating our findings. The partitioning of ET into its components, such as evaporation from the soil (Es) and transpiration from plants (Ec), can vary significantly among

[revised manuscript text omitted]

**Other comments:**

Overall, there are numerous instances in the text which exhibits repetition and with typos, with numerous lines conveying similar information and occasionally out of context. Therefore, significant restructuring of the article's text is necessary.

**Reply**: We apologize for the repetition and typos. We went through the manuscript again to improve it.
* * *
**For instances:**

1. The passage from lines 60-65 highlights the significant uncertainty surrounding evapotranspiration (ET) estimation on the Tibetan Plateau (TP). However, the paragraph falls short in effectively conveying how the present research differs from existing literature. It is evident that this study introduces novelty to the field, particularly through its comprehensive comparison of various ET products with in-situ observations in TP. This contribution warrants greater emphasis in the introduction section.

**Reply**: Thank you. We emphasized the novelty in the introduction section in the revised manuscript. Previous validations were generally based on either in-situ measurement by the eddy covariance system or the basin-scale ET estimated by water balance method, which represent the surface net water flux at different scales, while these ET products mainly focus on the upward water vapour flux. Recently, Chen et al. (2024) evaluated several ET products with spatial resolutions ranging from 1km to 50km against site-scale eddy covariance observations. It is important to note that the observations from tower-based eddy covariance systems have a very small footprint (approximately several hundred metres depending on weather conditions), and direct comparison of site-scale observations with the coarse-resolution ET products (e.g., 25km) is problematic due to the severe problem of spatial mismatch. In order to increase the credibility of currently available ET products, this study will undertake a more comprehensive evaluation, taking into account both in-situ observations and basin-scale measurements.


**Reply**: We moved the classifications in the discussion sections to the Section 2.2.2 ET Products when they are first introduced. We also revised the results section to emphasize the discussions relevant to Section 4.1.

**On the introduction section, it appears:**

These validations were generally based on either in-situ measurement by the eddy covariance system or the basin-scale ET estimated by water balance method, which represent the surface net water flux that integrates different processes (e.g., plant transpiration for the dense vegetation regions, snow sublimation for the dry snow cover periods for the eddy covariance system observations, even condensation when negative latent heat flux occurs),while these ET products mainly focus on the ET (positive upward latent heat flux), which attributes to the validation uncertainty.

*While in the section 4.1.1, it appears:*
The eddy covariance system observation represents the net water flux integrated across different processes (e.g., plant transpiration                in the dense vegetation regions, snow sublimation during the dry snow cover periods, evaporation of canopy-intercepted water when                the canopy is wet due to intercepted rainfall). The vaporization process observed by the eddy covariance system depends on the land                surface condition, which may vary seasonally and yearly due to factors such as snow/ice, intercepted water, and vegetation. Meanwhile, eddy covariance system observation includes condensation when negative latent heat flux occurs. Remote sensing-based ET products mainly focus on positive ET (positive upward latent heat flux) and omit processes such as condensation.

These two instances basically convey same information. I do agree this is important point to make reader aware about the validation. However, I think the author could be concise about it and avoid unnecessary repetitions.

**Reply**: In the introduction, we intended to introduce generically the uncertainty caused by the validation method, while in the discussion we focused on the processes captured by tower-based observations, as documented by our findings. To avoid the repetitions, we revised the introduction and discussions accordingly.

4. In line 100, it might be more appropriate to adhere to existing climatic regime classifications, such as those based on AI or other established frameworks. Because the term rather "monsoon" is kept here in between arid and humid climate types. So, how "different" is "monsoon" from the humid in these classifications? Or what does that monsoon mean when compared with "arid" and "humid"?

**Reply**: We agree with you that it is more appropriate to use the existing climatic regime classification and monsoon is not a standard climate type. According to the Köppen classification, there are dry, subtropical, temperate, subpolar and polar climate types in the TP. These climate types are influenced by both westerlies and the Asian monsoon, which is also enhanced by the thermal forcing of the TP (Zhou et al., 2009; Wu et al., 2012; Yang et al., 2014). The aridity index (P/PET) or Budyko dryness ratio (PET/P) are also widely utilized to characterize the aridity level. A recent study has shown that the dryness ratio has a large spatial variability in the TP, from humid climate with dryness ratio less than 0.3 to hyper-arid climate with dryness ratio larger than 3 (Feng et al., 2024). We revised the description accordingly to avoid any ambiguity.


Wang, B.*, Y. Ma*, Z. Su, Y. Wang and W. Ma. Quantifying the evaporation amounts of 75 high-elevation large dimictic lakes on the Tibetan Plateau. Science Advances, 2020, 6, eaay8558.

I agree to the author that they have collected more ET products in this study, but the generally conclusions are not really new compared with previous ET studies on the TP. Hereby, I suggest to focus more on ET components verification and their trends. This part has not been fully investigated by previous publications. The ET trends and annual ET estimation does not deserve more energy on it. This means that the title should be also changed. There are also some water balance ET studies. Hereby, this analysis is also not new. Introduction should really have a in depth review of previous work.

**Reply**: Thank you for providing the latest publications and constructive suggestions. It is true that ET components are important and not well studied, however we think clarify the total ET and ET trends is also helpful, especially considering that differences in ET components can surely lead to different total ET. Although the previous studies by Chen et al. (2024) and Yuan et al. (2024) have demonstrated the difference of area-averaged ET in the TP, they did not investigate the spatial variability of this difference which actually is very large. Furthermore, previous studies on ET mostly applied the old TP boundary, which only includes the region inside China. Recent studies emphasized the geographic integrity of the TP and a new boundary of TP was applied (Zhang et al., 2013; Zhang et al., 2021), which is larger than the area of the old boundary by 20%. This boundary is more reliable as it is based on geomorphology and formation processes that considers factors such as elevation and watershed boundaries. Hence, the comparison of ET amount and trend is still necessary. We strengthened the materials on ET components in the revised version following the suggestion.


**Reply**: We revised it accordingly.
* * *
Line 80, these specificities, are you talking about negative latent heat? If yes, please use negative latent heat directly.

**Reply**: We revised it accordingly.
* * *
Line 81, How accurate are these improved ET products, I understand that this question is already answered at least partly in Chen and Yuan`s publication. The snow/ice sublimation is new in this study. I suggest to revise the second question to: which processes play a significant role to the ET components trend. The third question, I did not find the author provide answers to which factor dominant different ET products. Hereby, the introduction should be rewritten and new scientific questions to be raised. Current formation is quite weak and not comprehensive.

**Reply**: It is true that previous studies did some evaluation already of new data products. However, it should be noticed that this validation was conducted only at site scale by comparison of eddy covariance observations and ET products. The tower-based eddy covariance observations have a very small footprint (roughly several hundred square meters depending on the weather conditions), and direct comparison of site-scale observations with the coarse-resolution ET product (e.g., 25km), suffers severe problem of spatial mismatch. Hence, we only used site-scale observations to validate the high-resolution ET products ($\sim$ 1 km$^2$). We used basin-scale $ET_{wb}$ to validate both high-resolution and low-resolution ET product. In this sense, our comparison is more robust and comprehensive. To address this aspect, we included this point in our revised version.

For the second question, we intend to figure out how much water is vaporized in TP and which processes (e.g., plant transpiration, soil evaporation, snow/ice sublimation) dominant the total ET. The different components here correspond to different bio-geophysical processes, and this is addressed in Section 4.3 in the revised version. The plan transpiration from plant leaves is mainly controlled by the stomata behaviour in response to environmental conditions, soil evaporation is controlled by soil structure and soil water content, the rainfall interception is determined by canopy morphology and rainfall intensity, and vapour transport after sublimation is determined by near surface boundary layer conditions and the higher latent heat of sublimation. As regards the third question, we removed the section on the response analysis following the suggestion of RC#2.

The first aim of this paper is already investigated by Chen et al. Please change this point or further deep this aim. Actually, there are many attribution studies of TP ET trend. Please review their studies, then make a revision for the num 3 aim.
**Reply**: The first point is addressed in our reply to the previous comment above.

Line 94, I don`t really agree that pearson correlation analysis can provide us the response of ET to precipitation, Rn and LAI. Indeed, I don`t suggest to include this correlation analysis in this paper. These analysis weaken this paper, it does not benefit to this work.
**Reply**: We agree with you and removed it in the revised version.

Lines 122, These the sites, please correct this error.
**Reply**: We revised it accordingly.

Table 2, EB is a daily ET product, not monthly.
**Reply**: We revised it accordingly.

Figure 4, SEBS should be EB?
**Reply**: We revised it accordingly.

Please revise 'in Tibetan Plateau' to be 'in the Tibetan Plateau' or 'in the TP'.
**Reply**: We revised it accordingly.

Figure 5, the figure caption should explain what is meaning for different colored bars.
**Reply**: The global satellite remote sensing-based ET datasets are in dark blue, and the land surface model-based and analysis global ET dataset are in light blue, while the regional ET datasets are in red. We added this explanation in the revised version.

Figure 7, it is quite difficulty to recognize which bar represent which product. Add the product name corresponding each would be more useful. All the trends are ended in 2020? Their curves in figure 7 do not exhibit the same end year.
**Reply**: We revised Figure 7 to include the products' name. It is true that different products end in different years, and the end year for the trend analysis depend on the end year of the products. This information was added in the revised version.

Line 322, Among these products, there are nine that provide the main components of ET (Ec, Es, and Ei), it is better to directly say that 'Nine products provide …'.
**Reply**: We revised it accordingly.

It is important to note that there is no independent reference available for the ET components. I suggest to use the ensemble mean of ET components to check their differences with the ensemble mean. Nine products have provided the ET components. It's a lot. Their ensemble may be close to the truth.

**Reply**: Thank you for the suggestion. We also notice that averaging properly would not provide good estimates, since the it applies only to random errors, not to the use of the wrong algorithm. According to the results in Section 3.2.3, the median values of the ratio of Ec, Es, and Ei to total ET was 50%, 30%, and 5%. The ET partitioning ratios, i.e. 52%, 43%, and 5% by ETMonitor are the closest ones to the median values. A recent study shows the contributions of Es, Ec, and Ei to total ET are 68.21 %, 23.57 %, and 8.21 %, respectively in the Three Rivers Source of the Tibetan Plateau (Zhuang et al., 2024), which is actually quite close to our estimates.

Furthermore, we discussed the likely reasons causing such differences and the reliability of the partitioning results among different products in the discussion. For example, there are already reports that the overestimation of the Ec/ET ratio by GLDAS-VIC and GLEAM is due to the "big leaf" vegetation scheme assumption that there are no canopy gaps or exposed soil between plants, so soil evaporation only occurs in unvegetated areas (Bohn and Vivoni 2016; Sun et al., 2021; Miralles, et al., 2016). In contrast, GLDAS-CLSM tends to underestimate the Ec/ET ratio and to overestimate Es/ET, possibly due to the parameter problems related to the soil evaporation resistance or vegetation related resistance or the non-traditional approach to consider the subgrid heterogeneity of soil moisture (Feng et al., 2023; Sun et al., 2021). Therefore, to avoid the bias due to these already-known uncertainties, we removed these products in the calculation of the ensemble mean values.

Figure 8, the blue color around TP lakes may not reflect the truth. Please check if this is caused by a wrong lake mask.

**Reply**: We do not use a lake mask here. Please notice that here *Ew* represents the open water evaporation, which actually comes from either lakes or other water bodies, e.g., rivers, snow/ice melt water, flooded pixels, etc.
* * *
Figure 9, there are some reports about the annual ET amount for the TP lakes. Please cite these papers to verify Ew shown in the figure. I understand that Wang et al. Science Advance should also provide the Ew estimation for the TP. This study could benefit to verify the result in the figure.

**Reply**: We added some comparison with these results in the revised version. According to Wang et al. (2020), the total water evaporation is about $29.4 \pm 1.2$ km$^3$/yr ($\approx$1111.5 mm/yr) from the 75 lakes in the TP with total area of 26,450 km$^2$ (accounting for approximately 56.9% of the total lake area in the whole TP), and the total lake evaporation ($51.7 \pm 2.1$ km$^3$/yr) for all plateau lakes. The total open water evaporation amount from ETMonitor gives a value of 945.3mm/yr for the permanent water surface over the TP. The total water area is $1.29 \times 10^6$ km$^2$ in the TP when seasonal water bodies are taken into account, which is much larger than the permanent water surface. ETMonitor takes into account the seasonality of water surface areas when estimate ET, and the multi-year mean total annual water evaporation in the TP estimated by the ETMonitor is about at 44.4 km$^3$/yr, which is lower than that given by Wang et al. (2020).
* * *
Section 3.3, this part is not really persuasive. A simple correlation is not meaningful, in addition, other factors were not fully considered, such as air temperature, soil moisture, wind speed etc. In addition, the correlation of abnormal should be analyzed, not the original signal. I suggest to remove this section.

**Reply**: Thank you again, and we removed it accordingly.
* * *
"the daily land cover inputted" please revise this.

**Reply**: We revised it accordingly.

---

## Referee Report (RR1)

**How much water vapour does the Tibetan Plateau release into the atmosphere?**

The revised manuscript by Zheng et al. is streamlined and conveys the scientific message more clearly than the previous version. The authors compare different ET products regarding their usability in the Tibetan Plateau and find that high-resolution datasets perform better than others, despite significant discrepancies between models. In addition, models incorporating dynamic vegetation cover and water stress modules to predict ET outperform those without these features. Although land surface temperature-based remote sensing models generally perform worse, they tend to do well in arid regions. Additionally, regional parameterization may be key to improving ET products for specific regional applications. These findings add value to the existing knowledge about the strengths and weaknesses of ET products and their usability in the Tibetan plateau. Nevertheless, the paper is quite lengthy, and I believe it could be made more concise to enhance clarity and readability. I suggest making these revisions before publication and recommend this to the editor.

Please find some of my suggestions here:

1)      I believe it is necessary to include a land cover map in the supplementary section (omit if already added) and refer to it when justifying certain results, such as in lines 242-244, 282-286, and 291-293. Currently, land cover is introduced in Table 1, which does not align with the descriptions in the justifications, such as "densely" or "sparsely" vegetated, or "arid."

2)      For Section 3.2.2, "Temporal Variability in ET Across the TP," since a basin-wide analysis is provided, I believe more emphasis should be placed on basin-wide discrepancies. The following key points could be addressed in the text, but are not limited to:

      a) Please clarify whether the analysis was conducted for every product in each basin or if only the median of the products was analyzed across basins.

      b) Are there any variations in the profiles between the basins? If so, to what extent?

      c) How do the basins differ from each other in terms of land cover or other characteristics?

      d) Do all basins, regardless of land use, exhibit the same level of uncertainty?

3) Line 76-79: In my understanding, basin-scale ET estimates represent the net water vapor flux from land to atmosphere (whether positive or negative), as they are derived from the

water balance. Hence, they differ from insitu measurements, which capture both upward and downward water vapor flux. So, sentence needs restructuring.

4) Lines 85-89 and 90-94 convey the same meaning and can be merged for conciseness.

5) Figure 2: Please indicate overlap meaning consistent time frame, 2003-2013, it it's the case.

6) Line 279-280: Cross reference missing.

7) Figure 9: These differences, at least in sublimation, could be due to how snow extent is treated in each model. For instance, it seems that GLEAM considers snow extent differently than ETM, or could this simply be an effect of plotting?

8) Section 4.1 Title The paragraph does not align with the content, as it does not address which vaporization processes are more relevant. Instead, it focuses on the differences between processes captured in in-situ and basin-wide data, and how these differences impact ET comparisons.

9) Line 417-421: I find this quite interesting. Nice finding.

10) Line 437-439: Is not PM also an energy balance model?

11) Section 4.2.3 Title Since this section does not only incorporate ensemble ET products, please consider changing title here.

12) Section 4.3: Page 26: I believe this section could be condensed. While I enjoy reading this, it doesn't seem to align with the focus of the study and could be made concise without writing in such details. Alternatively, for each model used, a supplementary table could be created to detail differences in forcing, model structure, calibration, spatial heterogeneity, and other relevant factors and refer it.

13) Line 582- 588: I believe this resembles the previous version, as the information comes across abruptly to the reader. Since the related section has been removed in the current version, it now lacks the justification behind the statement.

Goodluck!

Prajwal

---

## Author Response (AR2)

Dear Editor and Reviewers,

Thank you for your editing and comments. We have revised further our manuscript according to your and the reviewers' comments. Please find some of the specific corrections and responses below. The reviewer comments appear in black and our responses appear in blue.

**I. Major changes:**

✓ We have condensed the discussion as suggested by the reviewers. The following paragraphs or sentences are deleted or revised: the first paragraphs in Section 4, the first and third paragraphs in Section 4.1, the third, fourth and fifth paragraphs in Section 4.3, the last paragraph in Section 4.4.1.

✓ We add numbering of the subfigures.

✓ The difference among the in-situ observations, water balance estimates and ET products in the fourth paragraph of the introduction section is revised to address the reviewer's concerns.

✓ The description of the vegetation condition and climate in Section 2.1 are revised.

✓ The land cover classification in Table 1 is updated.

✓ The statement in section 3.2.1 is revised to avoid potential conflicts.

✓ We have revised the supplementary materials by making some necessary edits and adding a new subfigure on the NDVI map of the Tibetan Plateau to illustrate the vegetation condition .

**II. Point-by-point response:**

**1. Reply to the comments by REVIEWER #1**

The manuscripts has undergone some substantial improvements, however there are still a few outstanding issues in my opinion which require revisiting. These include:

**Reply**: We thank you again for the review and the constructive feedback that helps us to improve our work.

- the use of the entire Synthesis ET dataset. It is very clear that this dataset has some major issues regarding consistency. Despite that you "carefully checked it for several times and we are pretty sure about the existence of the temporal inconstancies." doesnt mean you have to use the entire dataset in your analyses. The <2003 and >2018 data seems to be derived from different combinations of datasets (and overestimating <2003 and underestimating >2018). An example how problematic that is shown in figure 2 where the validation results for the Namco site (data available 2019-2020) is very bad for this data product. Authors are advised to carefully reconsider using the entire dataset. Also since none of the validation analyses cover the period before 2003 (used in WB analyse), including this period in the trend analyses may not add to the manuscript (this applies for all products - and therefore the trend analyses prior to 2003).

**Reply**: Thank you very much for the suggestion. It is true that SynthesisET has some issues and the validation cannot be done for the period before 2003. We also agree with you that SynthesisET data before 2003 and after 2018 should be removed from the analysis. For the inter-comparison analysis of different products (Section 3.2.1), we use the period (2003~2013) when all the products are

available. So, this result is not affected by the issue of SynthesisET. For the trend analysis, since our results are based on 22products, similar conclusions can be obtained as described in Section 3.2.2 even if we remove SynthesisET. We kept showing the SynthesisET data before 2003 and after 2018 in Figure7 to make its temporal inconsistence (extremely high value before 2003 and low value after 2018) visible.

- I am still not convinced about the discussion section which (my interpretation) tries to explain how the different categorised ET products are performing, however, the there is no quantification of how the different categorised ET products perform (as opposed to individual products), it is therefore difficult to validate the statements the authors make regarding the categorised products. In addition, the authors discuss in detail aspects such as parameterisations, model setup, input data etc, however this is not/can not be derived from the research presented. I would strongly advice to reduce the discussion towards aspects that can be supported by the analyses implemented and compared to other research in the field and not to venture into areas that are not substantiated by your research.

**Reply**: We agree with you to condense the discussion and focus on the issues supported by the analyses implemented. We revised the manuscript accordingly. It is true that we do not directly quantify how the different categories of ET products perform. The accuracy rank of individual products against EC observations is: PMLV2 > ETMonitor > MOD16-STM (PM-type models) > GLASS > SynthesisET (data-driven model) > SSEBop (energy balance model). The accuracy ranking against $ET_{wb}$ is: ETMonitor > PMLV2 > MOD16-STM (PM-type models) > SSEBop (energy balance model) > GLASS > SynthesisET (data-driven models) when comparing We can clearly see the best 3 high-resolution products are PM-type models (Section 3.1 and Section 4.1).. The information on each product is based on literature as described in section 2.2.3 and it helps explaining the ranking.

- the interpretation of the split of the ET data into the different components is also a bit of a stretch. As these datasets are far from validated, the only conclusion that can be drawn is that these sub-datasets are likely in the ranges provided by the different datasets.

**Reply**: This ET components have not been fully investigated in previous studies and this limits our understanding of the vaporization process. We agree with you that the evaluation of different ET components was limited in this study due to the scarcity of observations on ET components. We did not state that our analysis documents which data product estimates better the ET components, but we can, for example, draw the conclusion on the basis of a land cover map that soil evaporation accounts for the largest share of ET in the TP and it varies considerably among data products and regions. This actually answers the second proposed question in the introduction, i.e. "which processes play a significant role in determining the total ET".

specific comments:
- since you are evaluating 22 different data products the manuscript is difficult to read as in tables/graphs the order of the products keeps shifting around. Also the addition of the methods in table 3 is appreciated, the link to the discussion is not clear (better to classify the products into the same categories as you use in the discussion)

**Reply**: Thank you, and sorry for the inconvenience for reading. We checked and revised the order of products in all figures following the order in Table 3. The categories mentioned in the discussion are described in the first paragraph of Section 2.2.3.

- ETMonitor in the discussion is classified under PM methods, however from table 3 it appears only Ew and Ess are obtained using PM, but not the main components (Shuttleworth-Wallace) , also GLEAM uses Priestley-Taylor but this data set is described under PM methods

**Reply**: Both Priestley-Taylor equation and Shuttleworth-Wallace equation are developed based on the Penman-Monteith equation. So, we grouped all of them as PM-types models. This is explained in Section 2.2.3.

- please update the text to mention that for the EC validation only 7 data products were considered (eg this is not clear in line 382)

**Reply**: We stated that 7 data products were validated by EC observations in Section 2.3.1. We delete this paragraph including line 382 according to your following suggestion (first paragraph of the discussion reads like a summary, that is not the purpose of a discussion section).

- line 270 (caption) what do you mean with the 'data from all five basins were used together'? does it mean you merged the area for the five basins and compared the WB with the total outflow of all these five basins or you appended the timeseries for all basins to calculate the performance indicators? If the first then how did you deal with the short time series for the Heihe river basin?

**Reply**: We use the second one, i.e., pooling the ET timeseries for all basins to calculate the performance indicators. We kept the evaluation of the WB separate for each basin. We revised the wording to make it clear.

- line 278, what do you mean with pixel-wise ET?

**Reply**: We mean each pixel in the images.

- line 279 how do you know the two peaks are related to the non-vegetated /sparsely vegetated area (did you evaluated the ET values per land use classes?)

**Reply**: We deleted this sentence to avoid potential conflicts.

- line 282 I think you mean figure 4b (when presenting more than one figure with one caption, number them, especially if in the text you are referring to only one of the figures presented)

**Reply**: Thank you. We revised the references to figures and subfigures.

- line 282-283 "large differences" if you look at the SD/mean the differences are about 10-20% do you really consider this large?

**Reply**: We intended to show that differences across products in the central and western TP are much larger than in the eastern TP. To avoid any potential misunderstanding, we revise it to 'differences among different products were larger in the central to western TP than in the eastern TP'.

- line 285 'overestimation in arid regions' you havent explained which basins/regions are located in arid climates, then how do you know there is an overestimation in those areas? Plus overestimation

compared to what? what is the baseline?

**Reply**: We revised this sentence to 'illustrating the relatively larger uncertainty in arid regions'. The regions/basins located in arid climate are detailed in Section 2.1.

- line 290 average ET 'over different products' dont you mean 'of all products'?

**Reply**: Yes, we revised it accordingly.

- caption of figure 5 the last sentence is disconnected from the previous sentence. Also not clear how much of the area is missing in the regional products (excl areas outside of China)

**Reply**: The missing area roughly accounts for 17% of the TP. We revised the sentence to 'It should be noted that some products do not have full spatial coverage, e.g., MOD16, FLUXCOM-RS and FLUXCOM-RS-Meteo only provide ET values for the vegetation-covered regions, and two regional products, i.e. MOD16-STM and PMLV2-Tibet, do not cover the regions outside China, accounting for 17% of the TP roughly.'

- first paragraph of the discussion reads like a summary, that is not the purpose of a discussion section

**Reply**: Agree. We deleted this paragraph.

- line 390-396 this section is not very relevant, so you used two different approaches, this has been done many times before with comparison with in-situ data and water balance as the most used approaches (see section 4.1 of this paper https://hess.copernicus.org/articles/27/4505/2023/)

**Reply**: Thank you. We deleted this paragraph.

- the following paragraph only evaluates the products that were validated both with the in-situ data and the WB method, but you are completely ignoring the other ET products. How does this limit the conclusions you are drawing?

**Reply**: It true that this section is based on the 7 products validated by both methods. Considering these products are the main-stream global ET products with high resolution, we think the conclusion here is well documented.

- rest of the same section: I am failing to understand how you can relate the results of the validation to the actual processes happening

**Reply**: A critical validation requires that the validated data and ground-truth data represent exactly the same ET processes. This may not be always the case, however. We noticed that depending on the definition of ET (e.g., some include canopy rainfall interception loss while some not) and the algorithms, different ET products may represent different processes. For example, some algorithms constrained Rn to positive values to avoid negative ET, which is different from the eddy covariance observation, which can also capture negative latent heat flux due to condensation and deposition.

- paragraph starting line 420. What about the energy balance closure of the EC towers?

**Reply**: To reduce its impact, energy closure correction was conducted before validation for EC observed half-hourly latent heat flux data, as descripted in Section 2.2.1. In this study, the overall

energy closure ratio is 0.83 (±0.30). The slope (LE+H against Rn−G based on all the available half-hour data) was less than 1 for all site, ranging from 0.44 to 0.83, with a mean of 0.66. This energy closure ratio lays in the range of previous studies (Wilson et al., 2002; Mauder et al., 2024), who showed that the mean imbalance of FLUXNET observations is in the order of 20%.

Reference:

Wilson K., Goldstein A., Falge E., et al. 2002. Energy balance closure at FLUXNET sites. Agricultural and Forest Meteorology, 113(1–4), 223-243.

Mauder M., Jung M., Stoy P., Nelson J., Wanner L. 2024. Energy balance closure at FLUXNET sites revisited. Agricultural and Forest Meteorology, 358, 110235.

- line 432: did you consider storage changes?
**Reply**: Yes. As described in Section 2.2, when estimating ET using the water balance, changes in terrestrial water storage were derived from GRACE data.

- line 447 'we found that PM models incorporating soil moisture' is the incorporation of soil moisture the only difference between these models? you can only make such statements if/when you create two datasets using the same method one with and one without soil moisture.
**Reply**: Thank you. Some models (e.g, ETMonitor and GLEAM) require soil moisture as input, thus we cannot remove soil moisture from the model. We revised the text accordingly. It is revised to 'For instance, to improve the accuracy of ET estimates, ETMonitor, which achieves high accuracy in the TP, uses high-resolution soil moisture data to refine the parameterizations of soil and canopy surface resistances to estimate soil evaporation and plant transpiration'.

- line 490, but arent these products specifically made for climate studies?
**Reply**: Yes. We deleted this sentence.

- line 535-565 how is this all based on the research presented in this paper?
**Reply**: We understand your concerns. This section was intended to discuss the reasons that cause the discrepancy in the ET components across different datasets as shown in Section 3.3. Although we do not have direct results to evaluate the impact of these factors, this discussion will inspire our further study to explore the ET components and contribute to better ET partition methods.

**2. Reply to the comments by REVIEWER #2**

The revised manuscript by Zheng et al. is streamlined and conveys the scientific message more clearly than the previous version. The authors compare different ET products regarding their usability in the Tibetan Plateau and find that high-resolution datasets perform better than others, despite significant discrepancies between models. In addition, models incorporating dynamic vegetation cover and water stress modules to predict ET outperform those without these features. Although land surface temperature-based remote

sensing models generally perform worse, they tend to do well in arid regions. Additionally, regional parameterization may be key to improving ET products for specific regional applications. These findings add value to the existing knowledge about the strengths and weaknesses of ET products and their usability in the Tibetan plateau. Nevertheless, the paper is quite lengthy, and I believe it could be made more concise to enhance clarity and readability. I suggest making these revisions before publication and recommend this to the editor.

**Reply**: We thank you for the positive and constructive feedback that helps us to improve our work.

Please find some of my suggestions here:

1) I believe it is necessary to include a land cover map in the supplementary section (omit if already added) and refer to it when justifying certain results, such as in lines 242-244, 282-286, and 291-293. Currently, land cover is introduced in Table 1, which does not align with the descriptions in the justifications, such as "densely" or "sparsely" vegetated, or "arid."

**Reply**: A land cover map was included in the supplementary materials, which was extracted from the ESA CCI global land cover map. The ESA CCI land cover classification is also added in Table 1, while the actual land cover according to the field survey is shown within the brackets. We also checked and revised the manuscript to make the description of land cover classes consistent. The multi-year averaged NDVI map was also added in the supplementary materials to have a better illustration on the vegetation condition. The dense vegetation cover (generally with high NDVI) is located mainly in the eastern TP, and sparse vegetation cover or bare land (generally with low NDVI) located in the central and western TP.

2) For Section 3.2.2, "Temporal Variability in ET Across the TP," since a basin-wide analysis is provided, I believe more emphasis should be placed on basin-wide discrepancies. The following key points could be addressed in the text, but are not limited to:

a) Please clarify whether the analysis was conducted for every product in each basin or if only the median of the products was analyzed across basins.

**Reply**: We revised accordingly. We conducted the analysis for every product in each basin. But, it would have been too detailed and possibly confusing to present the analysis in detail for each basin and every product. As regards the monthly variability, we described the seasonal evolution of ET in different basins (Fig.6). As regards the yearly variability, we only focus on the statistics for the entire TP and we provided the results for each basin in the supplementary materials.

b) Are there any variations in the profiles between the basins? If so, to what extent?

**Reply**: Yes. As stated in the second paragraph of section 3.2.2: "At the basin scale, the difference in annual trends between different products is also clearly illustrated (Supplementary Figure S6). Most basins showed a significant increasing trend of ET, especially in the Yellow, Yangtze, Mekong, Tarim, Hexi Corridor, Tarim, and Qaidam basins, where most products had a positive ET trend. The median ET trend is either negative or close to zero in the Ganges, Brahmaputra, Amu Darya, and Inner TP basins, probably indicating a decreasing or non-monotonic trend for these basins.".

c) How do the basins differ from each other in terms of land cover or other characteristics?

**Reply**: The characteristics of each basin are described in Section 2.1. The five basins (Hexi, Tarim, Qaidam, Amu Darya, and Inner TP) located in the northern, western, and central parts of the TP receive relatively low precipitation with arid or semi-arid climate. These basins are generally covered by sparse vegetation or bare land (Supplementary Figure S1). The remaining basins receive high precipitation, and they are characterized by relatively dense vegetation

d) Do all basins, regardless of land use, exhibit the same level of uncertainty?

**Reply**: Surely no. The uncertainty of each basin is detailed in Supplementary Table S2. As stated in Section 3.2.1, "The uncertainty on ET is highest in basins with low ET and sparse vegetation cover, i.e. the Qaidam, Inner TP, Hexi Corridor, Tarim, and Amu Darya basins. The uncertainty is expressed as the ratio of standard deviation to the mean values (Supplementary Table S2). Uncertainty is also high in the Indus and Brahmaputra basins, most likely due to their complex topography, extreme altitude range, and large areas of permanent glaciers and snow, which make it difficult to obtain reliable estimates.".

3) Line 76-79: In my understanding, basin-scale ET estimates represent the net water vapor flux from land to atmosphere (whether positive or negative), as they are derived from the water balance. Hence, they differ from in situ measurements, which capture both upward and downward water vapor flux. So, sentence needs restructuring.

**Reply**: We revise it to 'These evaluations have generally been based on either in-situ measurements using eddy covariance systems or basin-scale ET estimates using water balance method. The *in-situ* eddy covariance measurements at 30min temporal resolution capture both upward and downward water vapor flux at site scale, and the integrated daily or monthly ET depended on whether the upward water vapor flux is included. Water balance estimates capture the net liquid water flux at the surface and at basin scales, while ET products are estimates of the upward water vapour flux, unless separate estimates of condensation and deposition are provided. This difference contributes to the uncertainty.'.

4) Lines 85-89 and 90-94 convey the same meaning and can be merged for conciseness.
**Reply**: Thank you. Revised accordingly.

5) Figure 2: Please indicate overlap meaning consistent time frame, 2003-2013, it it's the case.
**Reply**: Revised accordingly.

6) Line 279-280: Cross reference missing.
**Reply**: Revised accordingly.

7) Figure 9: These differences, at least in sublimation, could be due to how snow extent is treated in each model. For instance, it seems that GLEAM considers snow extent differently than ETM, or could this simply be an effect of plotting?

**Reply**: It is true that the difference of snow/ice sublimation can be partly explained by differences in the input snow cover extent or fraction. ETMonitor use MOD10 snow/ice cover data at 500m resolution as the boundary conditions for sublimation estimation at global scale. If a pixel is covered by snow/ice at certain day according to MOD10 date, sublimation is estimated by ETMonitor. GLEAM use GLOBSNOW daily snow water equivalent at 25km resolution for the Northern Hemisphere and NSIDC monthly snow water equivalent climatology product at 25km resolution for the Southern Hemisphere.

8) Section 4.1 Title The paragraph does not align with the content, as it does not address which vaporization processes are more relevant. Instead, it focuses on the differences between processes captured in in-situ and basin-wide data, and how these differences impact ET comparisons.

**Reply**: We revised the title to 'Contribution of the study to a better understanding of the vaporization processes.

9) Line 417-421: I find this quite interesting. Nice finding.

**Reply**: We thank you for positive feedback.

10) Line 437-439: Is not PM also an energy balance model?

**Reply**: It is true that PM equation considers the surface energy balance since PM equation fundamentally combine the energy balance and the mass transfer principles. The wording "surface energy balance model" generally refers to algorithms like SEBS or SSEBop that utilize land surface temperature to estimate sensible and latent heat fluxes.

11) Section 4.2.3 Title Since this section does not only incorporate ensemble ET products, please consider changing title here.

**Reply**: We deleted 'ensemble' in the title.

12) Section 4.3: Page 26: I believe this section could be condensed. While I enjoy reading this, it doesn't seem to align with the focus of the study and could be made concise without writing in such details. Alternatively, for each model used, a supplementary table could be created to detail differences in forcing, model structure, calibration, spatial heterogeneity, and other relevant factors and refer it.

**Reply**: Thank you for positive feedback. The section is meant to provide information useful to understand some of the observed differences in the accuracy and ET partitioning of data products. In any case we revised this section to condense it.

13) Line 582- 588: I believe this resembles the previous version, as the information comes across abruptly to the reader. Since the related section has been removed in the current version, it now lacks the justification behind the statement.

**Reply**: We deleted these sentences.

**3.  Reply to the comments by REVIEWER #3**

**Reply**: We thank you again for the review and the constructive feedback that helps us to improve our work.

'2020), Recent studies', here should be "." not ","
**Reply**: Revised accordingly.

"certainly contributes to the understand of the ET process" should be "certainly contributes to the understanding of the ET process"
**Reply**: Revised accordingly.

"Calibration of model parameter" should be "Calibration of model parameters."
**Reply**: Revised accordingly.

We would like to thank the reviewer for their constructive comments on our work that enriched our manuscript.

With kind regards,
Chaolei Zheng, on behalf of co-authors